# A place-based assessment of biodiversity intactness in sub-Saharan Africa

Hayley S. Clements[1,2,3 ✉], Reinette Biggs[1,4], Alta De Vos[1,5], Emmanuel Do Linh San[6], Gareth P. Hempson[7,8], Birthe Linden[9,10], Bryan Maritz[11], Ara Monadjem[12,13], Chevonne Reynolds[7], Frances Siebert[14], Nicola Stevens[7,15], Matthew Child[16], Enrico Di Minin[1,2,17,18], Karen J. Esler[19], Maike Hamann[1,20], Ty Loft[15], Belinda Reyers[21], Odirilwe Selomane[1,22], Geethen Singh[7,23,24] & Andrew L. Skowno[16,25]

Maintaining biodiversity is central to the sustainable development agenda[1]. However, a lack of context-specific biodiversity information at policy-relevant scales has posed major limitations to decision-makers[2,3]. To address this challenge, we undertook a comprehensive assessment of the biodiversity intactness of sub-Saharan Africa[4] using place-based knowledge of 200 African biodiversity experts[5]. We estimate that the region has on average lost 24% of its pre-colonial and pre-industrial faunal and floral population abundances, ranging from losses of <20% for disturbance-adapted herbaceous plants to 80% for some large mammals. Rwanda and Nigeria are the least intact (<55%), whereas Namibia and Botswana are the most intact (>85%). Notably, most remaining organisms occur in unprotected, relatively untransformed rangelands and natural forests. Losses in biodiversity intactness in the worst-affected biomes are driven by land transformation into cropland in grasslands and fynbos (Mediterranean-type ecosystems), by non-agricultural degradation in forests and by a combination of the two drivers in savannas. This assessment provides decision-makers with multifaceted, contextually appropriate and policy-relevant information on the state of biodiversity in an understudied region of the world. Our approach could be used in other regions, including better-studied localities, to integrate contextual, place-based knowledge into multiscale assessments of biodiversity status and impacts.

Biodiversity is an integral part of sustainable development[2,3]; however, we are rapidly losing biodiversity[1] and failing to embed biodiversity management in policy and planning[6]. One major limitation to achieving conservation goals is the lack of information on the impacts of diverse human activities on biodiversity and resulting ecosystem functions and services[2,3]. To be useful in national and international decision-making, such information needs to be comparable across spatial and temporal scales and capture changes in biodiversity relevant to sustaining societies and economies[2,3]. Ecosystem condition or integrity represents the degree to which the composition, structure and function of an ecosystem resembles that of its reference state[7]. This biodiversity metric is increasingly being adopted in multilateral environmental agreements (for example, the 2030 Global Biodiversity Framework (GBF)) to address these needs. The biodiversity intactness index[4] (BII) is an indicator of ecosystem condition that holds promise for mainstreaming biodiversity into policy and planning[7]. The BII assesses human impacts on the abundance of a wide range of species that contribute diverse functions and capture the multidimensional nature of biodiversity in a way that can be compared across multiple scales and time periods[4,8].

However, the limited availability of appropriate data to quantify indicators such as the BII is a major constraint to decision-making, especially in the Global South[1,9]. Available assessments of ecosystem

[1]Centre for Sustainability Transitions, Stellenbosch University, Stellenbosch, South Africa. [2]Helsinki Lab of Interdisciplinary Conservation Science, Department of Geosciences and Geography, University of Helsinki, Helsinki, Finland. [3]African Wildlife Economy Institute, Stellenbosch University, Stellenbosch, South Africa. [4]Stockholm Resilience Centre, Stockholm University, Stockholm, Sweden. [5]Department of Environmental Sciences, Rhodes University, Makhanda, South Africa. [6]Department of Biological and Agricultural Sciences, Sol Plaatje University, Kimberley, South Africa. [7]School of Animal, Plant and Environmental Sciences, University of the Witwatersrand, Johannesburg, South Africa. [8]School of Biodiversity, One Health and Veterinary Medicine, University of Glasgow, Glasgow, UK. [9]Department of Life Sciences, Aberystwyth University, Aberystwyth, UK. [10]Faculty of Science, Engineering and Agriculture, University of Venda, Thohoyandou, South Africa. [11]Department of Biodiversity and Conservation Biology, University of the Western Cape, Bellville, South Africa. [12]Biological Sciences, University of Eswatini, Kwaluseni, Eswatini. [13]Mammal Research Institute, Department of Zoology and Entomology, University of Pretoria, Pretoria, South Africa. [14]Unit for Environmental Sciences and Management, North-West University, Potchefstroom, South Africa. [15]Environmental Change Institute, School of Geography and the Environment, University of Oxford, Oxford, UK. [16]South African National Biodiversity Institute, Cape Town, South Africa. [17]Helsinki Institute of Sustainability Science, University of Helsinki, Helsinki, Finland. [18]School of Life Sciences, University of KwaZulu-Natal, Durban, South Africa. [19]Centre for Invasion Biology, Department of Conservation Ecology and Entomology, Stellenbosch University, Stellenbosch, South Africa. [20]Centre for Geography and Environmental Science, University of Exeter, Penryn, UK. [21]Centre for Environmental Studies, University of Pretoria, Pretoria, South Africa. [22]Department of Agricultural Economics, Extension and Rural Development, University of Pretoria, Pretoria, South Africa. [23]Centre for Invasion Biology, Stellenbosch University, Stellenbosch, South Africa. [24]Fynbos Node, South African Environmental Observation Network, Centre for Biodiversity Conservation, Cape Town, South Africa. [25]Department of Biological Sciences, University of Cape Town, Cape Town, South Africa. ✉e-mail: hayleyclements@sun.ac.za

condition are criticized for being top-down; that is, based on global, decontextualized pressure–impact relationships that extrapolate across data-poor regions and taxa[10–12]. These assessments can have lasting consequences for planning and prioritization[10]. For example, global assessments of ecosystem condition typically do not differentiate between planted pastureland and untransformed rangeland—a key distinction in the context of sub-Saharan Africa where rangelands predominate[13,14]—and the validity of such assessments has been questioned[15]. Large tracts of the supposedly degraded rangelands of the region are inappropriately identified for 'restoration' through tree planting, which can undermine both biodiversity and livelihoods[16,17]. At the same time, the fastest-growing human populations on Earth are in sub-Saharan Africa[18]. Moreover, the ecosystems of the region are undergoing rapid transformation that could compromise sustainable development into the future in the absence of more context-appropriate biodiversity information to support policy and planning[19,20].

The Intergovernmental Science-Policy Platform on Biodiversity and Ecosystem Services (IPBES) calls for regional biodiversity information to close these knowledge gaps[1,19]. Place-based knowledge, which encompasses diverse forms of knowledge—including scientific, experiential and local—is rooted in specific landscapes and contexts and an important potential source of regional biodiversity information[21]. Mobilizing such knowledge to inform sustainability policy and planning requires approaches that retain social and ecological specificity while enabling comparisons across broader scales[12,22]. Here we demonstrate a robust approach to mobilizing place-based knowledge to assess the biodiversity intactness of sub-Saharan Africa, one of the most poorly represented regions in global biodiversity datasets and assessments[9]. Such regional assessments can serve as a bridge between place-based and global sustainability assessments to overcome cross-scale integration challenges through contextualized generalizations[21,22].

Our bottom-up approach overcomes critical data gaps and limitations of top-down biodiversity models by quantifying biodiversity intactness using the Biodiversity Intactness Index for Africa (bii4africa), a dataset that we previously co-produced and published with 200 experts in African fauna and flora[5]. These experts embody place-based African biodiversity knowledge, which holds credibility, legitimacy and saliency for mainstreaming into national decision-making[23] and contributes to inclusivity and decoloniality in science[24]. The bii4africa dataset[5] contains standardized estimates by experts of the impact of the predominant land uses in sub-Saharan Africa on diverse functional groupings of species that represent around 50,000 terrestrial vertebrates and vascular plants. Here we integrate ten spatial datasets to map these land uses, which we combine with bioregional lists of indigenous taxa and the associated bii4africa data[5] to map the BII across sub-Saharan Africa (Extended Data Fig. 1).

## Assessing biodiversity intactness

The BII indicates the average remaining proportion of intact populations of indigenous species in a particular area given the dominant human land uses and activities[4]. Intactness is defined relative to a reference state: typically, before alteration by modern (industrialized, colonial and post-colonial) society, with large protected or wilderness areas serving as a contemporary reference. The index accommodates as wide a range of species as possible, with all species weighted equally. The BII is spatially explicit and standardized on a scale from 0 to 100%, which reflects completely transformed to intact areas and has the same meaning at all scales to facilitate comparative assessments.

In addition to being an indicator of ecosystem condition, the BII has been proposed as a measure of the planetary boundary 'functional biosphere integrity'[25]. Our approach aligns with suggestions to address challenges with biosphere integrity in the planetary boundaries framework[26,27], including assessments of biome integrity and BII at regional

(as opposed to global) scales[8,26,28]. We significantly advance the original BII approach[4] by using data produced through a structured expert-led process to improve rigour[29], including data from many more experts, considering the impact of land-use intensity[30] and disaggregating the BII into more nuanced functional groups of species[5]. Our comprehensive assessment of the biodiversity intactness of sub-Saharan Africa provides insights at policy-relevant scales into the human activities that are contributing to the retention and loss of biodiversity across countries, ecoregions and biomes.

## Biodiversity intactness of the region

Sub-Saharan Africa has a current estimated biodiversity intactness of 76% (±14%; Fig. 1). This means that indigenous vertebrate and plant populations across the region have on average declined to 76% of their intact reference abundances. The BII of vertebrates is 71% (±9%), which is lower than that of terrestrial vascular plants (79 ± 17%). Mammals have experienced the greatest losses, whereas graminoids (grasses, sedges and rushes) and forbs (non-graminoid flowering plants with no or limited aboveground lignification) have experienced on average the lowest losses (Fig. 1). All reported uncertainties around BII values are based on 95% confidence intervals around average expert estimates of intactness in the bii4africa dataset[5].

There is high variability in intactness among functional groups of mammals, which range from 20 to 82% (Extended Data Fig. 2). Large herbivore and carnivore species (>20 kg) have experienced the greatest declines in abundance (BII = 20–52% and 25–51%, respectively), followed by primates (46–65%). These groups are relatively low in species richness and therefore contribute less towards total intactness compared with the more species-rich orders of bats (BII = 64–80%), insectivores (64–74%) and rodents (61–82%), which have retained on average almost double the intactness of larger mammals. There is less marked variability in intactness in the other vertebrate taxa, with birds ranging from 47 to 85%, reptiles from 56 to 77% and amphibians from 55 to 74%. Forest interior and cavity-breeding large savanna birds have been the most affected (BII = 47% and 58%, respectively), whereas grassland birds (except for ground nesters) and aerial feeders have been the least affected (82% and 85%, respectively). Among reptiles, chelonians and large specialist snakes and lizards have experienced the largest declines (BII = 56% and 57%, respectively), whereas small generalist snakes and lizards and rupicolous reptiles have experienced the smallest declines (75% and 77%, respectively). Amphibians that breed in plant or tree hollows or in seep or spray zones have been worst affected (BII = 55%), whereas those that breed in ephemeral streams have been least affected (74%). Plant functional groups have large variability in intactness, ranging from 55 to 91%. Shade-tolerant (forest) and swamp trees and shrubs, together with epiphytes, have suffered the greatest losses (BII = 55–56%), whereas forbs and graminoids that resist disturbance[31] have been the most resilient to land-use changes (≥90%).

## Variation across nations and ecosystems

Twelve out of the 42 countries in sub-Saharan Africa are estimated to have retained >80% of their biodiversity intactness, with Namibia and Botswana having the highest BII (87%; Fig. 2a). Fifteen countries have retained <70% of their BII, with Rwanda (48%) and Nigeria (53%) having the lowest BII. The remaining 15 countries have retained intermediate levels of BII (70–80%). Sierra Leone and Ethiopia are middle of the range (72–73%).

Biodiversity intactness varies considerably across ecoregions, from an average of 37% in the Lowland Fynbos and Renosterveld to 92% in the Etosha Pan halophytics (Fig. 2b). With each species considered equally in the BII, plants contribute more towards BII than vertebrates in most ecoregions given their higher species richness

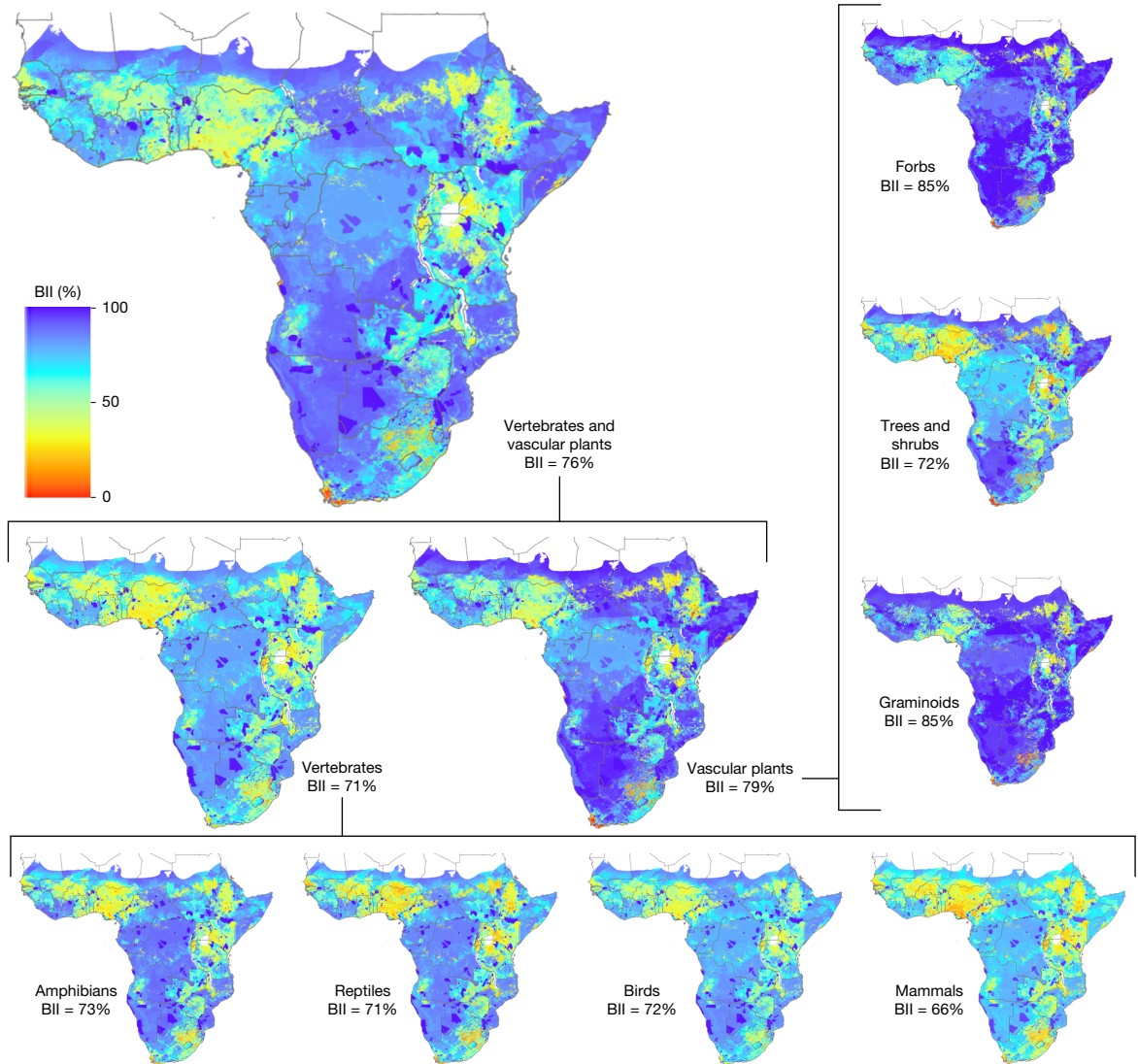

**Fig. 1 | The BII across sub-Saharan Africa.** BII scores for terrestrial vertebrates and vascular plants collectively and disaggregated into the constituent species groups. The overall BII score of 76% for the region shows that on average across all indigenous species, 76% of individuals remain compared with intact (pre-modern industrial society) reference populations. Maps were created using ArcGIS Pro v.2.7.0.

(Extended Data Fig. 3). The exceptions are most desert ecoregions, where vertebrates are more species-rich than plants, and grasslands and *Acacia* savannas, where plants and vertebrates have similar species richness.

Comparisons of the major biomes of sub-Saharan Africa show that BII is highest in the more arid biomes (86% in desert and 83% in shrubland), and lowest in the fynbos (a Mediterranean-type ecosystem and biodiversity hotspot; 56%) and grassland (68%) biomes (Fig. 2b and Extended Data Fig. 4). On average, BII is lower for vertebrates than plants across most biomes (grassland, thicket, humid savanna, *Acacia* savanna and shrubland; Extended Data Fig. 4). The exceptions are fynbos and desert, where vertebrates fare better than plants, and forest, where BII is similar for both species groups.

## The impact of land-use intensity

The average BII is 95% (±8%) across strictly protected lands, 79% (±14%) across unprotected untransformed lands, 48% (±16%) across croplands, 43% (±18%) across tree croplands, 34% (±15%) across settlements and 29% (±17%) across timber plantations (Fig. 3b). The variation in BII

scores in these land uses is caused by spatial variation in both land-use intensity (Fig. 3a) and species composition (Extended Data Fig. 3), as BII varies among species groups (Fig. 3c). All species groups have the lowest levels of intactness in settlements and timber plantations (Fig. 3c). Plants tend to have higher intactness in croplands (for example, maize and wheat) compared with tree croplands (for example, coffee and fruit), whereas reptiles tend to have higher intactness in tree croplands than croplands. There are greater differences in intactness between protected and unprotected untransformed lands for vertebrates compared with plants.

Land-use intensity has a notable impact on biodiversity intactness in the two most extensive land uses. In unprotected untransformed lands, the highest intensity rangelands have an average BII of 51% compared with 85% in the lowest intensity 'near-natural' lands (Extended Data Fig. 5a). In croplands, the average BII is 26% in the highest intensity croplands, which is notably less than in the lowest intensity, smallholder croplands (54%; Extended Data Fig. 5b). The distribution of land-use intensity is right-skewed across sub-Saharan Africa (Extended Data Fig. 5c,d). In other words, non-intensive activities are more common than intensive activities in each of these two land uses, which has a

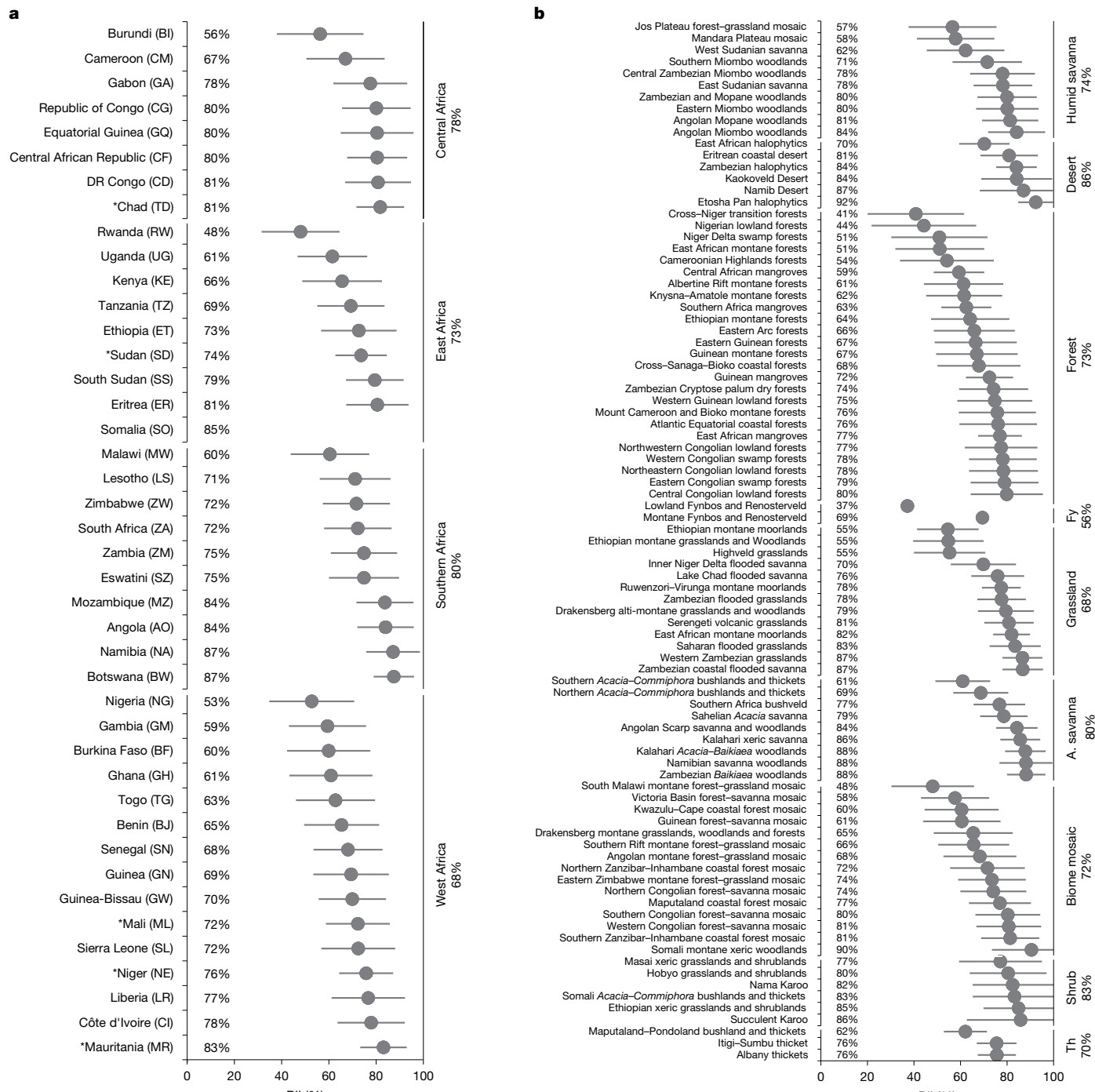

**Fig. 2 | The BII across countries and ecoregions of sub-Saharan Africa.**
**a,b**, Average BII scores are depicted in ascending order for countries per African Union region (**a**) and for ecoregions per biome or biome mosaic (**b**). Uncertainty around average BII values is based on 95% confidence intervals around average expert estimates of intactness in the bii4africa dataset[5]. Asterisks indicate countries that are only partially in sub-Saharan Africa. A. savanna, *Acacia* savanna; DR, Democratic Republic; Fy, fynbos; Th, thicket.

substantial impact on the BII of the region given the extent of these land uses.

## Directing conservation efforts

Our results highlight which land uses make the largest relative contributions to lost and remaining biodiversity intactness, and those that contribute disproportionately given their extent (Fig. 4a). Notably, the majority (84%) of remaining BII across sub-Saharan Africa occurs in unprotected, largely untransformed lands, which cover 80% of the region. Given their vast extent, these areas also contribute the most (68%) to the total BII that has been lost across the region. These findings highlight the critical importance of sustainably managing these areas. Strictly protected lands contribute disproportionately to remaining biodiversity intactness, comprising only 6% of the area of the region but contributing 7% of the remaining BII and just 1% of the lost BII. Croplands contribute a larger amount (9%) to the remaining BII than protected lands, but cover over double the area of protected lands (14%) and are

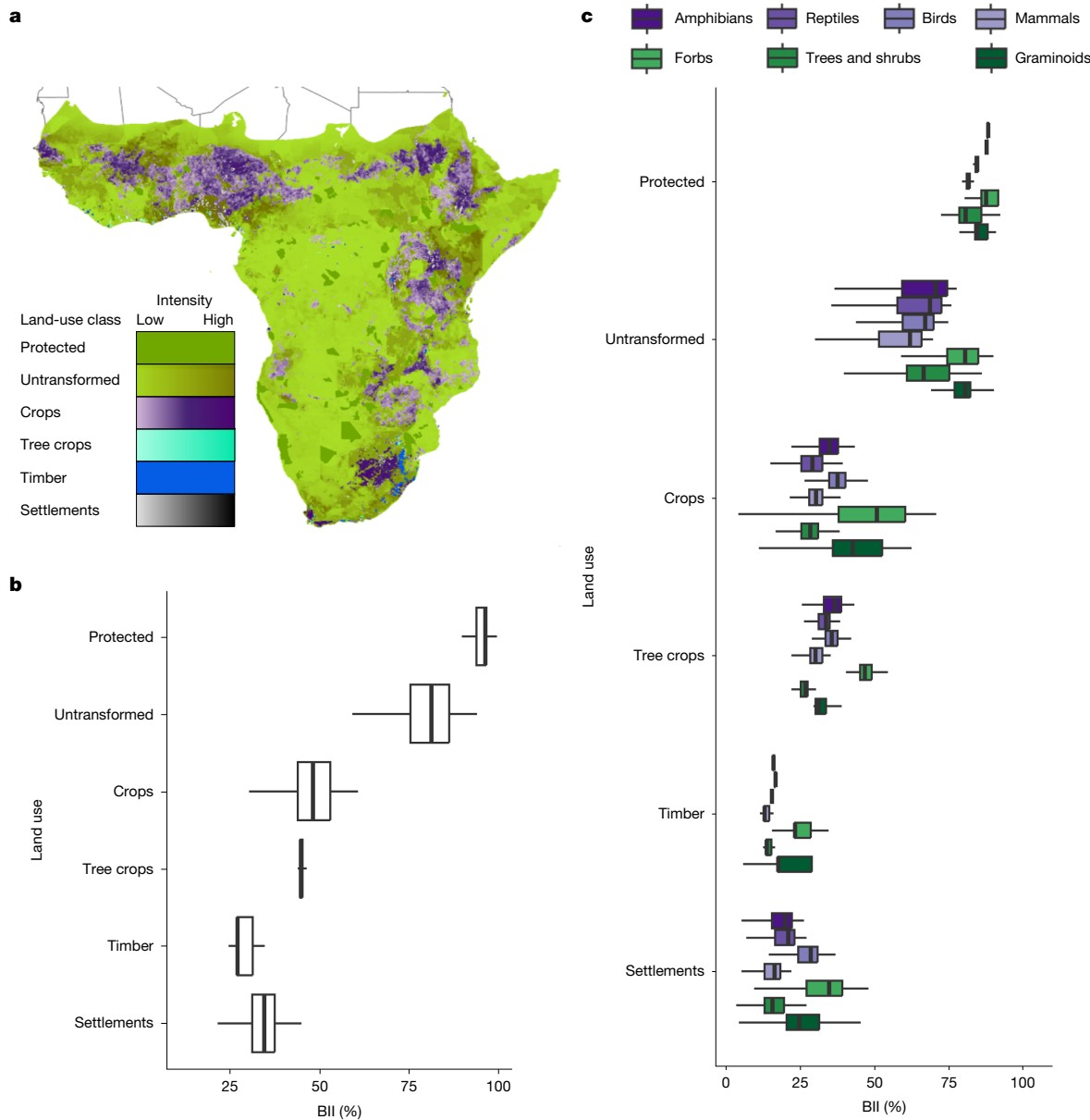

**Fig. 3 | Land use across sub-Saharan Africa and its influence on the BII.**
**a**, Six distinct land uses are predominant across the region, with notably variable intensity in four of these land uses: settlements, tree crops, crops and untransformed (unprotected) lands. **b,c**, Absolute BII scores in each land use for all plants and vertebrates collectively (**b**) and the major species groups (**c**).

Boxplots show median BII scores across pixels, interquartile ranges and maximums and minimums within 1.5× the interquartile range. Variability in the BII in a land use arises from differences in species composition and land-use intensity. The map in **a** was created using ArcGIS Pro v.2.7.0.

responsible for 29% of the lost BII across the region. Settlements, tree croplands and timber plantations each cover <1% of the region, support <1% of remaining BII and are responsible for ≤1% of lost BII, respectively.

At the biome scale, the highest relative contributions to remaining BII are similarly made by unprotected untransformed lands: predominantly near-natural lands in forests, savannas and arid biomes, and rangelands in thickets, grasslands and fynbos (Fig. 4b). In desert and fynbos, strictly protected lands also make major contributions to remaining BII (41% and 23%, respectively), whereas their lower contributions (5–10%) in other biomes largely reflect their more limited extent in those biomes. Croplands are responsible for notable losses in BII across the grassy biomes (grassland, *Acacia* savanna and humid savanna) and fynbos, with less-intensive croplands being more common in the savannas compared with more-intensive croplands (largely in South Africa) in the grassland and fynbos (Fig. 4b). Rangelands are

the major driver of lost BII in the thicket biome (Fig. 4b). Degradation of near-natural lands contributes more to lost BII in forests than in the other biomes. Deforestation to make way for rangelands and croplands also contributes to losses in the forest biome.

As with biomes, countries with a higher proportion of their land transformed (mostly to cultivated lands) tend to have notably lower remaining biodiversity intactness (Fig. 4c). However, there is variability in this relationship. For example, Burundi has the third lowest country-level BII but is notably less transformed than the two countries with the lowest BII (Nigeria and Rwanda; Fig. 4c). The degree of land transformation in Burundi is comparable with Tanzania and Zimbabwe, which have considerably higher BII scores. This variability reflects national differences in the intensity of both transformed and untransformed land and in how the species in a country respond to those pressures.

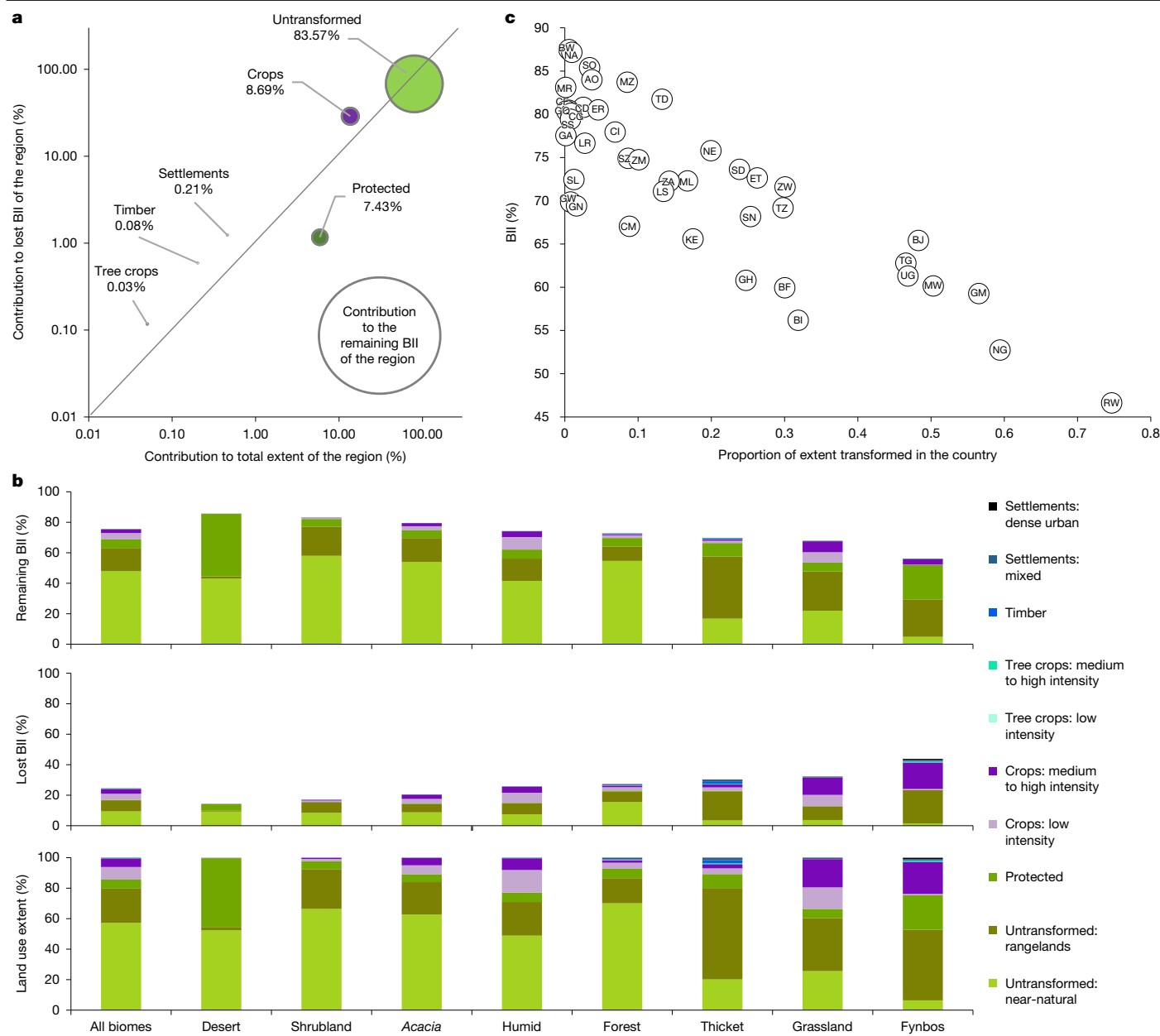

**Fig. 4 | Contributions to remaining and lost BII in sub-Saharan Africa.**
**a**, The relative contributions of six distinct land uses to the total lost (*y* axis)
and remaining (circle size) BII scores across the region compared with their
contributions to the total extent in sub-Saharan Africa (*x* axis). Land uses above
the diagonal line contribute disproportionately to losses relative to their extent.
**b**, The contributions of land uses in each biome to remaining (top) and lost
(middle) BII relative to their extent (bottom), differentiating between low and
medium to high land-use intensities in settlements, tree crops, crops and
untransformed lands. The 'All' bars on the left depict all biomes collectively
(that is, the full extent of sub-Saharan Africa). Individual biomes are otherwise
ranked on the basis of decreasing remaining BII. **c**, The average BII of each
country relative to the proportion of its land extent that is transformed
(that is, covered by settlements, timber, tree crops or crops). Country names
corresponding to each two-letter code referenced here are depicted in Fig. 2a.

## Validation and uncertainties

A challenge with broad-scale biodiversity assessments is the feasibility of performing independent validations to document the degree of error[11], particularly in data-poor regions. Errors can arise from biases of the experts, which we mitigated by adopting evidence-based guidelines to improve elicitation rigour[29]. However, it was not possible to eliminate potential error arising, for example, from knowledge gaps and potential systematic biases in the expert group or from data limitations in our land-use mapping (see the section 'Caveats' in the Methods). Such biases and errors may have had a directional impact on our assessment (that is, leading to consistent overestimation or underestimation). To

assess these potential errors, we critically evaluated our assessment in multiple ways, including the degree of consensus between experts and corroboration between our results and other assessments of human pressure and threat.

The structured expert elicitation process that forms the basis of our results included a critical review of the results by participating experts as a validity check embedded in the process[5,29] (Extended Data Fig. 1). The uncertainty we report around BII scores reflects the degree of consensus among experts, which highlight taxa and land uses for which knowledge is currently more uncertain or disputed; therefore the risk of error may be greater. There was higher uncertainty for plants than vertebrates and for cultivated lands (timber, tree croplands and

croplands) than for other land uses (Extended Data Table 1). These findings can guide future research to close knowledge gaps[4].

Correlations between our BII map and three human-pressure maps followed the expected directions, with lower BII in biodiversity hotspots (areas of exceptional endemism that have lost ≥70% of their primary vegetation[32]; Extended Data Fig. 6a–d). Although these datasets cannot be considered entirely independent of our own, given that they all rely on (imperfect) land-cover data, the existing global BII map[13] showed unexpected relationships with these datasets, which raises concerns about its validity[15]. Although expected correlations corroborated our BII assessment to some extent, the variability is also important. That is, the BII provides insights into how diverse species groups respond to different human pressures and is therefore not synonymous with aggregated human-pressure indices.

When considering the International Union for Conservation of Nature (IUCN) Red List, we found that the threat status of a vertebrate species is a significant predictor of its BII across its sub-Saharan African range (Extended Data Fig. 6e). Critically endangered species have significantly lower BII than endangered, vulnerable and near-threatened species, which in turn have significantly lower BII than least-concern species. These broad trends demonstrate the robustness of our approach, although the Red List may share some of the unknown biases inherent in the BII given the central role of expert knowledge in both assessment processes. Moreover, large within-category variation results in relatively small absolute differences in mean BII between threat categories. A review of Red List assessments for a random sample of outlier species indicated that this variation arises from differences in the purpose of assessments of intactness versus threat of extinction, as well as knowledge gaps (Supplementary Table 1). Such gaps include, for example, assessment inaccuracies for poorly known species, potential BII overestimates for localized species in ineffective de jure protected lands and potential BII underestimates for species prevalent in de facto protected lands. Of note, the BII is not intended as a species-level index, and caution should be exercised when considering species-level results beyond general trends.

Compared with previous assessments of BII (Extended Data Fig. 7), our BII estimate of 80% for the southern African sub-region is a plausible decrease from the 84% estimated for the sub-region in 2005 through a simplified expert elicitation and mapping process[4]. However, both our and the 2005 BII estimates for southern African are higher than the 74% predicted for the sub-region by the global BII model in 2016 (ref. 13). Considering the full region, the global model estimated a higher BII for sub-Saharan Africa (84%) than our approach (76%). These differences are due to the global model estimating lower BII in lower-rainfall biomes (desert, shrubland and *Acacia* savanna) and higher BII in higher-rainfall biomes (forest, humid savanna and grassland) than our assessment (Extended Data Fig. 7c).

Taken together, these diverse comparisons corroborate our BII assessment, which is in contrast to the existing global BII model, which lacks such corroboration[15]. The reported uncertainty around our BII scores gives an indication of uncertainty in the underlying expert scores. We also note that this does not fully account for potential systematic biases and other unknown potential sources of error in our assessment, some of which may be shared with the corroborating datasets.

## Discussion

Our assessment of biodiversity intactness across sub-Saharan Africa integrates the place-based knowledge of 200 experts[5] into a regional measure that can be consistently applied at multiple policy-relevant scales to address longstanding cross-scale integration challenges in sustainability science and practice[3,12,21]. This bottom-up approach accounts for context-specific complexities to help overcome critical data gaps that limit the availability of credible biodiversity information for national policy and planning[20,23,33].

We estimate that sub-Saharan Africa has lost just under a quarter of its pre-industrial biodiversity intactness. A notable finding is that >80% of the remaining wild organisms in the region persist in unprotected and largely untransformed natural forests and rangelands where people coexist with and depend on biodiversity. Conserving and restoring biodiversity, while working towards just and sustainable development, requires a focus on these working lands that sustain more than 500 million people[17,34–36]. Our results indicate nuanced differences in both the threats to biodiversity and the resilience of different species groups to human activity across the region and point to land-use approaches and policies that can support more sustainable coexistence between nature and people.

Large tracts of high-integrity humid forest remain in Central Africa[37,38], which contribute to relatively high BII for Central African countries and ecoregions. By contrast, much of the West African forest is highly degraded[37,38], thereby contributing to very low BII. Degradation of near-natural lands (for example, through faunal and floral overharvesting[37,39]) is a major cause of diminished intactness in these forested ecosystems, with forest-dependent species incurring some of the greatest regional BII losses. West African humid savannas have also been extensively degraded[19,38], with smallholder croplands and rangelands contributing to low BII. Policies that promote sustainable harvesting or alternative livelihood opportunities are key to retaining biodiversity across these systems[19,37,40]. Our findings are congruent with other assessments of degradation across the African tropics[19,37,38] and contrast with the high estimates across West Africa in the existing global BII model[13].

Croplands are regarded as the greatest threat to biodiversity across sub-Saharan Africa[41,42]. The two countries with the greatest crop cover (Nigeria and Rwanda) have the lowest BII scores in the region. In contrast to this finding, the existing global BII model[13] estimates that these highly transformed countries have some of the highest remaining intactness (>90%) in sub-Saharan Africa, with serious potential implications for national and regional policy. We found that biodiversity intactness of the high-yielding intensive croplands in this region—most of which lie in the grasslands—is notably less than its least-intensive, smallholder croplands that are more common in the savannas. These trends have stark implications given that cropland is projected to double and cereal demand to triple in sub-Saharan Africa by 2050 (ref. 43). This increase will probably entail significant changes to current agricultural practices in a region where 75% of the cropland comprises smallholder farming, which have some of the lowest crop yields in the world[43]. Our results highlight the importance of mitigating the impacts of conventional commercial agriculture and incorporating biodiversity-positive elements of traditional smallholder farming systems[44]. These aims are aligned with agroecological approaches promoted under the United Nations Convention to Combat Desertification and other global sustainability frameworks.

Rangelands (primarily used for grazing livestock) are a notable contributor to both remaining and lost BII across the region given their extensiveness. In the small thicket and fynbos biomes of South Africa, rangelands drive the low BII estimates largely as a result of commercial livestock farming[45]. By contrast, lower-intensity pastoralism (livestock herding in search of grazing lands) is a common rangeland practice across other grassy biomes and countries[17,19]. The increasingly restricted mobility and therefore growing intensity of pastoralist practices is of significant concern and requires policy attention to retain biodiversity[17,19]. Rangelands are likely to become a focus of area-based conservation expansion under mechanisms such as other effective area-based conservation measures[46] to achieve the goals of the GBF and United Nations Convention to Combat Desertification. Ensuring such policy measures are ecologically effective and socially legitimate will depend on understanding and supporting governance arrangements, value systems and land-use practices that support biodiversity and people across these landscapes[34,46].

Despite being a major global conservation strategy, we found that strictly protected areas sustain less than 10% of the remaining indigenous plants and animals of the region given their limited extent. Protected areas are particularly important for large mammals, which are especially vulnerable to human impacts[40,47]. Megafauna have BII levels half those of other species groups and are largely absent outside protected areas. However, countries like Namibia and Botswana—with the highest BII in our study—demonstrate that large-mammal conservation can be successfully integrated into broader land governance systems with models such as community conservancies and wildlife economies that promote development by empowering local people as the custodians of biodiversity[48,49]. Given the limited data on the effectiveness of protected areas, we may be overestimating BII across de jure strictly protected areas of the region while underestimating BII in some de facto protected areas. This uncertainty underscores the need to better understand how different conservation models—including community and customary systems—contribute to biodiversity outcomes to inform more inclusive and adaptive conservation strategies under the GBF[46].

Countries around the world are grappling with the pressing need to develop in ways that sustain the ecosystems on which societies depend. In particular in Africa, countries have numerous overlapping international commitments towards just and sustainable development and constraints on credible knowledge to fulfil such commitments[50]. We addressed this constraint by integrating place-based ecological knowledge into a biodiversity assessment that is both sensitive to local context and applicable at broader, policy relevant scales. Beyond addressing knowledge gaps on current biodiversity condition across the region, our results could be used to explore the consequences of future land-use scenarios or strategies to bend the curve of biodiversity loss and contribute to emerging multidimensional biodiversity assessment frameworks that account for the diverse values that underpin nature–human relationships[12]. Our expert-based approach could be used in other data-limited contexts, and can provide advantages in data-rich contexts for which prevailing modelling approaches often miss contextual variation in ecosystem condition[10,15]. This approach could be integrated into global models to act as a bridge between place-based and global assessments and significantly improve our understanding of biodiversity condition around the world through contextualized generalizations. Revealing not only where biodiversity is at risk but also where it is actively sustained, our approach and assessment offer practical and hopeful directions for policy and planning to support inclusive strategies to sustain biodiversity and human well-being[34,46].

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

## Methods

The BII represents the average remaining proportion of the abundance of indigenous faunal and floral populations in an area relative to an intact reference state. In this case, it means pre-colonial and pre-industrial conditions[4]. Calculation of the BII requires three sets of information: (1) a map of human impacts across the area (typically proxied by land use, which captures major land covers, uses and associated activities); (2) the richness of indigenous species that occur in the area; and (3) intactness scores—estimates of the remaining proportion of intact reference populations of these indigenous species under different conditions of human impact—on a scale from 0 (no remaining individuals) to 1 (same abundance as the reference) and, in rare cases, to 2 (two or more times the reference population)[4,5]. For species that thrive in human-modified landscapes, scores can be greater than 1 but not exceed 2 to avoid extremely large scores biasing aggregation exercises[4,5]. The intact reference is the population abundance that would have occurred in the area before alteration by modern industrial society (around 1700 CE)[4]. In most parts of sub-Saharan Africa, this corresponds to populations before the substantial alteration of the landscape triggered by colonial settlement, although we recognize that some declines would already have occurred by this point in time. Because information on species populations from this era is almost non-existent, standard protocol is to reference a remote protected or wilderness area with a natural disturbance regime (a hybrid–historical approach[51]) where necessary[4].

The BII for a unit of land (pixel) is calculated by averaging, across the richness $R$ of indigenous species that should occur in that pixel, the intactness score $I_{sk}$ of species $s$ given the human impacts (that is, land use) in that pixel $k$, such that each species counts equally. That is,

$$BII_{pixel} = (1/R)\Sigma_s I_{sk} \quad (1)$$

The index can accommodate data scarcity. In the absence of intactness scores per species per land use, the BII can use intactness scores that represent groups of species that are expected to respond similarly to human impacts (functional response groups) and species richness information for the broader area (for example, ecoregion)[4]. In such cases, the BII for a pixel is determined as follows:

$$BII_{pixel} = (1/\Sigma_i R_i)\Sigma_i R_i I_{ik} \quad (2)$$

where $I_{ik}$ is the intactness score for functional response group $i$ given the land use in that pixel $k$, weighted by the richness $R_i$ of functional response group $i$ (number of species or proportion of total species) in the relevant area (commonly, the ecoregion).

BII can be averaged across $n$ pixels in an area of interest (for example, continent, biome, ecoregion or country) to provide a single BII score that accounts for the composition of both species and land uses:

$$BII = (1/n)\Sigma BII_{pixel} \quad (3)$$

BII scores can be calculated for all species or a subset (for example, vertebrates, amphibians, among others). Here we made use of equations (2) and (3) to accommodate data scarcity. BII estimates were converted to percentages by multiplying by 100.

### Intactness scores co-produced by experts

Determining intactness scores for indigenous species in sub-Saharan Africa based on field-collected data of population abundances across different land uses (including in comparable intact reference areas) is limited by a lack of appropriate data for most species[20,52]. Instead, we made use of a published dataset of intactness scores that were estimated as part of our Biodiversity Intactness Index for Africa project (https://bii4africa.org/) by way of a structured expert elicitation process that involved 200 experts in African flora and fauna[5]. The bii4africa dataset contains intactness scores ($I_{ik}$ in equation (2)) for species groups representing terrestrial vertebrates (tetrapods: ±5,400 amphibians, reptiles, birds and mammals) and vascular plants (±45,000 forbs, trees and shrubs and graminoids), including all mainland Afrotropical ecoregions[53] across 9 specific land uses of varied intensity (Supplementary Table 2). A detailed description of our elicitation process and resulting bii4africa dataset can be found in our previously published paper[5], which recognizes the contributing experts as co-authors. Here we provide a summary (Extended Data Fig. 1).

A broad definition of expertise was used to identify experts, which was centred on experience of how sub-Saharan species are affected by human land uses. Diverse types of people can have such experience (for example, researchers, field guides, park rangers, conservation practitioners and museum curators). An expert was identified to lead an elicitation for each broad taxonomic group. Lead experts then recruited additional experts that met our definition of expertise, focusing on diversity geographically, taxonomically and professionally to promote elicitation rigour and to mitigate biases[29,54–56]. Additional experts were identified by snowballing among experts where needed to achieve a target group size of about 20 and followed best-practice guidelines[29]. We particularly sought African experts to overcome persistent biases towards Global North experts in ecological literature[57,58]. Biases towards well-published experts, taxa and geographies were challenging to completely overcome. Of the 200 experts who participated, there were more with experience in Southern (55% of experts) and East (33%) than Central (27%) and West (26%) Africa, with expert numbers also varying among taxonomic groups (12–38). Just over half the participating experts had university affiliations. Most resided in the region (72%) and 59% were from the region. All experts had place-based knowledge from the region, which informed the creation of a quantitative dataset through an iterative and interactive expert elicitation process. The process was informed by a published, modified-Delphi 'IDEA' protocol[29] (multiple rounds of individual, independent expert estimation interspersed with group discussion and review), which draws on advances in expert elicitation that have been shown to improve data consistency.

For each broad taxonomic group, participating experts were convened in an online meeting to introduce the aim of the project, the notion of BII and how it is calculated and to explain the task of estimating intactness scores. Experts were given the opportunity to ask questions and voice concerns. It was not practical to ask experts to provide intactness scores for every species exposed to every combination of human activities across sub-Saharan Africa. Rather, we made use of a functional grouping approach[4], asking experts to estimate the intactness of different groups of species ($i$ in equation (2); 'Sp_Groups' tab in the bii4africa dataset[5]) in nine land uses characteristic of the region ($k$ in equation (2); Supplementary Table 2). This contextualized generalization[22] assumes that a functional group of species responds in the same way to the same type of land use across its regional range[4]. The exceptions were the 102 species of large mammals, for which experts decided to provide estimates at species level (with $i$ representing a species in these instances). The intactness scores of plant groups were estimated by experts in each of the eight major biomes in the region (forest, Caesalpinioid-miombo humid savanna, mixed-*Acacia* savanna, grassland, shrubland, thicket, desert and fynbos)[59–61]. Both the species functional response groups and biomes were proposed by lead experts and refined on the basis of feedback from participating experts during the introductory meetings. Experts were instructed to make their intactness estimates at a landscape scale (that is, one or several square kilometres), considering the integrated impact of all characteristics of that landscape on each group of species.

Participating experts were given 2 weeks after the introductory meeting to independently provide their best-guess intactness scores through a survey spreadsheet. Experts estimated on average 155

intactness scores. The survey prompted experts to provide comments relevant to each estimate (for example, assumptions or uncertainties) to mitigate the risk of overconfidence and to detect potential inconsistencies between experts (for example, caused by linguistic uncertainties related to the survey[62]). A discussion meeting was convened in which the aggregated (anonymized) results per functional group and land use were presented to participating experts, based on evidence that group discussion can improve elicitation rigour[29,54,55]. Project and expert leads reflected on key trends and sources of variability. Experts were encouraged to share their experiences when undertaking the survey and to reflect on the aggregated results. We emphasized that the purpose of the discussion was not to reach consensus but rather to reduce uncertainties and biases by interrogating sources of variability, improving the consistency with which experts interpreted the survey instructions and cross-examining reasoning, assumptions and evidence to promote learning[29,54,55]. Experts were given 1 week to independently revise any of their scores based on insights gained in the discussion. Several experts withdrew during the process, stating time limitations or insufficient expertise, with 200 participating to completion. Their final scores are presented in the published bii4africa dataset[5].

An average of 10 (s.d. = 7) experts provided an intactness estimate for each combination of functional response group (or species for large mammals) and land use (and biome for plants)[5]. Here we used mean intactness scores (that is, the average estimate across independent experts for each combination) as the most common form of data aggregation in structured expert elicitation protocols to counter the biases of individual experts[29,54]. The variability in scores among our sample of experts (expressed as 95% confidence intervals) was used to reflect the degree of uncertainty around BII estimates, which arose, for example, from differences in the place-based knowledge of the experts, unknowns or disagreements around how some functional response groups are affected by land uses and variability in the number of experts contributing to each aggregated estimate.

## Species richness across ecoregions

For vertebrates, each species in the IUCN Red List with a sub-Saharan African range was allocated to a species functional response group in the bii4africa dataset[5] by lead experts, with input from the participating experts where needed. This list was coupled with a list of species per ecoregion to determine the number of species ($R$ in equation (2)) in each functional response group $i$ in each ecoregion. Species lists per ecoregion were obtained by intersecting the Ecoregions2017 Resolve map[53] with species range maps available through the IUCN Red List[63] and Birdlife International[64], including historical ranges and extinct species, as the BII is relative to pre-colonial and pre-industrial conditions.

Given the large number of vascular plant species, the bii4africa dataset[5] instead lists the proportion of plant species per biome that occur in each functional response group in the broad groups of forbs, trees and shrubs, and graminoids. Proportional richness across the three broad plant groups in each biome was estimated based on the RAINBIO[65,66] dataset of tropical African vascular plant species distributions. Species in RAINBIO were allocated into the three broad bii4africa plant groups as follows: graminoids, all species in the families Poaceae, Cyperaceae, Juncaceae and Restionaceae; trees and shrubs, all remaining species with the growth form tree, shrub, liana or epiphyte; and forbs, all remaining species with the growth form herb, shrublet or vine. The geolocations of each RAINBIO species entry were overlaid onto the Ecoregions2017 Resolve map[53] to determine the proportion of species in each broad plant group in each ecoregion and associated biome. The proportion of total plant species in each functional response group in each biome could then be determined and multiplied by total vascular plant species richness estimates per ecoregion[67] to estimate plant species richness $R_i$ per functional response group $i$ in each ecoregion.

## Mapping land uses and intensities

The bii4africa dataset[5] contains intactness scores for nine specific land uses to capture the major land cover types, uses and associated activities relevant to sub-Saharan Africa (Supplementary Table 2). These include the following types: (1a) mixed settlements; (1b) dense urban; (2) timber plantations; (3a/4a) non-intensive smallholder croplands; (3b) tree croplands; (4b) intensive large-scale croplands; (5) strictly protected areas; (6a) near-natural lands; and (6b) intensive rangelands. These land uses cover six broad classes, and high-intensity and low-intensity options for those classes that can vary notably in their intensity in the region (Extended Data Fig. 8). To map BII, we needed a land-use map that reflected these six broad land use classes and the spectrum of intensities that occur in four of these classes (Fig. 3a).

We used an established decision-tree algorithm[68,69] built on (area) standardized thresholds of human population density and/or land cover to allocate each pixel in sub-Saharan Africa into one of six broad land-use classes to reflect the differences between the land uses described by experts (Extended Data Fig. 8). Settlement pixels (1) were allocated first, followed by timber plantations (2), tree croplands (3) and croplands (4), and thereafter protected pixels (5)[68,69]. All remaining pixels were then allocated to the unprotected untransformed class (6).

In four out of the six broad land-use classes, we then scored each pixel based on its intensity, using a protocol comparable with other well-established human intensity indices[70]. Proxies of intensity relevant to each class were selected to reflect the land-use descriptions for which experts estimated intactness scores (Extended Data Fig. 8). Each variable was standardized using minmax scaling, such that the pixel in a land use class with the lowest and highest value scored 0 and 1, respectively. An average of the relevant standardized variables in a land-use class was computed and then rescaled using minmax scaling. This value represented the relative intensity of each pixel in a land-use class (that is, the pixel scoring 0 versus 1 had the respective lowest versus highest mean intensity score in that class). When scaling variables that did not have a natural upper bound and were therefore susceptible to high outlier scores (for example, population density, livestock density or nitrogen input; Extended Data Table 2), the 90th percentile value was used as the maximum in the minmax scaling.

Before scaling livestock (cattle, goat or sheep) density as a proxy for intensity (Extended Data Fig. 8), we accounted for the fact that some areas can support higher grazing pressure. A unimodal relationship has been shown between potential African mammal herbivore biomass and mean annual rainfall (MAR), peaking at about 1,700 kg km$^{-2}$ and around 700 mm MAR[71]. Moreover, potential African mammal herbivore biomass is higher in areas with higher nutrient soils[72]. Hence, we grouped pixels into comparable sets based on their MAR (in 400 mm bins) and soil nutrients (high, medium or low). We then minmax-scaled livestock density in each of these sets to ensure intensity was quantified across comparable areas.

One of the nine land use classes for which experts estimated intactness scores was a well-managed, strictly protected area (Supplementary Table 2). We limited our allocation of this land use to pixels occurring in protected areas with IUCN management categories I–III, as these have strict restrictions on human activities[73]. Four countries in sub-Saharan Africa do not make use of IUCN management categories, with strictly protected areas identified case by case (Supplementary Methods 1). Protected pixels were assigned after settlements and cultivated lands (Extended Data Fig. 8), which meant that any transformed lands in protected areas (that is, ineffective protected areas) were not allocated to the protected land use. We were unable to exclude strictly protected lands that are untransformed but subject to unsustainable resource harvesting—a limitation that means we may overestimate BII in some protected areas. By contrast, BII may be underestimated in lands that are strictly protected in practice but not on paper. The protected area

allocation and therefore BII map can be updated should management effectiveness data become available for all protected areas.

We mapped land uses in Google Earth Engine at a resolution of $1 \times 1$ km and $8 \times 8$ km to reflect the landscape scale at which experts estimated intactness scores. The finer resolution may be useful to some applications but is computationally demanding to work with (>20 million pixels). The coarser resolution is likely to be more appropriate for certain species groups, such as large mammals and wide-ranging birds, which are unlikely to survive in $1\text{-km}^2$ patches of habitat. All land-use variables were obtained from pre-existing map products that spanned the full region to ensure consistency of inputs (Extended Data Table 2). When a map pixel covered more than one unit of a variable layer, overlapping variable values were either summed (in the case of land cover) or averaged, weighted relative to their degree of overlap (remaining variables). Missing values were imputed using the nearest provided value.

## Calculating and mapping BII

BII scores for each pixel in the land-use map were estimated using equation (2) based on the average (across experts) intactness scores $I_{ik}$ for functional response groups $i$ in land use class $k$ (adjusted for intensity in four of the classes (see next paragraph)) and the ecoregion (which influences species richness $R$ per functional response group $i$). Fifteen ecoregions were classified as mosaics categorized by two biomes (for example, the Angolan montane forest–grassland mosaic ecoregion is a mosaic of the forest and grassland biomes)[5], with plant intactness scores for the two biomes averaged for these ecoregions. A BII score was determined for the following categories: (1) terrestrial vertebrates and plants; (2) vertebrates; (3) plants; (4) each of the four vertebrate classes (amphibians, reptiles, birds and mammals) and three broad plant groups (forbs, trees and shrubs, graminoids); and (5) each of 146 functional response groups (with the large mammal species being categorized into the 12 functional response groups proposed in the bii4africa dataset[5]). To reflect uncertainty, we similarly estimated 'lower limit' and 'upper limit' BII scores for each pixel based on the lower and upper limits of the 95% confidence intervals around average intactness scores $I_{ik}$. Lower-limit-BII and upper-limit-BII scores were determined for the following categories: (1) terrestrial vertebrates and plants; (2) vertebrates; and (3) plants.

To calculate BII (and lower-limit-BII and upper-limit-BII) scores in pixels with a land-use intensity continuum (Extended Data Fig. 8), the maximum (low intensity) and minimum (high intensity) intactness scores $I_{ik}$ were averaged and weighted according to the intensity of that pixel. For example, for settlement land-use pixels, those with an intensity of zero were allocated the '1a. Mixed settlements' intactness scores, whereas those with an intensity of one were allocated the '1b. Dense urban' scores (Supplementary Table 2). A pixel with an intensity of 0.5 received an average of the minimum and maximum scores. BII maps were produced in R.

## Analyses and validation

We calculated the BII for sub-Saharan Africa by averaging BII scores across all pixels in the $1 \times 1$ km and $8 \times 8$ km maps. Given the high degree of correspondence (75.53% and 75.54%, respectively), further analyses were performed using the $8 \times 8$ km maps. We quantified the BII per country, ecoregion, biome and land use by averaging scores across all relevant pixels. We quantified the uncertainty around these BII values by averaging upper-limit-BII scores and lower-limit-BII scores across all relevant pixels. The percentage of pixels in each broad land-use class and the average BII of those pixels were used to determine the proportional contribution of each land use to the total BII that had been lost and that remained. We similarly assessed the contribution of 'medium–high intensity' versus 'low intensity' land uses to lost and remaining BII in each biome (pixels with intensity scores ≥0.25 and <0.25, respectively).

We followed a previously described method[15] to assess the robustness of the BII map by comparing it to the human footprint index[74] (HFI)—a composite map of anthropogenic pressure on natural ecosystems—and the biomass modification index[75] (BMI)—a mapped synthesis of estimates of current vegetation biomass relative to that in the same location without human disturbance. We also related BII to the biodiversity habitat index[76] (BHI)—the effective proportion of habitat remaining in a pixel, adjusting for the effects of the condition and functional connectivity of habitat. We would not expect these indicators to perfectly correlate with BII, as the BII accounts for varied responses of different species populations to human pressures and accounts for important regional contextual factors that may be overlooked in the other global indicators, such as smallholder versus large-scale croplands and rangelands versus planted pastures. However, we would expect areas with a higher BII to generally have a lower HFI, lower BMI and higher BHI, and vice versa[15]. To assess these relationships, we calculated the Spearman's rank correlation of the average BII score and the average HFI, BMI and BHI scores across ecoregions. Using the previously described method[15], we categorized each ecoregion as a biodiversity hotspot if the area of the ecoregion overlapped by ≥50% with a biodiversity hotspot (a priority area of exceptional endemism that has lost ≥70% of its primary vegetation)[32,77]. Hotspot ecoregions should typically have lower BII scores than other ecoregions, and we tested this by comparing the average BII scores of hotspots and other ecoregions using a two-sided Mann–Whitney $U$-test.

As an additional validity check, species-level BII estimates were compared to the risk of extinction of each species (according to the IUCN Red List[63]), with the expectation that a valid BII assessment should generally predict lower BII for more threatened species. We quantified the average BII of each vertebrate species (according to its functional response group or at species-level for large mammals) across its regional range. We then ran a generalized linear mixed-effects model (equation (4)), with a Gaussian distribution given data normality, to assess whether the IUCN threat category ('IUCN category') of a species (that is, critically endangered, endangered, vulnerable, near threatened or least concern) was a significant predictor of its average BII across its range ('intactness index'). We included 4,887 sub-Saharan African vertebrate species with a threat category (we excluded 479 data-deficient species and 3 extinct species). We included IUCN threat category as a fixed effect and the range size of the species ('range size scaled') to isolate the effect of IUCN threat category on intactness, independently of the influence of range size (with range size being a consideration in the IUCN Red Listing process). The model also accounted for variation in intactness across taxonomic classes ('Class') and their nested functional response groups ('RG'). These nested random effects capture baseline differences in intactness due to taxonomic and functional groupings, which ensures that the fixed effects are not biased by class-level or RG-specific variability.

$$\text{Intactness index} \sim \text{IUCN category} + \text{range size scaled} + (1 \mid \text{Class/RG}) \tag{4}$$

This model was run using the lme4 package in R. We checked that the response variable and model residuals were normally distributed and that the two fixed effects were not competing for variance in the global model (variance inflation factor close to 1). Range size was scaled and centred to avoid overdispersion and leverage. We ran an analysis of variance to test whether the fixed effects were significant predictors of intactness and post hoc Tukey tests to check for significance between IUCN categories (corrected for multiple pairwise comparisons).

Finally, we compared the results of our BII assessment with those of two previous assessments. First, the original BII assessment that was published for southern Africa in 2005 based on a simplified expert elicitation[4] and second, a globally modelled assessment that was published in 2016 (ref. 13).

## Caveats

People are susceptible to a range of cognitive biases that can erode the quality of expert-elicited data (for example, based on information availability, overconfidence, linguistic uncertainty or groupthink)[56,62,78,79]. It can also be challenging to identify who is an expert[54]. We sought to mitigate such challenges by selecting a diverse group of experts, based on experience rather than status[29,54,55,80], and taking them through a structured, iterative elicitation using an evidence-based protocol for improving rigour[29]. Our data-aggregation approach reduces the effect of expert biases[29], assuming such biases are independent rather than systematic (for example, no groupthink[81,82]). However, expertise was overrepresented for certain geographies (for example, southern Africa), nationalities (for example, South Africa), taxonomic groups (for example, large mammals) and professions (for example, researchers)[5], which may have resulted in unknown systematic biases in aggregated expert scores. Limitations to diverse expert participation probably include digital inequities, lack of incentive for non-researchers to participate in research and language barriers (particularly in Francophone countries). Furthermore, although overconfidence can be mitigated by asking experts to provide an upper and lower plausible estimate together with a best guess[29], we found this to be cumbersome and confusing in a pilot with lead experts, given the high number of estimates. Instead, we mitigated against overconfidence by nudging experts to consider uncertainties and assumptions underlying their estimates, and encouraging critical evaluation during the discussion meeting[29,54,55]. Given the uncertainties inherent in expert elicitation, applications of this BII assessment should take note of the confidence intervals around mean estimates, which reflect known uncertainty among experts, while keeping in mind potential unknown uncertainties such as possible systematic bias in our expert group that are not reflected in these confidence intervals and that may have had a directional effect on our results.

Our approach of contextualized generalization—integrating place-based knowledge into a broad-scale regional product that speaks to national and international decision-making needs—required several epistemological and methodological compromises. Epistemologically, the use of a pre-defined, relative (to a reference state) and bounded notion of biodiversity intactness is influenced by Western ways of thinking about science and ecology[24] and reflects decisions made by the analysts as opposed to the experts[83]. We countered Western dominance by bringing more African expertise into ecology and adding insight around important aspects of regional context that are often ignored while using concepts and metrics that are embedded in international agreements such as the GBF, which countries are required to report to.

Methodologically, to limit the number of estimates that each expert was asked to provide, we assessed BII at the level of functional response groups and land-use categories. We therefore compromised on potentially important variability among species in a functional group or land-use configurations in a category, which may influence BII. Furthermore, our analyses relied on published land-use and species-distribution datasets that may contain classification or coverage limitations. Notably, some land-use activities (for example, harvesting) are harder to map than those that can be seen from space (for example, crop cover), which limited our ability to account for these activities to mapped proxies such as population density and human infrastructure[14,74]. This limitation means we probably overestimate BII where human activities such as harvesting are notably more intensive in a given land use than predicted on average by our experts or mapped intensity proxies, and the converse in places that have notably lower-than-expected exploitation of wildlife due to, for example, taboos against bushmeat. Furthermore, the intensity of rangeland use is particularly challenging to map, and we advanced standard approaches by controlling for soil and rainfall in our translation of livestock density into a measure of land-use intensity in attempt to better account for local context. The accuracy of BII mapping based on our published expert estimates[5] can probably be substantially improved over time with advances in land-use mapping.

Despite these limitations to our approach, the resulting BII corroborated other assessments of human pressure. However, we note that these other assessments cannot be considered entirely independent given that they all rely on land-cover data and its associated limitations. Trends in the BII estimates for individual vertebrate species across their particular ranges were also robust when compared with their IUCN threat status. This comparison demonstrates that despite uncertainties at each step, our approach of functionally grouping species, asking diverse experts to estimate how those groups would be affected by characteristic land uses and mapping those estimates based on land uses and species ranges across the region led to intactness predictions that correspond as expected with an independent (although also expert-informed) assessment of the threat categories of individual species.

## Reporting summary

Further information on research design is available in the Nature Portfolio Reporting Summary linked to this article.

## Data availability

The expert-elicited bii4africa dataset used in this study is available on Figshare (https://doi.org/10.6084/m9.figshare.c.6710463.v1)[5]. Input data on species range maps and threat categories are available through the IUCN Red List (https://www.iucnredlist.org/)[63] and Birdlife International (http://datazone.birdlife.org/species/requestdis)[64]. Input data on ecoregions are available through Ecoregions2017 Resolve (https://ecoregions.appspot.com/)[53]. Input data on plant forms in the RAINBIO[65] dataset are available on GitHub (https://gdauby.github.io/rainbio/). Previous BII assessments to which we compared our assessment are available from the Natural History Museum for ref. 13 (https://data.nhm.ac.uk/dataset/global-map-of-the-biodiversity-intactness-index-from-newbold-et-al-2016-science), and from Oonsie Biggs for the Scholes and Biggs[4] map. The land-use and BII maps generated during this study are available on Figshare (https://doi.org/10.6084/m9.figshare.29773169.v1)[84], and can be visualized and downloaded on a Google Earth Engine App (https://geethensingh.users.earthengine.app/view/bii). Source data are provided with this paper.

## Code availability

The land-use mapping code is available on GitHub and Zenodo (https://doi.org/10.5281/zenodo.17597480)[85], and the R scripts and data for mapping the BII and for producing plots (Fig. 3b,c and Extended Data Figs. 2, 4 and 6e) are available on Figshare (https://doi.org/10.6084/m9.figshare.29773169.v1)[84].

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

**Acknowledgements** This paper is part of the bii4africa project (https://bii4africa.org/), which was funded by a Jennifer Ward Oppenheimer research grant awarded to H.S.C. R.B. received support from the South African Research Chairs Initiative (SARChI) (grant 98766). E.D.M. was funded by the European Union (ERC, BIOBANG, 101171602). Views and opinions expressed are, however, those of the author only and do not necessarily reflect those of the European Union or the European Research Council Executive Agency. Neither the European Union nor the granting authority can be held responsible for them. H.S.C. and E.D.M. would also like to thank the KONE Foundation (project number 202309134). We are grateful to the late R. Scholes for conceptual input into this project.

**Author contributions** H.S.C. led project conceptualization and data co-production, oversaw map development, led the data analyses and wrote the manuscript. R.B. and A.D.V. contributed to initial project conceptualization and provided input during an inception and two feedback workshops. E.D.L.S. (small carnivores), G.P.H. (large mammals and graminoids), B.L. (primates), B.M. (reptiles and amphibians), A.M. (bats, rodents and insectivores), C.R. (birds), F.S. (forbs) and N.S. (trees and shrubs) contributed as lead experts to method conceptualization and results interpretation for their respective taxonomic groups, as well as giving input during a feedback workshop. C.R. also produced some of the figures and assisted with statistical analyses. G.S. produced the land-use map. T.L. produced the BII maps and gave input during a feedback workshop. M.C., K.J.E., M.H., B.R., O.S. and A.L.S. provided input during an inception and feedback workshop. All authors gave feedback on the development of the land-use and BII maps and provided input on multiple iterations of the manuscript.

**Funding** Open Access funding provided by University of Helsinki (including Helsinki University Central Hospital).

**Additional information**
**Correspondence and requests for materials** should be addressed to Hayley S. Clements.

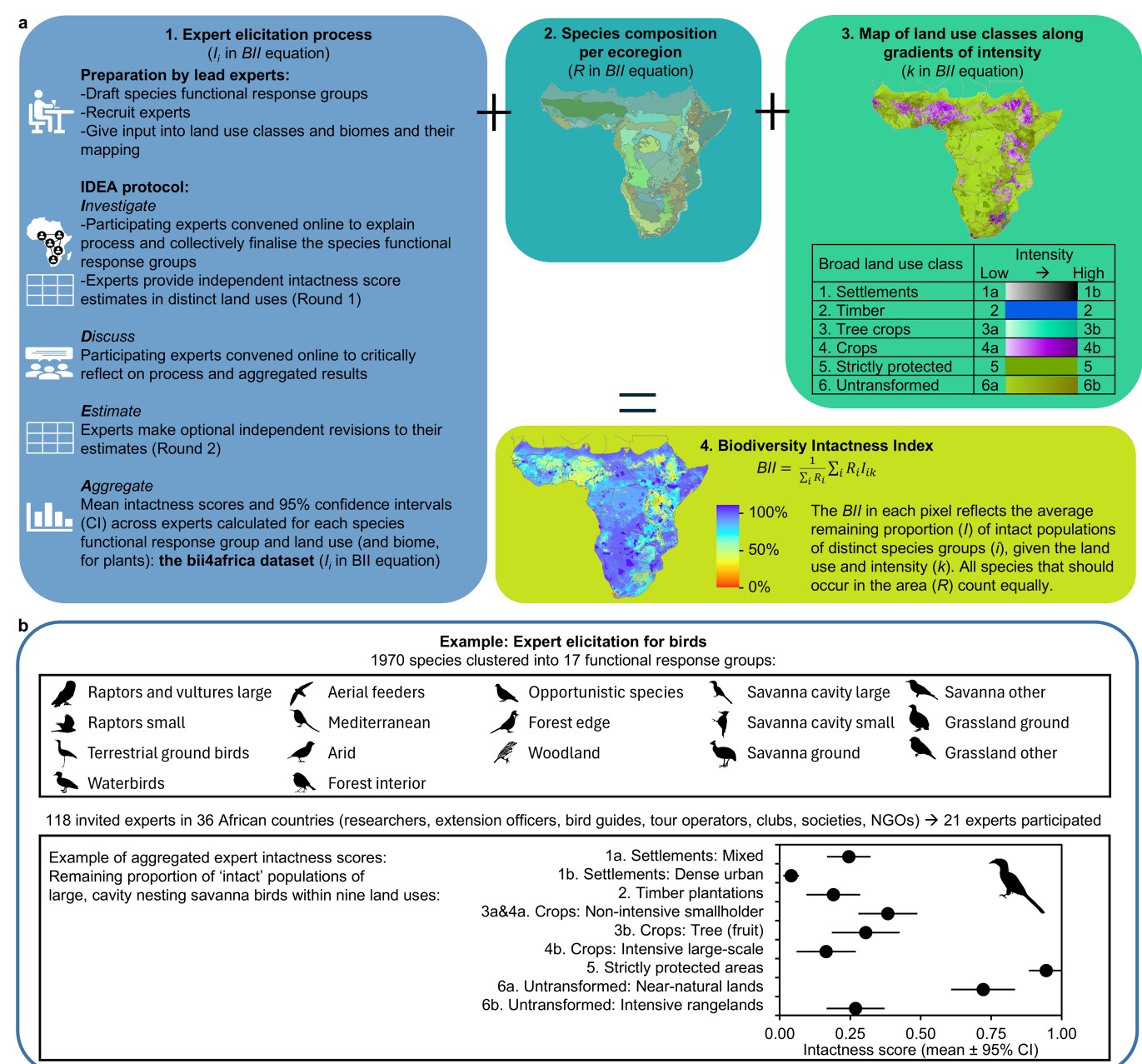

**Extended Data Fig. 1 | Summary of the approach taken to assess sub-Saharan Africa's Biodiversity Intactness Index (BII).** (a) At the core of this approach was (1) a structured expert elicitation process involving 200 experts in African biodiversity to produce the bii4africa dataset[5] of intactness scores estimating the proportion of intact populations of fauna and flora that are likely to remain within nine distinct African land uses. The bii4africa dataset was combined with (2) species composition data and (3) a land use map to quantify (4) the BII. (b) An example of the expert elicitation process to produce intactness scores for birds. Icons in **a** reproduced from Microsoft PowerPoint. Maps in **a** (boxes 2, 3 and 4) were created using ArcGIS Pro v.2.7.0. Silhouettes in **b** obtained from PhyloPic (https://www.phylopic.org).

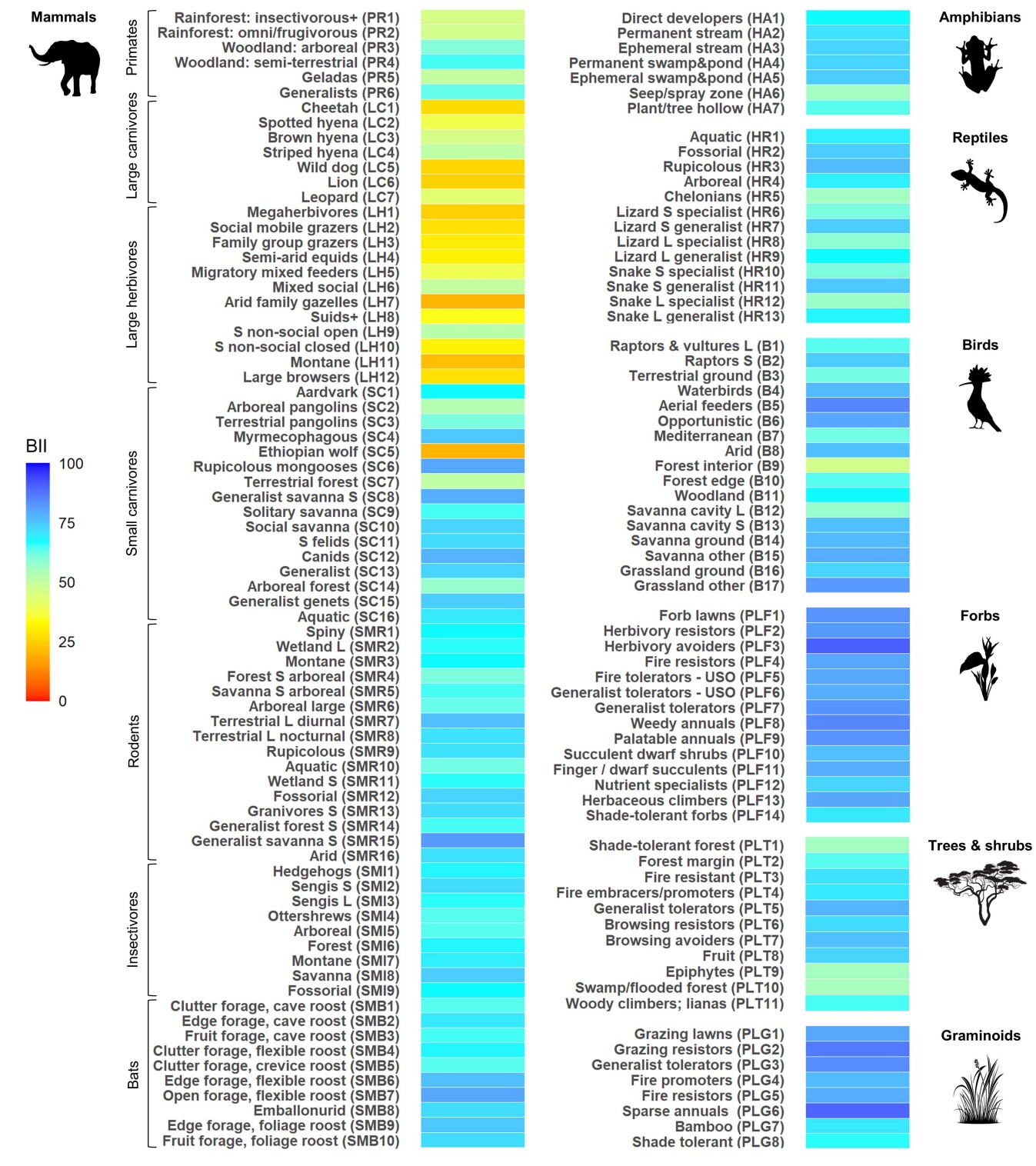

**Extended Data Fig. 2 | The Biodiversity Intactness Index (BII) for different functional groups of vertebrates and plants.** Average BII scores across >300,000 pixels spanning sub-Saharan Africa are shown as a heatmap. Group codes correspond to distinct functional response groups of species in the published bii4africa dataset[5] that includes detailed descriptions of each group. S–small, L–large. Silhouettes obtained from PhyloPic (https://www.phylopic.org).

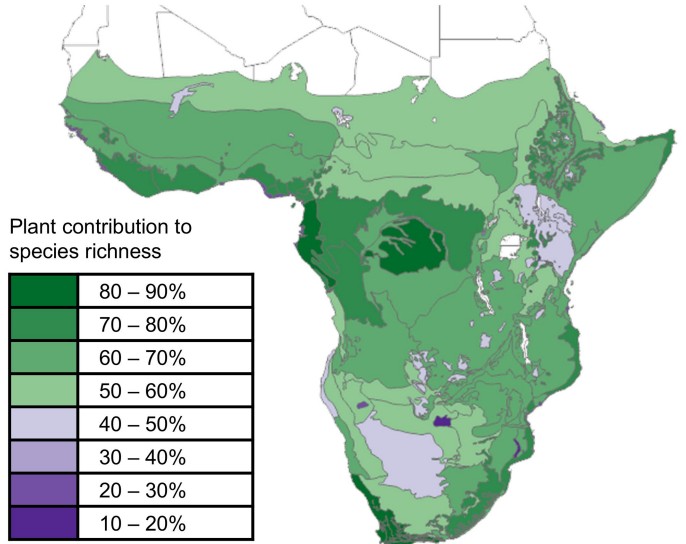

Plant contribution to
species richness

| | |
|---|---|
| ■ | 80 – 90% |
| ■ | 70 – 80% |
| ■ | 60 – 70% |
| ■ | 50 – 60% |
| ■ | 40 – 50% |
| ■ | 30 – 40% |
| ■ | 20 – 30% |
| ■ | 10 – 20% |

**Extended Data Fig. 3 | Percentage plant contribution (as opposed to vertebrates) to total species richness per ecoregion.** As each species is weighted equally in the Biodiversity Intactness Index, relative species richness influences which species groups contribute more to the index in a given ecoregion. Maps were created using ArcGIS Pro v.2.7.0.

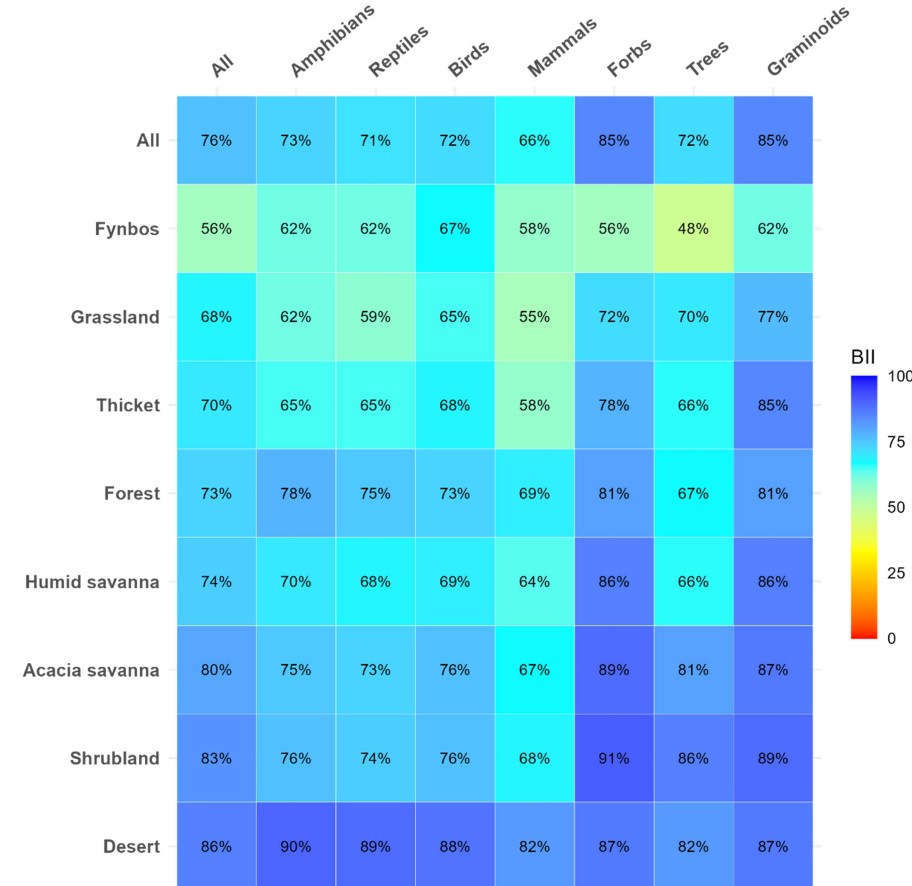

**Extended Data Fig. 4 | The average Biodiversity Intactness Index (BII) for the major species groups across sub-Saharan Africa's eight biomes.** Individual biomes are ranked based on increasing overall BII (All−all biomes collectively).

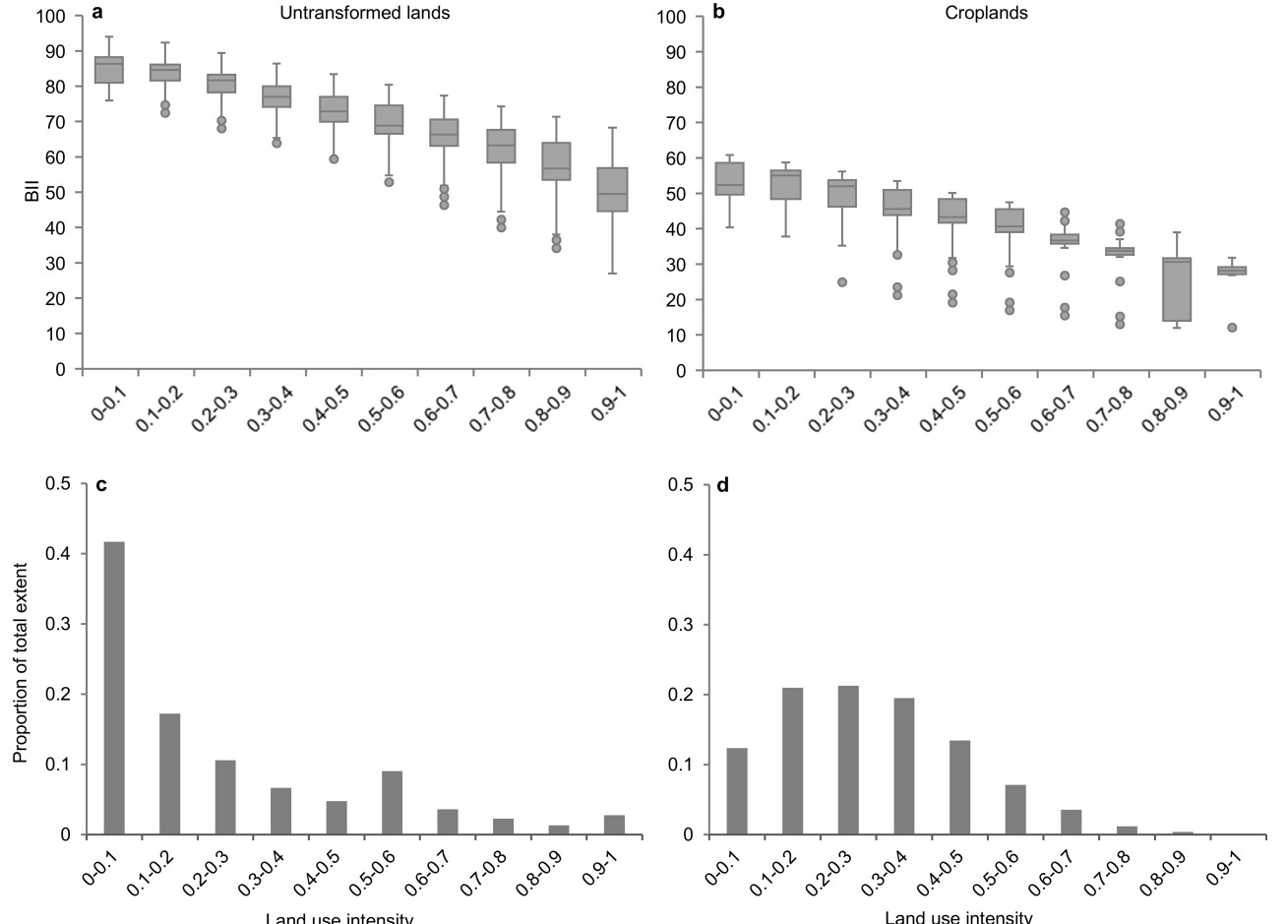

**Extended Data Fig. 5 | The intensity of land use across sub-Saharan Africa and its influence on the Biodiversity Intactness Index (BII).** The two most extensive land uses, unprotected untransformed lands and croplands, are shown. Boxplots show the distribution of absolute BII values for pixels of different land use intensities within (a) unprotected untransformed lands (n = 252,087 pixels) and (b) croplands (n = 43,137 pixels). Boxplots show median BII scores across pixels, interquartile ranges (IQR), maximums and minimums within 1.5*IQR and outliers. Histograms show the relative contribution of pixels of different land use intensities to total extent within (c) unprotected untransformed lands and (d) croplands. The land use intensity of each pixel is categorised on a scale from 0–0.1 (lowest intensity) to 0.9–1 (highest intensity).

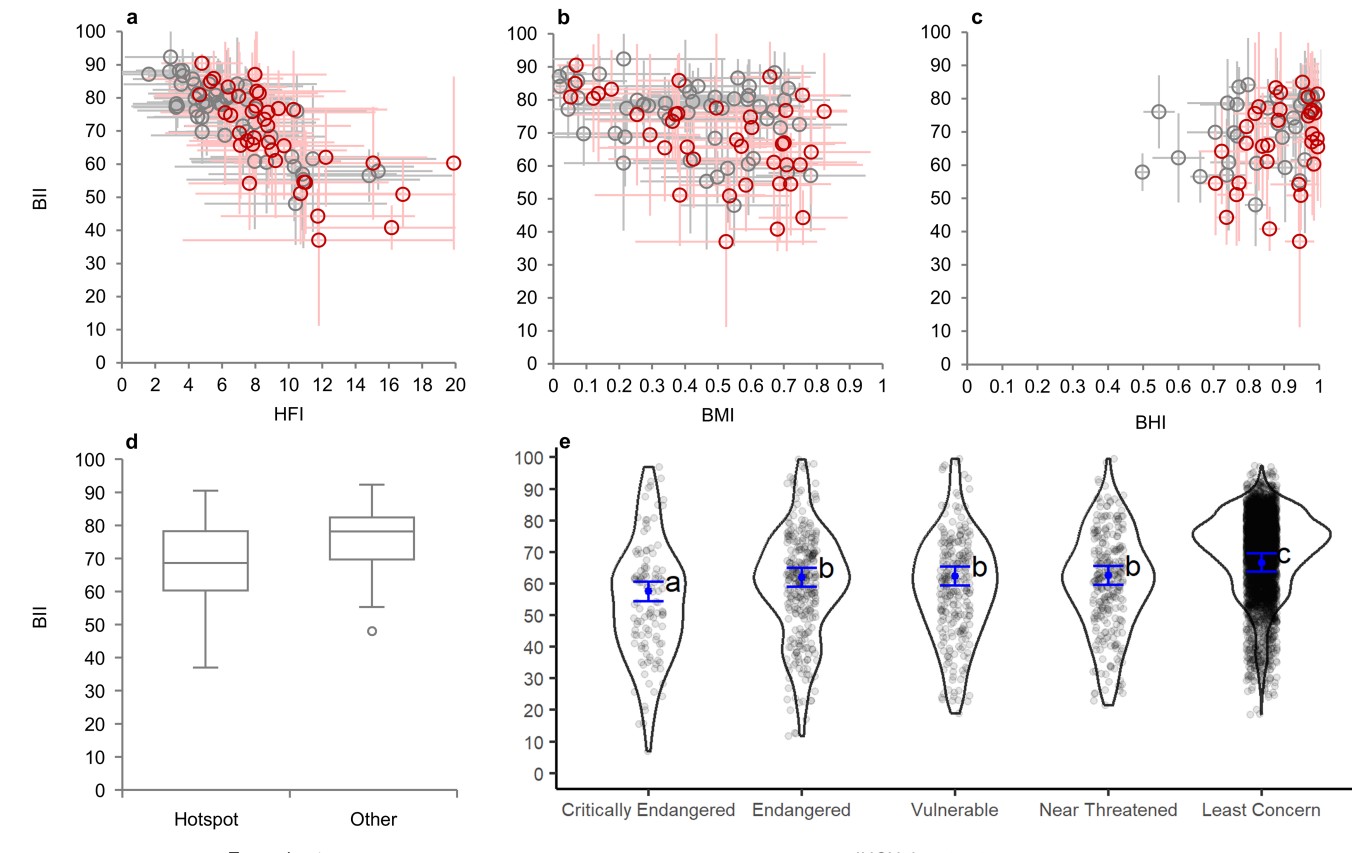

**Extended Data Fig. 6 | Significant relationships between the Biodiversity Intactness Index (BII) and other human pressure indicators.** These indicators include the (a) Human Footprint Index (HFI) (Spearman's $\rho = -0.773$), (b) Biomass Modification Index (BMI) ($\rho = -0.414$), (c) Biodiversity Habitat Index (BHI) ($\rho = 0.578$), (d) biodiversity hotspots (Mann-Whitney $U = 656$, $n_1 = 51$, $n_2 = 38$, $p = 0.010$) and (e) International Union for the Conservation of Nature (IUCN) Red List threat categories (ANOVA $F = 41.455$, $df = 4$, $p < 0.001$). The scatterplot points (a–c) show average indicator values for each of sub-Saharan Africa's 89 ecoregions, with lines reflecting standard deviations around these averages, and ecoregions with ≥50% of their area inside versus outside a biodiversity hotspot shown by red versus grey points and lines, respectively. The boxplot (d) shows the spread of BII values for hotspot and other ecoregions. The violin plot (e) shows the spread of BII values (grey points) for 4,887 vertebrate species within different IUCN threat categories, with predicted BII means and standard errors for each category shown in blue and different letters indicating significant differences in these means, accounting for the nested nature of the data and controlling for species range size (CR–EN Tukey $t = -4.164$, $p < 0.001$; CR–VU $t = -4.229$, $p < 0.001$; CR–NT $t = -4.599$, $p < 0.001$; CR–LC $t = -9.384$, $p < 0.001$; EN–VU $t = -0.331$, $p = 0.997$; EN–NT $t = -0.782$, $p = 0.936$; EN–LC $t = -7.375$, $p < 0.001$; VU–NT $t = -0.425$, $p = 0.993$; VU–LC $t = -6.37$, $p < 0.001$; NT–LC $t = -6.204$, $p < 0.001$; CR–Critically Endangered, EN–Endangered, VU–Vulnerable, NT–Near Threatened, LC–Least Concern).

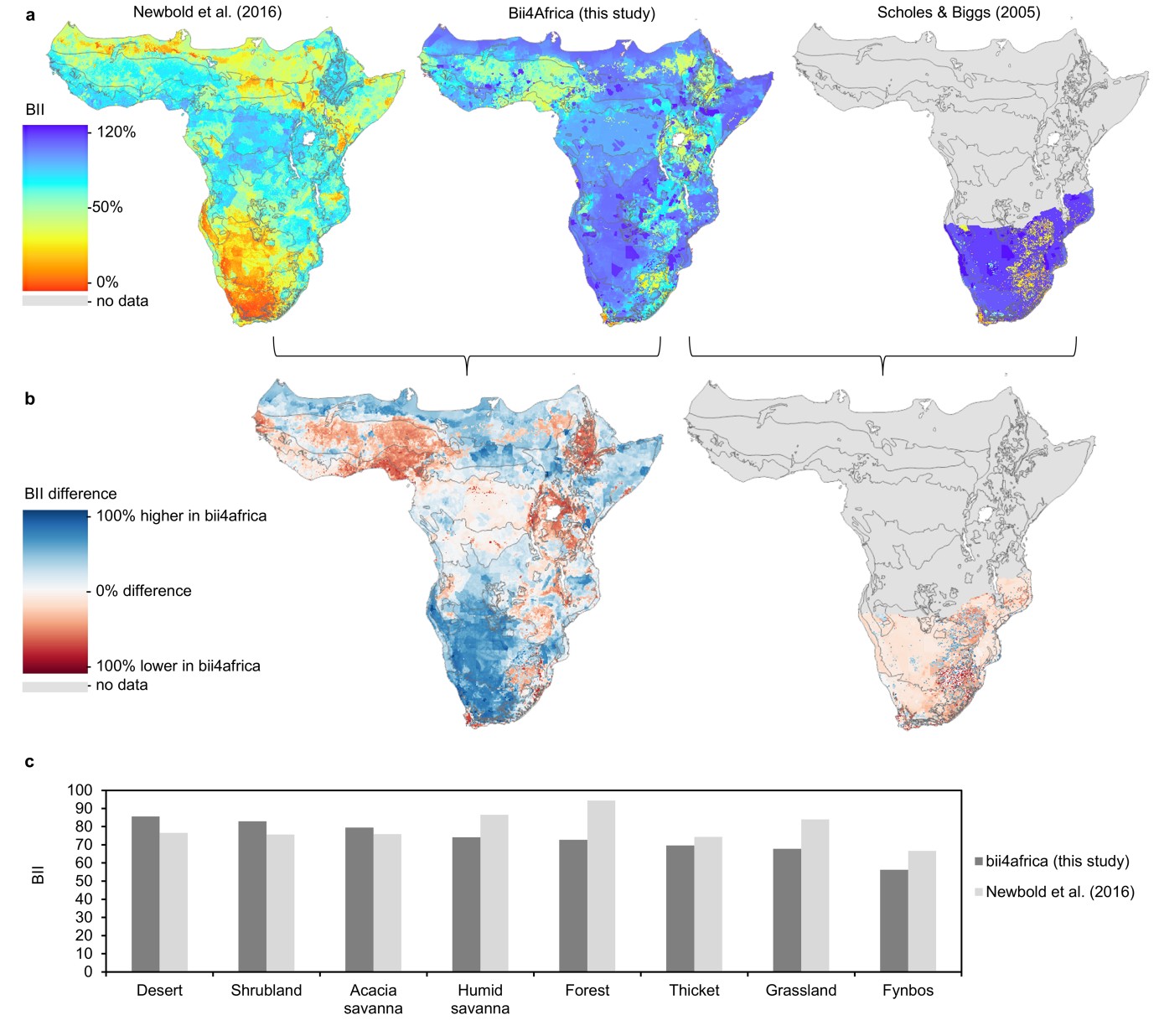

**Extended Data Fig. 7 | Comparison with previous assessments of the Biodiversity Intactness Index (BII).** (a) This study's BII across sub-Saharan Africa (with biomes delineated), compared with that predicted for the region by a global model in 2016 (Newbold et al.[13]), and for southern Africa by an expert elicitation in 2005 (Scholes and Biggs[4]); (b) the differences between this study's BII and the two earlier maps; and (c) the average BII per biome according to this study and Newbold et al.[13]. Maps **a** and **b** were created using ArcGIS Pro v.2.7.0.

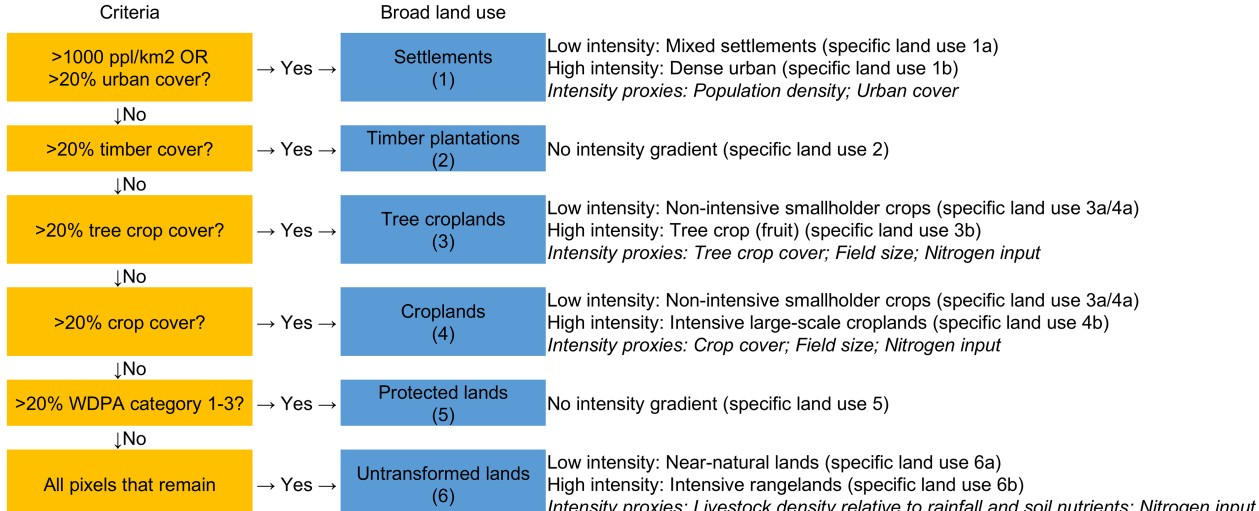

**Extended Data Fig. 8 | Land use allocation algorithm and relevant intensity gradients and proxies.** Specific land use labels correspond with the nine land uses for which intactness scores were estimated by experts[5] (Supplementary Table 2). Note that specific land use 3a/4a represents the low-intensity version of broad land use classes 3 and 4, given that non-intensive smallholder croplands typically include cash crops together with tree crops. A single intensity of land use was used for timber plantations, which are not common in the region and typically intensive in nature, and protected lands, for which intensity is inappropriate (effectiveness is more appropriate but challenging to quantify on a continental scale). (ppl/km$^2$—people per square kilometre, WDPA—World Database on Protected Areas).

**Extended Data Table 1 | Levels of uncertainty in the Biodiversity Intactness Index (BII) estimates across the predominant land uses and species groups**

| Land use | All | Plants | Vertebrates |
|---|---|---|---|
| All | ±14% | ±17% | ±9% |
| Urban | ±15% | ±18% | ±9% |
| Timber | ±17% | ±20% | ±11% |
| Tree crops | ±18% | ±21% | ±11% |
| Crops | ±16% | ±20% | ±11% |
| Untransformed | ±14% | ±17% | ±9% |
| Protected | ±8% | ±10% | ±5% |

Uncertainty around the BII score for each of > 300,000 pixels spanning sub-Saharan Africa reflects the 95% confidence interval around average expert estimates of intactness in the bii4africa dataset[5], with the lower and upper confidence limits averaged across pixels in each land use based on the lower-limit-BII and upper-limit-BII values, respectively.

**Extended Data Table 2 | Datasets used to produce a land use and intensity map refs. 86–95**

| Variable | Source | Year | Resolution |
|---|---|---|---|
| Urban cover (%) | Marconcini et al.[86] | 2019 | 10 m |
| Population density (No./km$^2$) | CIESIN[87] | 2020 | 1000 m |
| Timber plantation cover (%) | Du et al.[88] | 2015 | 30 m |
| Tree crop cover (%) | Du et al.[88] | 2015 | 30 m |
| Crop cover (%) | Potapov et al.[89] | 2020 | 30 m |
| Field size (category) | Lesiv et al.[90] | 2017 | N/A – vector |
| Nitrogen input (kg/ha) | Houlton et al.[91] | 2015 | 15 arc min |
| Cattle density (No./km$^2$) | Gilbert et al.[92] | 2010 | 5 arc min |
| Sheep density (No./km$^2$) | Gilbert et al.[92] | 2010 | 5 arc min |
| Goat density (No./km$^2$) | Gilbert et al.[92] | 2010 | 5 arc min |
| Rainfall (average mm/year) | Funk et al.[93] | 1991–2021 | 5566 m |
| Soil type (category) | Bell[94] | 1982 | N/A – vector |
| Protected area cover (%) | IUCN & UNEP-WCMC[95] | 2021 | N/A – vector |

# Reporting Summary

## Statistics

For all statistical analyses, confirm that the following items are present in the figure legend, table legend, main text, or Methods section.

| n/a | Confirmed | |
|-----|-----------|---|
| ☐ | ☒ | The exact sample size (*n*) for each experimental group/condition, given as a discrete number and unit of measurement |
| ☒ | ☐ | A statement on whether measurements were taken from distinct samples or whether the same sample was measured repeatedly |
| ☐ | ☒ | The statistical test(s) used AND whether they are one- or two-sided *Only common tests should be described solely by name; describe more complex techniques in the Methods section.* |
| ☐ | ☒ | A description of all covariates tested |
| ☐ | ☒ | A description of any assumptions or corrections, such as tests of normality and adjustment for multiple comparisons |
| ☐ | ☒ | A full description of the statistical parameters including central tendency (e.g. means) or other basic estimates (e.g. regression coefficient) AND variation (e.g. standard deviation) or associated estimates of uncertainty (e.g. confidence intervals) |
| ☐ | ☒ | For null hypothesis testing, the test statistic (e.g. *F*, *t*, *r*) with confidence intervals, effect sizes, degrees of freedom and *P* value noted *Give P values as exact values whenever suitable.* |
| ☒ | ☐ | For Bayesian analysis, information on the choice of priors and Markov chain Monte Carlo settings |
| ☒ | ☐ | For hierarchical and complex designs, identification of the appropriate level for tests and full reporting of outcomes |
| ☐ | ☒ | Estimates of effect sizes (e.g. Cohen's *d*, Pearson's *r*), indicating how they were calculated |

*Our web collection on statistics for biologists contains articles on many of the points above.*

## Software and code

Policy information about availability of computer code

| Data collection | No original data was collected for this study |
|-----------------|-----------------------------------------------|
| Data analysis | The land use maps were developed using custom code in Google Earth Engine. The BII maps were developed using custom code in R. All code is available, as outlined in the data availability section. |

For manuscripts utilizing custom algorithms or software that are central to the research but not yet described in published literature, software must be made available to editors and reviewers. We strongly encourage code deposition in a community repository (e.g. GitHub). See the Nature Portfolio guidelines for submitting code & software for further information.

## Data

Policy information about availability of data

All manuscripts must include a data availability statement. This statement should provide the following information, where applicable:
- Accession codes, unique identifiers, or web links for publicly available datasets
- A description of any restrictions on data availability
- For clinical datasets or third party data, please ensure that the statement adheres to our policy

The expert-elicited bii4africa dataset used in this study is available on Figshare (https://doi.org/10.6084/m9.figshare.c.6710463.v1)5. Input data on species range maps and threat categories are available through the IUCN Red List (https://www.iucnredlist.org/)63 and Birdlife International (http://datazone.birdlife.org/species/requestdis.). Input data on ecoregions are available through Ecoregions2017© Resolve53 (https://ecoregions.appspot.com/). Input data on plant forms in the

## Research involving human participants, their data, or biological material

Policy information about studies with human participants or human data. See also policy information about sex, gender (identity/presentation), and sexual orientation and race, ethnicity and racism.

| | |
|---|---|
| Reporting on sex and gender | n/a |
| Reporting on race, ethnicity, or other socially relevant groupings | n/a |
| Population characteristics | n/a |
| Recruitment | n/a Note that this paper makes use of a published dataset co-produced by 200 biodiversity experts, produced as part of the broader project, with recruitment detailed in that publication (Clements, H. S. et al. The bii4africa dataset of faunal and floral population intactness estimates across Africa's major land uses. Scientific Data, https://www.nature.com/articles/s41597-023-02832-6) |
| Ethics oversight | Ethical clearance for the project was provided by Stellenbosch University (project number 15182). |

Note that full information on the approval of the study protocol must also be provided in the manuscript.

# Field-specific reporting

Please select the one below that is the best fit for your research. If you are not sure, read the appropriate sections before making your selection.

☐ Life sciences          ☐ Behavioural & social sciences          ☒ Ecological, evolutionary & environmental sciences

For a reference copy of the document with all sections, see nature.com/documents/nr-reporting-summary-flat.pdf

# Ecological, evolutionary & environmental sciences study design

All studies must disclose on these points even when the disclosure is negative.

| | |
|---|---|
| Study description | We quantified the biodiversity intactness index (BII) across sub-Saharan Africa. We make use of a published bii4africa dataset that contains experts' standardised estimates of the impact of sub-Saharan Africa's predominant land uses on diverse functional groupings of species that represent ~50,000 terrestrial vertebrates and vascular plants. In this paper, we integrate ten spatial datasets to map these land uses, which we combine with bioregional lists of indigenous taxa and the associated bii4africa expert data to map the BII across sub-Saharan Africa. |
| Research sample | We include all sub-Saharan African terrestrial vertebrate species and plant groups. Species richness per ecoregion was drawn from existing datasets including the IUCN redlist (vertebrates) and Sosef et al 2017 and Kier et al 2005 (plants). Intactness scores for these species were drawn from the published bii4africa dataset (Clements et al 2024). |
| Sampling strategy | No sampling was done - each pixel across sub-Saharan Africa was included in analyses. Pixel size was selected to match the scale at which experts estimated intactness scores, as described in the paper. |
| Data collection | No original data were collected. A published dataset, produced as part of the broader project, was used in this study (https://www.nature.com/articles/s41597-023-02832-6) with data collection methods described in detail in that paper, and summarised in the current paper. |
| Timing and spatial scale | No original data were collected in this study. The published expert dataset that we made us of was produced during expert elicitations that ran from the end of 2020 to the beginning of 2022 (https://www.nature.com/articles/s41597-023-02832-6). The published land use map layers that we used were produced between 1982 (soil map) and 2021 (protected area map), as outlined in Extended Data Table 2. Species range maps were accessed from the IUCN redlist and Birdlife International in 2022 for vertebrates, while plant richness information was drawn from Kier et al 2005 and Sosef et al 2017. Mapping extent was sub-Saharan Africa, including all mainland Afrotropical ecoregions. |
| Data exclusions | No data were excluded |
| Reproducibility | n/a - study did not follow an experimental design |
| Randomization | n/a - no samples were taken |
| Blinding | Expert data were anonymised during aggregation and analysis, as described in the published dataset that we made use of (https://www.nature.com/articles/s41597-023-02832-6) |

Did the study involve field work?  ☐ Yes  ☒ No

# Reporting for specific materials, systems and methods

We require information from authors about some types of materials, experimental systems and methods used in many studies. Here, indicate whether each material, system or method listed is relevant to your study. If you are not sure if a list item applies to your research, read the appropriate section before selecting a response.

## Materials & experimental systems

| n/a | Involved in the study |
|-----|----------------------|
| ☒ ☐ | Antibodies |
| ☒ ☐ | Eukaryotic cell lines |
| ☒ ☐ | Palaeontology and archaeology |
| ☒ ☐ | Animals and other organisms |
| ☒ ☐ | Clinical data |
| ☒ ☐ | Dual use research of concern |
| ☒ ☐ | Plants |

## Methods

| n/a | Involved in the study |
|-----|----------------------|
| ☒ ☐ | ChIP-seq |
| ☒ ☐ | Flow cytometry |
| ☒ ☐ | MRI-based neuroimaging |

## Plants

| | |
|---|---|
| Seed stocks | n/a |
| Novel plant genotypes | n/a |
| Authentication | n/a |

