## [Peer Review File · Nature]

A place-based assessment of Sub-Saharan Africa's biodiversity intactness

Corresponding Author: Dr Hayley Clements

Version 0:

Reviewer comments:

Referee #1

(Remarks to the Author)

This is a really interesting paper describing a first proper attempt to estimate the biodiversity intactness across the African continent. Recognising the challenge involved in direct measurement of this the authors opt for an expert-based assessment, identifying a cohort of over 200 taxonomic experts to help in the process. They find that relatively small proportions of current biodiversity are across the continent fall in protected sites, with much the larger part remaining in unprotected but untransformed landscapes, co-occurring alongside human activities. These results are important and broadly in line with what I would have expected - I'd love to have seen some comparisons with other continents to put this into context, but as a top level result this seems plausible.

I have to admit I was excited about reading this paper - Local and Indigenous Knowledge is a hot topic in current biodiversity discussions. IPBES has been praised for delivering a better job on this than many previous attempts, and the K-M GBF identifies improving its involvement as a key target. So I was slightly disappointed that despite the title and the talk of co-developing knowledge, which made me think there was going to be a new method of harnessing indigenous knowledge, this isn't really the case at all. Rather, the underlying data are the result of a fairly standard Delphi process involving experts from across the world (though in reality a disappointingly small fraction of experts are not from N. America, Europe or South Africa). I struggle to really see this as 'local knowledge' when, for example, only two of the ornithologists are West African (both Nigerian), despite the existence of an active ornithological community in the region. To suggest that Southern African ecologists are more local to East or West Africa than geographically much closer European ecologists is not aligned with the CBD or IPBES use of the term 'local knowledge'. Nor do I see any sense of 'co-production' here in the usual usage of this term - I suggest the authors considerably revise their use of local and co-production.

Despite this slight disappointment caused by what might be considered some of the hype in the writing, the paper stands or falls on the basis of the work these 200+ experts undertook. For them to even attempt such a comprehensive review is highly commendable. The difficulties of this are clearly acknowledged, and have resulted in grouping of species and geographies in order to make the problem tractable. It is therefore crucial that we trust this data reduction. If I have understood the methods correctly, each expert identified their particular expertise and were then asked to estimate an average score for the change in population of functional groups within their taxonomic speciality by land use and biome. Thus an ornithologist would be asked to provide estimates for each of 17 groups of birds in each of 8 biomes, within which there may be up to 9 land uses. This would be (up to) 1224 total estimates for an ornithologist. (NB this summary wasn't easy to piece together involving both the paper for review and a cited paper and searching around the project website too and might be wrong, it would be great to see an example like this clearly laid out somewhere.) Hidden in here is the idea that an individual ornithologist from Nigeria, say, could reasonably estimate the average % decline in aerial feeders (swallows and swifts) in all the savanna areas of Africa across the 9 land use tasks. I am deeply skeptical that any expert could do this. While it might be possible within one's own country where one's local knowledge is valuable, to estimate it across all the Acacia savannas in the continent as a whole seems implausible. Moreover, the reader is left to trust that the groupings are reasonable (I couldn't really work out what they were fully) and that the implicit assumption they respond similarly within a biome to land use change is reasonable. I would like to see this tested at least at some level. Perhaps a quick evaluation of the current trends for those species included in the IUCN Red List would be useful - if we are to trust these data I would like to see

agreement between the average red list category for each functional group and the expert-based estimates of this, and I'd like to see that the IUCN status of species within each taxonomic grouping is more similar than between functional groups.

A similar test of internal consistency in assessments by global vs local experts would also be reassuring. For example, I would love to have seen some evaluation of the differences in assessment by more or less local contributors. At least for the more geographically restricted biomes it would be interesting to see if (truly) local experts had different average assessments of change than experts from elsewhere in the continent, vs those based elsewhere in the world. Similarly, it might be interesting to check whether the large national differences in estimated BII are reflected in the average scores of experts from those countries - do, for example, experts from the relatively intact Mozambique on average estimate smaller declines than those from the much more heavily impacted regions of Nigeria or Rwanda? For the method to work well this needs NOT to be the case. These sorts of analysis could not only help us identify the real value of true local knowledge, and reassure us that the expert group did have the detailed knowledge required to make these very large-scale assessments.

Without total confidence in the underlying evaluation, the rest of the analyses are harder to review. I do think they are a sensible way of cutting the data up for analysis, and (provided the underlying data are trustworthy) the justifications and conclusions are robust with most analyses adequately explained and apparently sensibly undertaken. The only real question I raised here concerns the figures in Extended Data Fig 2, where I struggled to understand why there are such obvious lines in the raw data - presumably reflecting groups of one land use, or biome, or taxonomic group? This could do with explanation and these figures could be tidier in order to allow us to interpret and know whether this artifact is a problem or not.

(Remarks on code availability)

Will happily do it if they come back with sensible further analyses convincing me the data are worth looking at...

Referee #2

(Remarks to the Author)

General Comments:

Clements et al. really did tremendous work on "The biodiversity intactness of Africa assessed through local knowledge" with about 200 experts to address the current knowledge gap on less-known and rarely well-represented sub-Saharan biodiversity in global biodiversity maps and data. Hence, the study is timely, crucial for biodiversity conservation in sub-Saharan Africa, and worth publishing in this journal. The data presented, which was used in estimating the biodiversity intactness index, are original and were collected through a clear and reproducible expert elicitation method. Further, the methods, estimates, and statistics in the main manuscript, supplementary, R scripts and the previously published bii4africa dataset paper are very clear enough and reproducible.

However, there are some concerns that should be addressed. First, given the disproportionately low representation of experts that are resident in nearly all the sub-Saharan Africa countries (except South Africa, as shown by the affiliations of the experts/authors on the published bii4africa dataset paper and website), I have concerns that the non-resident experts may not really know much about the history or current intactness of biodiversity in each country, ecoregion, or species group they represent during BII scoring, thereby weakening the "bottom-up" approach. For instance, an expert that had a long-term study or lived in a country over several years (like 10 years or more) but has long left the country or ecoregion or stopped working there for say 10 years or more before this data collection may overestimate or underestimate the BII. Hence, authors should add texts on how they, to a large extent, accurately identify the country/region of experience of every expert to avoid underestimating or overestimating BIIs of most countries (other than South Africa that had several people). Or authors, should add caveats to the manuscript about the challenges faced or potential weaknesses of the expert recruitment method despite their efforts to mitigate such challenges, as they did in Lines 403-406 and 635-636 for limited access to data from protected areas.

Second, I do strongly believe that the title can only explicitly include the "local knowledge" if the intactness scoring truly involved the "local non-expert residents or citizens" of sub-Saharan Africa that live and depend on the ecosystems that house these species. The traditional ecological knowledge of locals are often passed from generation to generation and it would have added more values to the study. So, given that "non-expert locals" were missing in the intactness scoring (unless I missed it), I suggest that the title be revised to: "The biodiversity intactness of Africa assessed through experts' local knowledge"

Generally, all aspects of the manuscript are well written except for the necessary edits, some rearrangements (and suggested analyses) in the result section, and, more importantly, the improvement of the discussion section that I mention in the specific comments below. I therefore recommend the manuscript for publication after the revisions I suggested here and below.

Specific comments:

Line 55: rewrite as "Grasslands and fynbos (mediterranean-type ecosystems) suffered the greatest declines, mostly from land transformation into cropland, while deserts and shrublands have been least impacted."

Line 61: "decision-makers"

Line 62: Two heavy words there. Replace "credible, legitimate" with "real"

Line 70: replace "it" with "biodiversity"

Line 72: delete "to"

Line 85: Before the IPBES sentence, please add one more sentence here, probably as an example, to shed more light on the sentence in Line 82-85.

Line 89: "We apply this approach to assess the biodiversity intactness of ecosystems in sub-Saharan Africa, ..."

Line 107: how about replacing "sub-Saharan Africa's" with "African"?

Line 109: how about replacing “Africa’s” with “African”?

Line 109: be specific with the “remaining vegetation”, please. Were you referring to the “vegetation in other continents” or “other vegetation types in Africa”?

Line 121-122: “the impact of sub-Saharan Africa’s predominant land uses and their intensities on ~50,000 species of terrestrial vertebrates and vascular plants.”

Line 124: “experts’ estimates...”

Line 125, 151 and other similar places like the text in Fig. 1, title of Fig. 3: Wherever you refer to regions within sub-Saharan Africa, which you already regarded as a region from the abstract, please consistently write “ecoregions(s)” or “biome(s)” accordingly not the ambiguous “region(s)”. For example, in Line 151, replace “vegetation types or regions” with “biomes or ecoregions”. Doing this align all texts and sections of the paper with the terms in Figures (both main and extended) and in results.

Line 129: delete “that”

Line 141: is there a mistake in “magnitude more experts”?

Line 161-163: This is true, as Africans engage in hunting and consume those animals as “bush meat”. It would be good that you mention this in your discussion, and what can be done (as replacement though) to stop this hunting of non-domesticated animals in the “wild” for food.

Figure 1: A very cool and interesting presentation here. Please edit the “Low-High” under the “Intensity” in the land use class legend to “Low  High” in Figure 1a: i.e., use arrow instead of hyphen, and let the “Low” and “High” be at the extreme left and right part of the box, respectively.

In the third to the last line on the first page of “Scientific Data File”: “unavailability” should replace “availability”

Line 195: I think the bii4africa dataset should be available as soon as the paper is accepted (by this journal or elsewhere) for reproducibility and further studies for interested.

Line 198: add “(Fig. 2a)” just after the word “intactness”.

Line 198: since the paragraph in Line 211-222 show the relative impacts of the predominant land uses in sub-Saharan Africa, please add either “absolute”, “independent” or “individual” in between “Considering the ...” and “... impact of the region’s predominant land uses on biodiversity intactness, we” in Line 198.

Line 199: do you really need to add “strictly” here?

Line 203-204: Consider replacing this sentence (i.e., “Depending on species composition and land use intensity, the BII of a given land use pixel varies (Fig. 2a)”) with the following more direct sentence: “Our estimates also show that the BII of each of the major species groups varies between the six land use classes (Fig. 2b).”

Line 207: with the Fig.2b results/boxplots, the “while the opposite holds true for reptiles” appears not really a true opposite of the first part of the sentence. Please check it and reframe for clarity.

Line 212: “its total extent”. Please use exact word to replace “its”. It makes the sentence ambiguous.

Line 211-222: While it is very nice that the BII results, which are the main focus of this study, occupies the 1st paragraph of the results section, Fig. 1a only shows land use intensity without the BII numbers quoted in Lines 214, 217 and 219 and it is not really related to the story in Fig. 1b&c and paragraphs 1&2 of the results section. Hence, for a better presentation and easy-to-follow flow of the results, please detach Fig. 1a from Fig. 1 and make it Fig. 2a; create a new Fig. (a box plot like the current Fig. 2a as Fig. 2b) to show the “relative” proportion of the remaining BII value (76%) that each of the 6 classes contribute. The new fig. 2b will further explain the new Fig. 2a (former 1a). With that adjustment, the current Fig. 2a becomes 2c, current 2b becomes 2d. And more importantly, please let the whole paragraph of Line 211-222 come before that of 198-209, which would show that you move from general to specific estimates.

Line 231: Move this sub-topic (Land use intensity impacts intactness) to the line preceding the contents in Line 211-222.

Line 233-234: please write “(Fig. 2a)” (i.e., the current Fig. 1a which I suggested above that you should make 2a) after the phrase “Considering the two most extensive land uses (croplands and unprotected untransformed lands)”.

Line 235: please add “two” before “land uses”

Line 236-239: This isn’t clear- something is missing in Line 238. Seems you wanted to add “compared with 54% BII in” before the word “smallholder”? If choose my guess, remember to remove the (54%) at the of the sentence.

Line 236-244: Please consider adding extended figure(s) on the percentages quoted here. Also, consider reporting the “relative” percentages (not the absolute ones) such that the BII comparison between the land use intensities within each of the two land uses adds up to 100, as the title of extended figure 2 and histogram imply.

Line 253: “Differences in intactness between countries, biomes, and ecoregions”

Line 254-257: And please make a sentence about those remaining 15 countries with 70-80% BII too. Given that those with less than <70% BII are relatively small in number, consider dwelling more on the likely drivers of such critically low BII score in the discussion section (and possibly with some supplementary data analysis of countries like Nigeria and Rwanda that would nail such huge biodiversity loss and trigger government and NGO’s attention).

Line 260-261: please rewrite as “Biodiversity intactness across (a) countries, (b) biomes, and (c) ecoregions³⁰ of sub-Saharan Africa. Average BII scores are shown in each map.”

Given that the ecoregions are many and you couldn’t insert them in Fig. 3c, I suggest you delete Fig. 3c from the manuscript, and make the “Extended Data Fig. 4” to be the main Fig. 3c.

Figure 3b: is there any biome particularly called “Mosaic”?

Line 266: “remaining BII are made by each biome in unprotected untransformed lands: ...”

Line 268-269: “The largest contributions of desert and fynbos to the remaining BII are in protected lands” Not so sure if my suggested revisions in Line 266, 268-269 are so clear, but I just want the authors to use direct, unambiguous sentences to convey Figure 4 as a follow-up to 3b in Line 265-269.

Line 278-279: “likely because crop production is generally unviable in these biomes (Fig. 4).”- so lack of cultivation could drive such poor BII in deserts and shrublands? I think that is a very unlikely speculation, unless you have evidences.

Line 280: list the grassy biomes in parentheses just after the word grassy biomes.

Line 283-284: this “, together with deforestation for rangelands, croplands and timber plantations” does not complement to the preceding clause in the sentence.

Line 285: list the five biomes in parentheses for clarity
Line 294: see the comment on this Fig. 3c above
Line 310: delete "species"
Line 317: "make use of here" "use here"
Line 319-320: Not so clear.
Line 333: do you really meant to write "average mean"?
Line 349: Consider adding comma sign before and after the "for example"
Line 352-355: This information appears not related with the preceding sentences and the story on "The validity of this biodiversity assessment" and Extended Fig. 6a-c. Maybe you need to create a premise for it or relocate it to a more pleasing spot in this paragraph.
Line 367: why don't you just say "200 experts" instead of "several hundred"?
Line 373: it would be nice that you briefly recap here that the remaining overall BII of sub-Saharan Africa is 76%.
Line 375: Based on my comment on Fig. 1 and on Line 211-222, the referenced "(Fig. 1) may not be Fig. 1 (alone or again).
Line 375-377: Absolutely! That's very important.
Discussion section: Despite that the results section was full of exhaustive and interrelated data analyses and that the discussion section should mirror the structure of the results section in flow for ease of understanding and linkage of the findings, it is strange that about 80% or more of the discussion section (especially from 412-439) repetitively (in multiple, different sentences) only summarily speculate (without data comparisons between the current findings and the previous) the potential benefits of the approaches and data used for this study. Hence, a vast portion of the intriguing findings in the study was left undiscussed — especially in relation with existing studies' reported BII values for sub-Saharan Africa; the potential (literature-supported) drivers of the BII values reported for across and within biomes, ecoregions, and countries in sub-Saharan Africa; whether there were over estimation or underestimation, and why this study potentially provide a more accurate BIIs etc. For example, authors should discuss the underlying drivers of the huge (\geq ~50%) decline in BII in West Africa and South-eastern Africa countries.
Line 410: "megfaunal" -> "megafaunal".
Line 465-466: "in such cases" or "in all cases"? I looked through the methods but couldn't find if any other BII formular apart from Equation 1 exist for cases without data scarcity. Reading through 499-505, I later found that the "in such cases" only apply to other than large mammals species. Please show and illustrate the equation for the "ideal situation" first (i.e., the situation you expressed in Line 502-505) before the "data scarcity situation", which the current equation 1 displays. If I am wrong, please make the texts here clearer for readers like me.
Line 465,469: "region (commonly, the ecoregion)"- please directly and simply say "ecoregion" whenever you meant "ecoregion" and limit the use of "region" to "sub-Saharan Africa" throughout the manuscript.
Line 512: how finer can a patch or scale would be for it to be regarded as "finer"? Please state the specific spatial scale that was regarded as being finer.
Line 525: simply replace "biogeographical vegetation types" with "biomes" as you did name them in next sentence and in Fig. 3b.
Line 544: replace "regions" with "ecoregions and biomes"
Line 603: Please cite the "Extended Data Table 2" after the nitrogen input inside the parentheses.
Line 617, 618, 619: "herbivore biomass capacity" and "herbivore biomass" are not that correct to be used as terminologies. You meant "herbaceous biomass", right? If yes, in Line 617, change "herbivore biomass capacity" to "herbaceous biomass (i.e., forage for herbivores)"; in Line 618, change "sustainable herbivore biomass" to "biomass declines", and in Line 619 just say "biomass". Replace ". In addition" with "and" in Line 618 to merge the 2 sentences.
Line 619: Please add "Hence," before "we clustered..."
Line 637: The latter were considered
Line 642: use multiplication sign for dimensions not x
Line 650: "sub-Saharan Africa region"
Line 649-650: Think this
Line 656: replace # with No. throughout the table
Line 681: change x to multiplication sign
Line 700-710: Not sure this is different from the preceding sentence. Moreso, "reasonable correlation" is subjective.
Line 703: Extended Data Fig. 6. shows "across ecoregions" correlations not "per ecoregion"
Line 826: But there are references mentioned in the Methods with serial numbers before this 51st reference.

(Remarks on code availability)

Yes, the code provided via the <https://figshare.com/s/00168eb685728c0be9d3>, guided by a self-explanatory code lines and README run well.

Referee #3

(Remarks to the Author)

Thank you for four submitted article, which involves providing biodiversity indices for policy in contexts of data paucity. Specifically, the use of expert input to evaluate sub-Saharan Africa's biodiversity intactness. Since my expertise is in expert elicitation and sustainable development, I will focus my review of those aspects (methods and claims made). This also corresponds with what the editor asked me to focus on.

First, I should point out that my initial submitted review did not account for the fact that most of the expert elicitation is presented in an accompanying article in Scientific Data. I am not used to having methods described in accompanying articles. Personally I lament that some of the most widely read journals tend to de-emphasize the presentation of methods (such as at the end of articles), but regardless I would recommend that the authors more clearly identify that the elicitation

methods can be found in the accompanying article. When I read this piece it read to me as if the authors point to the other article as a different study using a similar methodology. In fact, I believe that since expert elicitation is so prominent in the presentation of this study, the authors should provide much more details of the methods in the article itself (or at least in the supplement).

I think the topic is a very important and difficult effort, especially to do well. However, I have some concerns about the elicitation done, and also the framing of the piece in relation to development.

Expert elicitation, when done credibly, is a very meticulous process of identifying expertise and regulating the kinds of information and intuition you want to elicit while keeping the various biases and cognitive issues that can degrade data quality at bay. You also want to make sure that information acquired from experts reflects the expertise you seek, and is not ultimately controlled and shaped by the analysts that ask the questions and synthesize the results. I think at broad levels the authors adopt a good structure, but I have some reservations in the specifics in the identification, training and structure of the elicitation process. I also provided important references (which are a brief selection of articles in entire literatures on the topics).

I appreciate the description of experts chosen for the elicitation. When I first read the paper boast 200 experts my immediate reaction was "that's too many to handle effectively". However, I think the substructuring of expert teams into groups of 20 is well done (though I would say 20 is on the high end of good elicitation methods). The authors also demonstrate relevant information showing some dimensions of variation in experts involved in the process. However, I have some hesitation with the treatment of "lead experts" and their role in the further selection of experts I'm hoping the authors can clarify, expand on, or reflect on in their draft.

Selection of lead experts is vaguely, "identified based on their relevant expertise". I will note that expert status (as determined by credentials and markers of prestige) often does not have any bearing on performance or accuracy of expert input, and can even be counterproductive. The lead experts were also asked to identify subsequent experts. This kind of snowball approach to identifying experts is not necessarily problematic but can introduce issues of similarity biases ("groupthink"). Notably, focusing on experts from scientific initiatives is likely to leave out experts with local knowledge not often valorized in official and scientific working groups. Did the authors do any calibration to see if they collected a sufficiently broad range of expertise? The total number of experts may not represent knowledge breadth if experts are from similar training and collegial working relationships. Choosing diverse experts is analogous to increasing the "degrees of freedom" in a dataset – and asking similarly trained experts and colleagues will severely degrade the independence between experts and be akin to pseudosampling. See the following references for discussions on expert selection and elicitation processes

Burgman, M. 2005. Risks and decisions for conservation and environmental management. Cambridge University Press.

Burgman, M. A., M. McBride, R. Ashton, A. Speirs-Bridge, L. Flander, B. Wintle, F. Fidler, L. Rumpff, and C. Twardy. 2011b. Expert status and performance. PLoS One 6:e22998.

Singh GG, Sinner J, Ellis J, Kandlikar M, Halpern BS, et al. (2017) Correction: Group elicitation yields more consistent, yet more uncertain experts in understanding risks to ecosystem services in New Zealand bays. PLOS ONE 12(12): e0190326

Tetlock, P. E. (2005). Expert Political Judgment How Good Is It? How Can We Know?, Princeton University Press.
-While this book covers experts of social science, it is one of the best works documenting the limitations of expert status. It also largely corroborates the work Mark Burgman showed in conservation scientists.

I would suggest that one of the best ways to utilize a snowball approach to address knowledge breadth is to snowball across experts (and not just the "lead" expert) and track the "network" of experts to see if the expert pool saturates (or includes the majority) of potential experts. This process is probably modeled best in the following paper:

Ban, S. S., R. L. Pressey and N. A. J. Graham (2014). "Assessing interactions of multiple stressors when data are limited: A Bayesian belief network applied to coral reefs." *Global Environmental Change* 27: 64-72.

If the authors did not take this quality assurance step in their methods, I would encourage them to consider the implications of their results. Are there hidden uncertainties in their results? At worst, if there are significant selection biases in their experts the uncertainties would be systematic (biased) and the large sample of experts may actually reinforce the bias rather than reduce uncertainty.

There is also very little information on how experts were elicited to quantify their BII scores. There is much research to indicate that experts can more reliably provide ranges of estimates than point scores, and even then it is worth "nudging" experts by questioning their certainty to see what scores they provide.

Morgan, M. G. (2014). "Use (and abuse) of expert elicitation in support of decision making for public policy." *Proceedings of the National Academy of Sciences* 111(20): 7176-7184.

Speirs-Bridge A, Fidler F, McBride M, Flander L, Cumming G, Burgman M. Reducing overconfidence in the interval judgments of experts. *Risk Analysis*. 2009;30(3):512–23. pmid:20030766

I would encourage the authors to justify how species groupings were done relative to expert judgement. I understand the ecological justification for species groups, but it would speak more to data quality to relate these decisions to expert

familiarity. I understand the pragmatic necessity to group species given the vast numbers of species that are being addressed, but how species are grouped will affect expert scores. If the groups are made in such a way that experts are not familiar with a grouping or thinking about a group of species in the way identified, the quality of the scores will likely be questionable. I would extend the same comment to the grouping by biomes. Again it may make ecological sense, but the authors should relate the groupings to expert familiarity to help assess data quality.

The author description of identifying and describing land uses shows built-in training in the elicitation process, which is great, but I found the writing of the section confusing because of the constant switch between what was done and why other scale considerations would not be as good. First, in asking experts to think of “average” landscapes based on visual cues of land uses and descriptions – people are prone to the “availability bias” when trying to imagine representatives, which may vary considerably across people based on their experiences. Perhaps that was part of the design and the aggregation stage was meant to address this, but I think in communicating the methods the authors should address how they dealt with these cognitive biases.

My more serious concern here is that while experts were asked to consider integrated impact of all characteristics of the landscape, it seems that there is potential for experts to be considering different problems (because different experts may be identifying different impacts). This so-called “linguistic uncertainty” whereby experts may translate “all characteristics” differently can lead to underappreciated uncertainty and instability between experts.

Regan, H.M., Colyvan, M. and Burgman, M.A. (2002), A TAXONOMY AND TREATMENT OF UNCERTAINTY FOR ECOLOGY AND CONSERVATION BIOLOGY. *Ecological Applications*, 12: 618-628. [https://doi.org/10.1890/1051-0761\(2002\)012\[0618:ATATOU\]2.0.CO;2](https://doi.org/10.1890/1051-0761(2002)012[0618:ATATOU]2.0.CO;2)

I think the IDEA process is a good structure for elicitation, especially with iterative feedback. I do however have some comments on some of the details within the process presented. There is much research to indicate that experts can more reliably provide ranges of estimates than point scores, and even then it is worth “nudging” experts by questioning their certainty to see what scores they provide. The issues I present on the (lack of) details presented in how experts were elicited extends to the construction of the decision-tree used for the mapping – who was involved and how was this done? Finally, I note the extreme range of the numbers of experts involved in different groups of species. I understand the need for pragmatism (again) but the authors need to consider the range in reliability and robustness of responses as a result of this.

Morgan, M. G. (2014). "Use (and abuse) of expert elicitation in support of decision making for public policy." *Proceedings of the National Academy of Sciences* 111(20): 7176-7184.

Speirs-Bridge A, Fidler F, McBride M, Flander L, Cumming G, Burgman M. Reducing overconfidence in the interval judgments of experts. *Risk Analysis*. 2009;30(3):512–23. [pmid:20030766](https://pubmed.ncbi.nlm.nih.gov/20030766/)

I have made some mention of this already, but I am concerned about the extent to which results actually reflect experts and not the analysts. The expert input feed into a pre-defined notion of what biodiversity intactness is and is only part of equations to calculate this. Was there any work to engage experts to see if the definitions, standards, and reference points of the analysis correspond with how the experts think of the problem? Because of the bounding of the analysis, have the authors considered that the results are more affected by the pre-defined equations rather than the expert input? For a study showcasing expert input, I find the lack of engagement with these very fundamental issues troubling.

Aven, T. and S. Guikema. 2011. Whose uncertainty assessments (probability distributions) does a risk assessment report: the analysts' or the experts'? *Reliability Engineering & System Safety* 96:1257-1262.

Beyond the expert elicitation, I also have issues with the framing of the paper and some more quantitative analysis. I go into the specifics below

Title – since this assessment is for sub-Saharan Africa, the title should reflect that and not suggest that it is for the whole continent. This is to address both accuracy issues as well as avoid the often-made conflation of “Africa” as some homogenous place instead of diverse continent.

Line 66 – The links between biodiversity, and any specific status of the environment, with global sustainable development, are tenuous and not at all settled. The author refer to literature on the Planetary Boundaries, but this framework has been heavily criticized throughout its history, and perhaps most aggressively on issues of biodiversity. I know that the planetary boundaries authors have replied to many challenges, but in my view (and many others in development communities, and throughout many science disciplines) the responses do not address the severe challenges and problems raised. I am not suggesting the authors retract their statements, but they must make them in context of the unsettled nature of the debates if raised.

Biermann F, Kim RE. The boundaries of the planetary boundary framework: a critical appraisal of approaches to define a “safe operating space” for humanity. *Annu Rev Env Resour*. 2020;45:497–521.

Montoya, J. M., I. Donohue and S. L. Pimm (2018). "Planetary Boundaries for Biodiversity: Implausible Science, Pernicious Policies." *Trends in Ecology & Evolution* 33(2): 71-73.

You do not need to overextend the importance of biodiversity to make the case for this study. Or if you do, it should be done

fairly, while pointing to the variety of evidence that disagrees with these kinds of statements. Highlighting the local and regional importance of biodiversity is likely enough to point out, and this will be more accurate and less contentious

Lines 113 -The authors make the point that there are limitations of top-down approaches to understanding ecosystems and points to the novelty of their approach in consulting local experts. However, it's not clear to what degree the authors have considered that the structure of their analysis is very top-down: thinking of biodiversity intactness in reference to pre-industrial society is a subjective standard that historically largely follows European notions of nature. Have the authors looked to see if the experts they consult follow these notions? These kinds of questions are important because if the experts do not think on these terms, then the responses they provide may not be as robust. That is, if questions are posed in ways that experts are not familiar with, they may not be accurately considered experts.

Results –

Some results seems rather tautological. Since the BII is defined as intact nature relative to preindustrial settings, this measure shares the same information with indices about human activity and human footprint. I understand these relationships are partially meant to validate the measure, but here I have some questions as well. First, given the shared information, it's not really surprising that the best relationship seems to be between the BII and the Human Modification Index compared with the biomass modification and habitat indices. Perhaps this shows some evidence of predictive validity of the measure. However, it speaks to questions about the potential construct validity of the measure: since it does a better job tracking human footprint than it does biomass and other ecological variables, is it tracking what we want it to? I think questions about the tautological nature of the validation, and a broader discussion of validity is needed.

Second, the plotted figures only show the “average” indices of human modification and habitat indices in each ecoregion. To better represent the data, the variance or some measure of spread should also be illustrated. This will help understand how closely the BII corresponds to the range of measures used.

Quantitative analysis

-since mean \pm 95% CI was calculated across pixels, the authors need to be cautious of spatial autocorrelation effects potentially inflating measures of precision, and affecting statistics calculated. I encourage the authors to estimate sampling distances needed to eliminate spatial autocorrelation, then randomly sample pixels given this distance to calculate the mean and CI estimates.

Did authors account for test assumptions when conducting correlations and t-test? Independence is already questionable, but do other assumptions hold? Did the authors do any diagnostics? Some of the relationships look potentially heteroskedastic, for example, meaning the error structure varies (which violates a lot of statistical model assumptions). Perhaps nonparametric tests are more relevant.

(Remarks on code availability)

Version 2:

Reviewer comments:

Referee #1

(Remarks to the Author)

Firstly, to avoid confusion, I was reviewer 1 in the first round. I would like to thank the authors for their comprehensive approach to responding to my and other reviewer's comments. I believe the changes they have made have significantly improved the manuscript. I am aware that my suggestions for validation steps has required quite a lot of new work to undertake, and I believe have helped inform the analysis. I am boardly happy with this, and the manuscript remains easy to read throughout.

One thing the new analysis (and Figure S6 in particular) highlights is the considerable noise in these species-level assessments. I agree with the authors that it is reassuring that IUCN Red-list categories can be differentiated statistically in the BII scores, and this gives me some confidence in the remaining results. However, I think the authors need to be rather more transparent that this statistical significance is actually based on very tiny average differences in BII between groups - and there seem to be species IUCN list as LC that are apparently 80% depleted, and CR species that have only lost 5% of BII. Given that IUCN Red listing is also an expert based process, and despite the author's arguments in their response letter that there may not be be congruence between BII and RL status, I simply don't buy that all the species that make this possible are a result of those processes. Maybe the SM could pick some of those examples and transparently explain what is going on for a few idenitified species? The existence of such wide spread of BII scores within RL categories suggests to me that while on average overall patterns may be identified from these methods, we'd be really unwise to read anything into the details of each species. I think this uncertainty and the species-level errors that I'm sure are here should be given a bit more prominence. I know the authors want to present their results in the best possible light, but I think the overall aptterns identified will hold while they remain transparent about what can't be read into these data.

I also thank the authors for providing the github repository. However, this does not include the code used to generate the

plots or analyses in the text - it seems only to be the code for generating the 1 or 8km rasters, extracting covariate datasets and processing country-level analyses. I can't see any of the statistical analyses or plotting code. It is also seriously lacking in comments.

(Remarks on code availability)

The code as currently provided probably enable a very persistent analyst to reproduce the core workflow for generating maps. It does not seem to allow the reproduction of the figures and statistics described in the manuscript. It is not well commented.

Referee #2

(Remarks to the Author)

Clements et al. are appreciated for revising their manuscript (Sub-Saharan Africa's biodiversity intactness assessed with place-based knowledge). I am glad and thank the authors for addressing all my concerns and suggestions, particularly the addition of caveats about the place-based approach they used in the data collection, rearrangement of the results (as I suggested), and revision of the discussions accordingly. While I strongly think the manuscript has improved, some parts are somewhat too long to follow in the method section. I have highlighted that and few other things in the comments below. (Please note that my comments were based on the line numbers in the untracked PDF file.)

L1: I really like this revised title - it captures the study well.

L210: Should you have "followed by grassland (68%) biomes" here again when the highest and least have been mentioned?

L288-321: I do not really get the narratives here and what exactly they are addressing. Or you probably need to revise it for conciseness?

L636: No file was labelled "Supplementary Information 2".

L709-710: No file was labelled "Supplementary Information 2".

L625-782: I believe you have added new paragraphs here, possibly because other reviewer(s) called for that, but this is rather too long, particularly for the main manuscript. You might need to break it into sub-sections and maintain brevity for clarity.

L814, 861 and other places: "Supplementary Information" not "Supplementary Information 2". You might check the file name for this.

L866: Supplementary Information

L811-890: Too long to understand. Summarise for brevity and clarity.

L977-1031: Thank you for providing these caveats. I believe these will help in building on the place-based approach in this study and trigger new innovations for monitoring biodiversity.

L1036: remember to add this identifier

(Remarks on code availability)

Unfortunately, I did not go through the codes this time. So, I hope other reviewers would do this.

Referee #3

(Remarks to the Author)

I thank the authors for their thorough work in revising the manuscript and generously considering the comments from all three reviewers. I have gone through the responses to my earlier concerns and I was largely pleased with the extra details given to the i) recruitment of experts, and ii) the process of elicitation. Overall I found that my concerns were ameliorated, since it does seem like the authors did a thorough job following best practices in doing good expert elicitation, and they have provided much more detail highlighting that. However, I have lingering concerns about the implications of the elicitation in how the results are presented. Notably, because of the pragmatic decisions that authors needed to take, there are potential biases introduced in the data.

I appreciate the difficulties in recruiting experts generally, and even more so for such an ambitious undertaking (200 experts across taxa for almost a continent). I applaud the authors in their efforts to include experts that otherwise routinely are left out because they do not fit classic definitions of "expert" based on status and credentials. In general I found the authors' commitment to expert elicitation processes (from recruitment to execution) reassuring, however I would caution against any notion that adopting best practices mitigates all issues with error in results.

I do think given the extra text that the authors contributed significant effort to recruit relevant experts. However, their new text reveals that they could not fully mitigate selection biases in expert inclusion into the process. As other reviewers have also pointed out, there are geographic disparities in expert participation (notably for experts from Africa, most are from South Africa). I wonder if there are more selection issues present, notably in socioeconomic and employment variables. For example, it is widely known that South Africa is the productive (in terms of academic publications) of African countries and potentially also most likely to be similar to experts from the Global North.

Further, some details about the difficulties in recruiting and retention of experts make me think there may be some systematic selection biases in the experts that provided responses. As an example, the authors note that not all 200 experts were able to take part through the conclusion of the exercise. Attrition often follows specific systematic and particular causal reasons, and attrition bias is a pernicious effect that can render the accuracy and validity of measures suspect. Often participants (or in this case, experts) with other commitments, less economic freedom or roles focused on research having to leave early. If,

in these cases, the people who left were experts with more on-the-ground experience in wildlife management, might the authors' quantitative assessments underrepresent wildlife abundances? This is my main point – that commitments to best practices are great, and pragmatic decisions are understandable, but neither eliminates or mitigates error (especially systematic error).

I appreciate the practical limitations of this study (such as reaching saturation of relevant experts). I don't think every study needs to be perfect in order to be published, however where there are practical considerations that mean a study falls short of a design that would ensure the scope and accuracy, then the study needs to reflect on the errors introduced by the practical considerations of the design. Any study with the scope of this one is prone to error propagation and/or systematic error and this needs to be reflected in the results and discussion.

I do also appreciate that the authors have taken the time to further do validation tests (such as compare against IUCN datasets) but I think this may be yet another database that potentially lines up with some of the systematic biases I'm talking about. That is, my understanding is that IUCN data is usually the result of researchers from the Global North and researchers that conform to traditional Global North standards of expertise (such as credentials). So in some ways it's odd to validate a dataset supposedly built on local and place-based knowledge with more traditional datasets. Validation is a very difficult challenge in these settings and I do not envy the authors in wrestling with this. I say all this not to say that the authors are wrong but that I don't think there are easy analytical and quantitative methods to validate the results or mitigate the potential problems of systematic error. Rather the authors need to acknowledge potential sources and consequences of error (and I would prefer this would be in the results and discussion rather than the methods, but I leave that to the authors and editors).

Importantly, reflecting on the potentials for systemic error mean that the authors need to do more than note uncertainties reflected in the confidence intervals around mean estimates (lines 998-999, and repeated in the response to my earlier review). Confidence intervals only reflect potential effects of random error and not systematic error from issues like selection effects. Systematic errors have potential directional consequences to estimates (such as the example I give above regarding the attrition bias possibility).

Beyond these issues of translating the challenges of expert elicitation to the results gathered, I also have some lingering concerns about some of the specific results, especially quantitative results.

First, before my concerns, I'm glad about some of the changes made to the analyses. Regarding my comments on correlations and t-tests – I was happy to see that non-parametric measures generated the same conclusions as previous. My comments on diagnostics had to do with parametric correlative tests (regression is not a causal analysis, but a correlative one).

Second (and somewhat minor), my understanding of the authors response to my comment about the BII results being somewhat tautological is that instead of validating the BII, the authors are showing that their regional BII does better than the global BII. I do think this is a worthwhile result and I apologize for not understanding it on first read. All responses about validation, importantly, are about comparing the authors' map with indicators of human impact and landuse change (or consequences of impacts, such as species threat status), so they all do share information. I think it would be good to make this more explicit (that the focus was to show the lack of relationship with the global BII). It might help prevent misreads from people like me.

More importantly, I do not agree with the authors response about dealing with spatial autocorrelation. Autocorrelation is about shared information and how this affects estimates. Whether or not the data comes from reference data or expert input, data from nearer sites are likely to share information that sites farther away. In fact, if expert scores did not reflect this spatial similarity, I would be highly suspect of the accuracy of expert data (at least in principle). In computing confidence intervals, if samples are included that have shared data (sites closer together are more likely to be more similar) then the reported confidence intervals are likely too narrow due to pseudoreplication. That is, the reported values would reflect overconfidence in the scores as the authors are representing shared information as independent information. Again, I think the authors need to assess whether spatial autocorrelation is present and if necessary recalculate their confidence intervals.

(Remarks on code availability)

Version 3:

Reviewer comments:

Referee #1

(Remarks to the Author)

I've looked again at the manuscript and the authors responses. I think they have made some great progress towards meeting the reviewers' comments. I am particularly pleased to see some analysis of the exaple species withere RL and BII scores differ greatly - one minor addition that could go in that table is a simple column indicating whether the authors think reality for each species is better reflected by BII or RL status.

I also thank the authors for their work on the code at the figshare site. I still can't actually run it as data are missing, and the

commenting is still rather minimal, but it does look OK.

If there was one cheeky last-minute thought on looking again at the whole manuscript it would be to ask the authors to make a little note that a limit with BII is that it can only go down - species that are 'winners' in the anthropocene are capped at 100%. It's a general problem with BII type indices, but worth noting.

(Remarks on code availability)

I've done the review I can - looks OK, but I can't actually run it.

Referee #2

(Remarks to the Author)

Thank you for addressing all the concerns. Please consider "estimates" instead of "estimate" in L300 and delete the second "that" in L398.

(Remarks on code availability)

Referee #3

(Remarks to the Author)

I again thank the authors for their thorough work in revising the manuscript and generously considering the comments from reviewers. This work is no small task, and I again congratulate them on the work. Here I reply to the authors regarding their edits and the concerns I raised previously.

Overall, I think the authors have done a considerate job addressing my concerns, especially in clarifying their manuscript regarding the analyses conducted and the processes their analyses were based on. There are multiple layers of data (expert and geospatial) data that the authors analyze and summarize, and clearly clarifying what each analysis is based on really helped clarify and alleviate issues I had with the manuscript.

In summary, I am pleased with the authors' inclusion of text acknowledging potential systemic error into the manuscript, related to the pragmatic realities they faced in expert recruitment, and especially in acknowledging the kinds of processes (e.g. digital inequities, language barriers) that may have contributed to these errors. I also appreciate the acknowledgement of potentials for bias affecting some of the quantitative estimates (such as around mean scores and Cis). Finally, I thank the reviewer for clarifying the source of information on the specific results, and their considered response to my concerns around spatial autocorrelation. I think the authors are correct in that I did not realize that the reported mean and 95th Cis are about expert responses solely and not samples from the mapped results. The author's clarification on the spatial autocorrelation make sense to me.

My main lingering concerns are in the communication and implications of error. I will leave it to the editor to decide whether you are expected to respond to this, but while I appreciate the acknowledgement of potentials for systematic error/bias, I would prefer some thoughts around how this might affect results. So, if the process filtered out experts from given backgrounds, do you have some sense of what this might mean for the kinds of information you relied on? For example, might it have limited the considerations of species functional response groups, biomes, land uses, and then the intactness scores? So, could there have been some qualitative effect on what kinds of results you can report on? Do you have any sense of what directional effect these might have had on quantitative estimates?

More importantly, I still have a bit of a lingering concern around the communication of validation. In the manuscript and response, the authors suggest that comparing against other assessments (IUCN Red List, previous BII assessments, other literature) are useful checks against biases that I raise. However, I didn't only mean that these other assessments also have biases different from the authors' assessment (they do, and comparing against multiple assessments with different biases is a good way to triangulate findings); my point is that these assessments will have shared biases with the authors' assessments. In fact, they are likely more biased in regards to expertise than the authors' assessment, in the emphasis on western Science, reflecting expertise of Global North researchers receiving similar training, etc. I would prefer it if the shared biases between all these assessments was acknowledged. That is, in principle they are not fully independent. Again I don't mean to take away from the contribution of this piece, and validation for such an endeavor is very difficult, but the limitations of the validation work done here should be properly acknowledged.

(Remarks on code availability)

Dear Dr Clements,

I'm writing in the temporary absence of my colleague, Dr Anna Armstrong, about your manuscript "The biodiversity intactness of Africa assessed through local knowledge". We sincerely apologize for the time it has taken us to get back to you about the paper. However, we now have comments from three referees (appended below). When you read their comments, you will see that they raise serious concerns about the validity of the methodological approach you have chosen (for this application) and raise concerns about its implementation. These are quite substantial concerns, but we also recognize the importance of the work. We therefore ask that you send us a formal response - in the form of a point-by-point rebuttal - to these concerns. We will then use this response to determine whether we can consider a revision of this paper.

Please let me know if you have any questions. Otherwise, we look forward to receiving your response in due course.

Best wishes,
Mary Elizabeth Sutherland, PhD
Senior Editor

Referee expertise:

Referee #1: savanna biome, conservation, intactness and quantitative methods

Referee #2: in situ expertise

Referee #3: conservation, expert elicitation

RESPONSE: Thank you for the opportunity to address the reviewers' very helpful feedback on our manuscript, and the recognition of the importance of our work. Please see our accompanying cover letter where we elaborate on the key reviewer concerns and how we have addressed them. In this rebuttal document we have responded in detail to each point raised by the reviewers and outlined how we would revise the paper to address the concerns. We have already implemented the majority of the revisions that we propose. We have also provided a table summarising the main points of feedback from the reviewers and how we addressed them.

Table summarising our responses to the major points of critique

Reviewer Critique	Key Response(s)	Proposed Changes
1. Misalignment with IPBES definitions of 'local knowledge' and 'co-production'	Agreed with reviewers regarding local knowledge; replaced 'local knowledge' with 'place-based knowledge' and provided justification for why place-based knowledge is more suitable for this study. Justified why our iterative, interactive, engaged process is a form of knowledge co-production.	Replaced 'local knowledge' with 'place-based knowledge' throughout the manuscript and in the title; added definitions and explanations in the introduction and methods
2. Concerns about approach		
2a. Expert selection: biases and reliance on regional and 'non-local' experts	Elaborated on selection process and efforts to ensure diversity; reflected on challenges; clarified the selection criteria: place-based knowledge (which can include but is not limited to local knowledge).	Added additional details on the expert selection process in methods and referred to Clements et al. 2024 ¹ Figure 2 summarising expert numbers according to key dimensions of diversity; include caveats section

		reflecting on challenges and limitations of our approach
2b. Need to demonstrate that the contextualised generalisation approach (functionally grouping species and categorising land uses) led to robust expert estimates	Elaborated on the approach; conducted a new validation using the well-established species threat categories of the International Union for the Conservation of Nature (IUCN) Red List (from Critically Endangered to Least Concern), as suggested by Reviewer 1, and the results demonstrate the validity of the approach.	Added validation test to manuscript methods and results, and elaborated on elicitation process in methods
2c. Expert elicitation process: linguistic uncertainty, groupthink, overconfidence, expert familiarity with concepts and categories, availability biases	Explained the structured expert elicitation process (IDEA protocol) and efforts to minimise these potential issues; referred to new validation using IUCN Red List threat data that demonstrates robust outcomes.	Added additional details of expert elicitation process in methods; added new validation test; include caveats section reflecting on challenges and limitations of our approach
2d. Approach was top-down	Explained contextualised generalisation approach as a way to embed more place-based knowledge from an under-represented region into globally established indicators that are already embedded in global policy (e.g., Global Biodiversity Framework). Reflected on how this contributes to decolonising ecology while also acknowledging the limitations of our approach.	Added definition and explanation of place-based knowledge and highlighted need to decolonise ecology in the introduction and methods; include caveats section reflecting on challenges and limitations of our approach
3. Some figures need disaggregation or improved presentation; some misalignment in their order compared with the text	Reorganize figures (e.g., splitting Fig. 1a), rework figures (e.g., Extended Data Fig. 2; Fig. 3); improve explanations in legends and methods.	Split figure 1 and reorder first section of results; reworked Figure 3 and Extended Data Figure 2; and clarified legends and explanations
4. Missed opportunities to discuss key patterns and drivers of biodiversity intactness	Agreed—add discussion of regional patterns, comparisons with other BII maps, and drivers of notable results (e.g., major declines in Nigeria and Rwanda; less decline than expected in several countries with extended conflicts)	Expand discussion section; reduce repetitive text to make space for two new paragraphs

Referees' comments:

Referee #1 (Remarks to the Author):

This is a really interesting paper describing a first proper attempt to estimate the biodiversity intactness across the African continent. Recognising the challenge involved in direct measurement of this the authors opt for an expert-based assessment, identifying a cohort of over 200 taxonomic experts to help in the process. They find that relatively small proportions of current biodiversity are across the continent fall in protected sites, with much the larger part remaining in unprotected but untransformed landscapes, co-occurring alongside human activities. These results are important and broadly in line with what I would have expected - I'd love to have seen some comparisons with other continents to put this into context, but as a top level result this seems plausible.

RESPONSE: Thank you for the positive feedback; we are pleased that the reviewer thinks that the paper is interesting and the results important. Comparisons with other continents are difficult to do meaningfully at this stage since no other regional maps of BII are available (and in the paper we highlight the concerns raised about the validity of the global maps of BII). Hopefully, our work will inspire similar research in other regions, allowing for such comparisons that we agree would be most interesting.

I have to admit I was excited about reading this paper - Local and Indigenous Knowledge is a hot topic in current biodiversity discussions. IPBES has been praised for delivering a better job on this than many previous attempts, and the K-M GBF identifies improving its involvement as a key target. So I was slightly disappointed that despite the title and the talk of co-developing knowledge, which made me think there was going to be a new method of harnessing indigenous knowledge, this isn't really the case at all. Rather, the underlying data are the result of a fairly standard Delphi process involving experts from across the world (though in reality a disappointingly small fraction of experts are not from N. America, Europe or South Africa). I struggle to really see this as 'local knowledge' when, for example, only two of the ornithologists are West African (both Nigerian), despite the existence of an active ornithological community in the region. To suggest that Southern African ecologists are more local to East or West Africa than geographically much closer European ecologists is not aligned with the CBD or IPBES use of the term 'local knowledge'. Nor do I see any sense of 'co-production' here in the usual usage of this term - I suggest the authors considerably revise their use of local and co-production.

RESPONSE: In light of this feedback and after consulting IPBES and recent literature, we agree that our use of the term is not aligned with IPBES. We believe the term 'place-based knowledge'² is more appropriate for this paper and we have changed the term accordingly throughout the paper. While 'local' and 'place-based' both refer to knowledge that is contextual, the former is held by local people while the latter encompasses diverse forms of knowledge—including scientific, experiential, and local—rooted in particular landscapes or regions^{2,3}.

In our previously published data paper (Clements et al.¹) describing the expert elicitation process and resulting 'bii4africa' dataset, we explain that *"A broad definition of expertise was used to identify experts, centred on experience of how sub-Saharan species are impacted by human land uses. Diverse types of people can have such experience (e.g., researchers, field or tour guides, park rangers, conservation practitioners, museum curators, and consultants), and inclusion was thus not limited to specific qualifications or institutional affiliations."* We have added this sentence to the methods of the current paper (in the 'Intactness scores' section) and elaborated as follows: *"Many, but not all, of the participating experts were scientists (107 of the 200 have university affiliations)." "Most of them reside in Africa (72%) and/or are originally from Africa (59%), but*

those not from or currently residing in the region still met our criteria of having place-based knowledge as a result of working extensively in the region (mostly as scientists but also as e.g., tour operators or guides, for conservation organisations, etc). This place-based knowledge informed the creation of a quantitative dataset through an iterative, context-based and interactive expert elicitation process.”

While we agree with the reviewer that a standard expert survey does not qualify as knowledge co-production, we think that our *“iterative, context-based and interactive expert elicitation process”* does meet the requirements of a knowledge co-production process. Knowledge co-production is defined as *“Iterative and collaborative processes involving diverse types of expertise, knowledge and actors to produce context-specific knowledge and pathways towards a sustainable future⁴”*.

Our process aligns with the four principles of co-production laid out by Norstrom et al.⁴ as follows:

- 1. Context-based.** *Co-production processes should be considered and situated within the particular social, economic and ecological contexts in which they are embedded:* Our process specifically focused on place-based knowledge for particular land uses in the sub-Saharan African context.
- 2. Pluralistic.** *Co-production of knowledge must explicitly recognize the multiple ways of knowing and doing.* As quoted above, our definition of expertise recognised experienced-based knowledge held by diverse types of people. We also elaborate in response to Reviewer 3 on how we aimed to capture diversity in expertise in our expert selection process and provide a summary of the different dimensions of diversity across the expert group.
- 3. Goal-oriented.** *Knowledge co-production for sustainability is problem-focused and benefits from clearly defined and meaningful goals shared among participants.* We had the specific goal of mobilising expert knowledge to produce intactness scores that could be used to produce a regional assessment of biodiversity intactness to feed into national and international decision-making. This goal was introduced to all contributing experts in the invitation email and introductory meeting, as detailed in response to Reviewer 3. We also had the more immediate goal of making the co-produced bii4africa dataset of intactness scores widely accessible and this has been achieved through the publication of an open-access paper in *Scientific Data* that includes contributing experts as co-authors (Clements et al.¹).
- 4. Interactive.** *High-quality co-production requires frequent interactions among participants to occur throughout the process, extending from collaboratively framing and designing the research agenda, to conducting the research, and jointly using and disseminating the knowledge generated.* We elaborate in response to Reviewer 3 that our expert-elicitation process was iterative, and that during two meetings experts collaborated to produce the species functional groupings, as well as engaging in critical discussion around the estimates that emerged from the elicitation. Experts then contributed as co-authors to the publication of the resulting dataset, where they were given the opportunity to contribute to two drafts of the paper, as well as the revised paper subsequent to review. The lead author addressed 193 comments on aspects of the paper from contributing experts. As explained in response to Reviewer 3, lead experts were involved from the start of the project, including assisting in conceptualising the process; deciding on the reference state, the land use class descriptions and variables, the biomes; mapping and assessment; and results interpretation.

Place-based expert assessment (though not always through a structured, iterative elicitation as done here) is the backbone of widely accepted biodiversity indicators and assessments including the IUCN Red List, IPBES and IPCC and an important form of knowledge production. The place-based knowledge we make use of in this paper is knowledge of how different functional groupings of species in the sub-Saharan African region respond to land uses characteristic of the region. We estimate biodiversity intactness across the region by generalising this place-based knowledge

across similar land use contexts in a bounded way. Our approach aligns with recent research emphasising the need to integrate place-based research and knowledge into global sustainability policies and actions^{2,3,5}. Specifically, our approach speaks to “*the concept of region as a bridge between local and global sustainability initiatives*” to overcome cross-scale integration challenges (Balvanera et al.² pg. 1) through ‘contextualised generalisations’⁶. We will explain this important contribution of our approach and work in the revised introduction. We hope this also clarifies why our approach is not dependent on the extent to which experts are local knowledge holders, but rather the extent to which they hold place-based knowledge within the region.

Importantly, our approach presents an alternative to commonly used global mapping approaches that do not account for regional context⁷. In the paper we give the examples of croplands and rangelands: global maps typically do not differentiate between smallholder vs large-scale commercial croplands, nor pasture vs rangelands, but these distinctions are critical in the sub-Saharan context for a wide range of impacts, including on biodiversity, livelihoods and ecosystem services⁸⁻¹⁰. Our approach helps differentiate these impacts in ways that are often ignored in large-scale mapping exercises. We trust that our new validation against the IUCN dataset also confirms that our contextualised generalisations are robust.

Despite this slight disappointment caused by what might be considered some of the hype in the writing, the paper stands or falls on the basis of the work these 200+ experts undertook. For them to even attempt such a comprehensive review is highly commendable.

RESPONSE: Thank you for your commendation of the enormous effort this involved.

The difficulties of this are clearly acknowledged, and have resulted in grouping of species and geographies in order to make the problem tractable. It is therefore crucial that we trust this data reduction. If I have understood the methods correctly, each expert identified their particular expertise and were then asked to estimate an average score for the change in population of functional groups within their taxonomic speciality by landuse and biome. Thus an ornithologist would be asked to provide estimates for each of 17 groups of birds in each of 8 biomes, within which there may be up to 9 land uses. This would be (up to) 1224 total estimates for an ornithologist. (NB this summary wasn't easy to piece together involving both the paper for review and a cited paper and searching around the project website too and might be wrong, it would be great to see an example like this clearly laid out somewhere.)

RESPONSE: To address the reviewer’s important comment that it was not easy to piece together what was done, we propose to expand the methods section, and also include a Supplementary Methods document with a schematic of the expert elicitation process that provides a clear summary of the approach that is described in detail in our recently published data paper (Clements et al.¹). Importantly, experts only estimated for the land uses and functional groups (and biomes, for plants) that they knew well. As reported by Clements et al.¹: “*On average, each expert provided 155 intactness scores*”. We will add this clarification to the methods section.

Hidden in here is the idea that an individual ornithologist from Nigeria, say, could reasonably estimate the average % decline in aerial feeders (swallows and swifts) in all the savanna areas of Africa across the 9 land use tasks. I am deeply skeptical that any expert could do this. While it might be possible within one's own country where one's local knowledge is valuable, to estimate it across all the Acacia savannas in the continent as a whole seems implausible. Moreover, the reader is left to trust that the groupings are reasonable (I couldn't really work out what they were fully) and that the implicit assumption they respond similarly within a biome to land use change is reasonable. I would like to see this tested at least at some level. Perhaps a quick evaluation of the current trends

for those species included in the IUCN Red List would be useful - if we are to trust these data I would like to see agreement between the average red list category for each functional group and the expert-based estimates of this, and I'd like to see that the IUCN status of species within each taxonomic grouping is more similar than between functional groups.

RESPONSE: The reviewer is correct that our approach assumes that the population abundances of species within a 'response/functional group' are similarly affected by different land use changes. This published approach¹¹ of focusing at the level of functional groups instead of individual species to make the problem tractable is what makes it possible to translate experts' place-based knowledge into a regional map. To ensure rigour in these functional groupings, the groupings were themselves informed by the knowledge of the expert group through a structured, interactive, and iterative process. As described by Clements et al.¹: "*The lead expert proposed a draft set of species response/functional groups, based on their knowledge of the key organismal attributes likely to determine the impact of different land uses on populations. These draft species response groups were presented to the participating experts for that taxonomic group during an introductory planning meeting (see Structured expert elicitation section below) and revised based on experts' feedback*". We will include this detail in the Supplementary Methods document and have also added the following sentence to the methods section "*These groups were decided on by the participating experts during the introductory phase of the elicitation process and are summarised in the 'Sp_Groups' tab of the bii4africa dataset¹*" which is available online at <https://www.nature.com/articles/s41597-023-02832-6>. We will also provide examples of the groups in the new Supplementary Methods for easy reference.

We concur with the reviewer's concern that we need to provide additional evidence that our approach led to robust results and thank the reviewer for suggesting we use the IUCN Red List to do so. Below we present results of a test akin to the first comparison suggested by the reviewer (i.e., testing "*agreement between the average red list category for each functional group and the expert-based estimates of this*"), with the difference being that because our map can be disaggregated into its component parts we can actually do this comparison at species' level. In other words, we can assess whether a species' IUCN Red List category is a good predictor of its remaining intactness (i.e., the average intactness of its functional group across that species' particular range). However, we disagree on the usefulness of the second comparison that the reviewer suggests (demonstrating that species within functional groups have similar IUCN threat statuses, compared to species between functional groups). This comparison would not hold up for species' IUCN threat status, since the threat level for a species within a functional group depends strongly on the species' range and the land uses it is exposed to within that range. The threat level for species within a functional group can therefore vary widely, and comparing some kind of average threat status between functional groups is not informative. Our approach to estimating BII accounts for individual species' ranges and does not assume that all species within a functional group have the same range.

While we would expect to see general congruence between the predicted intactness of a species within its range and its IUCN threat category, it is important to note *a priori* that the relationship is unlikely to be perfect for several reasons. Firstly, intactness is estimated relative to pre-industrial populations while Red Listing is based in part on the proportional population decline over the past 10 years/3 generations (or projected into the future). Thus, if most population decline occurred before this period, the species may have lower intactness than expected based on its IUCN threat status. Similarly, if large future declines are projected, the species may have higher intactness than expected based on its IUCN threat status. Secondly, a species with a smaller absolute population size and/or smaller range size will be considered more threatened according to the Red List threat classification process, meaning two species with similar intactness scores could have different

threat statuses because of differences in their population or range size. We can account for the effect of range size in our comparison, but we cannot account for shifted baselines or absolute population sizes. In addition to these issues, we also cannot assume that the IUCN Red List is a perfect reflection of reality, given that it suffers from outdated assessments and various methodological and data-availability drawbacks (reviewed by Palacio et al.¹²).

With these caveats acknowledged, below we include the methods and results for the additional validation analysis comparing each species' average intactness across its range and its IUCN threat category. We find that IUCN threat category is a significant predictor of intactness, with Critically Endangered species having significantly lower remaining intactness than moderately threatened (Endangered, Vulnerable) and Near Threatened species, which, in turn, have significantly lower remaining intactness than Least Concern species (Fig R1; Fig R2; Table R1 below).

This comparison of two largely independent datasets demonstrates that our approach of functionally grouping species, asking experts to estimate how those groups would be impacted by characteristic land uses, and mapping those estimates based on land uses and species ranges across the region, led to an intactness index that corresponds as expected with an independent assessment of the threat categories of individual species. We have added this validation to the methods and the section of the results entitled "The validity of this biodiversity assessment". We have also included the relevant data and code as supplements to this rebuttal.

Validation test using the IUCN Red List

Methods

The BII algorithm is such that BII for a pixel can be calculated for all species or for a subset of species occurring in that pixel, based on expert-elicited estimates of how different functional groups will be impacted by the land use that occurs in that pixel. The `bii4africa` dataset published in Clements et al.¹ includes a list of vertebrate species within each functional group ('Sp_Vert' sheet), meaning that we can calculate the index for a single vertebrate species in each pixel that falls within its range, and then determine an average intactness estimate across its range.

We ran a Generalised Linear Mixed Effects Model (GLMM; equation A below), with a Gaussian distribution, to assess whether the IUCN Red List threat category ('IUCN_category') of a species (i.e., Critically Endangered, Endangered, Vulnerable, Near Threatened, or Least Concern) was a significant predictor of its intactness across its range ('intactness_index'). We included 4887 sub-Saharan African vertebrate species with a threat category (we excluded 479 data deficient species, and 3 extinct species). We included IUCN threat category as a fixed effect, as well as species' range size to control for species distribution differences (`range_size_scaled`). By including range size, the model isolates the effect of IUCN category on intactness, independently of the influence of range size (with range size being a consideration in the IUCN Red Listing process). The model also accounts for variation in intactness scores across taxonomic classes ('Class') and their nested response groups ('RG'). These nested random effects capture baseline differences in intactness due to taxonomic and functional groupings, ensuring the fixed effects are not biased by class-level or response-group-specific variability.

$$\text{intactness_index} \sim \text{IUCN_category} + \text{range_size_scaled} + (1 \mid \text{Class/RG}) \quad (\text{A})$$

This model was run using the `lme4` package in R. We checked that the response variable and model residuals were normally distributed, and that the two fixed effects were not competing for variance in the global model (VIF close to 1). Range size was scaled and centred to avoid

overdispersion and leverage. We ran an Analysis of Variance (ANOVA) to test whether the fixed effects were significant predictors of intactness, and post-hoc Tukey tests to check for significant differences between IUCN categories (corrected for multiple pairwise comparisons). We have included our data and code as supplementary files.

Results

The IUCN threat category of a species is a significant predictor of its intactness across its sub-Saharan African range ($F = 41.455$, $df = 4$, $p < 0.0001$), with Critically Endangered species having significantly lower intactness than moderately threatened (Endangered, Vulnerable) and Near Threatened species, which, in turn, have significantly lower intactness than Least Concern species (Fig R1 - model predictions; Fig R2 - raw data; Table R1). A species' range size is also a significant predictor of its intactness ($F = 15.715$, $df = 1$, $p < 0.0001$), with larger range species having higher intactness (with IUCN category controlled for). (This is an interesting finding in-and-of-itself, suggesting that smaller range species are more sensitive to land use change).

Figure R1. Predicted Intactness Index (modelled estimates and standard errors) for species within different IUCN threat categories ranked from most to least concern (CR - Critically Endangered, EN - Endangered, VU - Vulnerable, NT - Near Threatened, LC - Least Concern). Different letters indicate significant differences in intactness between IUCN threat categories (see statistics in Table R1).

Figure R2. The Intactness Index of vertebrate species within different IUCN threat categories across their sub-Saharan African range (data means and standard errors shown in red; distribution of data shown by violin plots and individual data points shown in grey). Different letters indicate significant differences in intactness between IUCN categories, accounting for the nested nature of the data and controlling for species range size (see statistics in Table R1).

Table R1. Tukey pairwise comparisons of intactness between species in different IUCN threat categories (CR - Critically Endangered, EN - Endangered, VU - Vulnerable, NT - Near Threatened, LC - Least Concern). *=significant difference

IUCN categories	estimate	SE	df	t.ratio	p.value
CR-EN	-0.045	0.011	4788	-4.164	0.0003*
CR-VU	-0.048	0.011	4799	-4.229	0.0002*
CR-NT	-0.052	0.011	4796	-4.599	<0.0001*
CR-LC	-0.091	0.010	4801	-9.384	<0.0001*
EN-VU	-0.003	0.009	4804	-0.331	0.997
EN-NT	-0.007	0.008	4801	-0.782	0.936
EN-LC	-0.046	0.006	4810	-7.375	<0.0001*
VU-NT	-0.004	0.009	4812	-0.425	0.993
VU-LC	-0.044	0.007	4835	-6.37	<0.0001*
NT-LC	-0.040	0.006	4807	-6.204	<0.0001*

A similar test of internal consistency in assessments by global vs local experts would also be reassuring. For example, I would love to have seen some evaluation of the differences in assessment by more or less local contributors. At least for the more geographically restricted biomes it would be interesting to see if (truly) local experts had different average assessments of change than experts from elsewhere in the continent, vs those based elsewhere in the world. Similarly, it might be interesting to check whether the large national differences in estimated BII are reflected in the average scores of experts from those countries - do, for example, experts from the relatively intact Mozambique on average estimate smaller declines than those from the much more heavily impacted regions of Nigeria or Rwanda? For the method to work well this needs NOT to be the case. These sorts of analysis could not only help us identify the real value of true local knowledge, and reassure us that the expert group did have the detailed knowledge required to make these very large-scale assessments.

RESPONSE: As explained in our response to the reviewer's helpful earlier comment on our inappropriate use of the term 'local knowledge', in this paper we identified experts based on their knowledge of how sub-Saharan species are impacted by land uses characteristic of the region. Since expertise was defined according to relevant place-based knowledge (as opposed to the need for experts to originate from these places), we do not think it is relevant to test whether 'more or less local' contributors differed in their estimates. It would also be difficult to make such a differentiation: country of nationality or residence is not a good proxy since many experts work across multiple countries. Please see our response to Reviewer 3's questions on our expert selection process and the additional information that we will include in the methods to elaborate on the process for selecting experts to promote diversity in place-based expertise. As one important aspect of diversity, we specifically aimed to have a good representation of expertise from across the region, asking experts to note their subregion(s) of expertise (i.e., Central, East, Southern and/or West Africa).

In response to the reviewer's suggestion, we have now tested whether an expert's sub-region of expertise impacted their intactness estimates. As the reviewer mentions, for our approach to work well we should not find sub-region to be a significant predictor of intactness scores for a given functional group of species in a given land use (e.g., experts working in more intact Central Africa should have provided similar scores for near-natural lands as those from less intact West Africa, etc). We tested this by running a GLMM, assessing whether sub-region (as a fixed effect) is a significant predictor of expert's intactness estimates (as a response variable), including taxonomic group as a nested random effect (as explained in the above methods for our IUCN validation), as well as including land use as a random effect (since land use affects intactness but in this case we simply want to control for this effect). We find that sub-region is not a significant predictor of experts' intactness estimates for vertebrates ($F = 1.762$, $df = 4$, $p = 0.14$) or plants ($F = 0.888$, $df = 3$, $p = 0.46$). This result hopefully helps assure the reviewer that this method of contextualised generalisation works. We can include details of this consistency check in the paper.

Our IUCN validation presented above provides additional internal consistency checks, as suggested by the reviewer. If experts' knowledge was indeed only locally applicable, and not generalisable across the region, then we would expect that our approach of contextualised generalisation would have resulted in inaccurate estimates of the intactness of species within their regional range. By contrast, we show that these estimates consistently relate as expected with species' IUCN threat categories. Notably, we find this relationship holds when controlling for species' range size, meaning it holds for both wide-ranging species and very 'local' (small range) species that few (if any) experts would have had specific place-based knowledge of. This consistency suggests that our approach of contextualised generalisation led to robust estimates. We flag this in the new validation results in the paper as follows "*Notably, this trend holds for*

both large- and small-range species, suggesting our approach led to robust outcomes even for very localised species that are likely to fall outside of most experts' specific place-based knowledge. This consistency suggests that our approach of contextualised generalisation led to robust estimates". We hope that this validation will reassure the reviewer that the process we used was robust to variations in the expert group's extent and diversity of knowledge, as we only used estimates that linked directly to their expertise and experience.

We can also add further information on the logic of the approach to the paper methods to explain why we expect, and indeed observe, this consistency in estimates between experts from different sub-regions and for both wide-ranging and localised species. For our approach to be practical, it cannot only apply local knowledge to those specific locations, as that would require potentially millions of experts to represent sub-Saharan Africa's 20 million km². Rather, we made use of a published functional grouping approach¹¹ to estimate how different types of species will be impacted by different types of land uses within the region. Experts have place-based knowledge of particular species and land uses, and we assume that their knowledge applies beyond the specific local contexts that they know, since a functional group of species is expected to respond in the same way to the same land uses across the wider region where that group occurs, i.e., the idea of contextualised generalisation. A critical aspect for this to work well is to ensure that land uses are well differentiated, and land uses that have different effects are not lumped together into one category. Different experts should then be able to provide consistent estimates for these land uses, whether they hold 'very local' knowledge (e.g., for just one example of species in a given functional group being impacted by a given land use), or more regional knowledge (e.g., from working across different species within a functional group and across different examples of a characteristic land use across the region). Similarly, estimates for functional group responses to specific land uses should be consistent between experts with knowledge from different areas. Even low intactness countries (e.g., Rwanda) maintain some high intactness land uses (e.g., Akagera National Park, Nyungwe forest), although these will make up a much smaller proportion of the country's total area than in highly intact countries. We therefore expect that experts with place-based knowledge from low intactness countries will be familiar with a range of land use classes, and able to estimate intactness for each. For example, Rwandan vs Mozambican experts should be able to consistently estimate impacts of dense urban areas vs near-natural lands on the populations of aerial feeding birds. The difference in BII between these countries lies in the relative extent of land use types and intensities, which is calculated from the mapping process. Experts only scored for the land uses (and species functional groups) they knew, so if a given land use was not present in the area an expert knew, they would have not provided an estimate. Our replacement of 'local' terminology with 'place-based' and the addition of details on what is meant by this in the revised introduction, along with details on how it was implemented in the methods section hopefully clarifies this approach of contextualised generalisation.

Without total confidence in the underlying evaluation, the rest of the analyses are harder to review. I do think they are a sensible way of cutting the data up for analysis, and (provided the underlying data are trustworthy) the justifications and conclusions are robust with most analyses adequately explained and apparently sensibly undertaken. The only real question I raised here concerns the figures in Extended Data Fig 2, where I struggled to understand why there are such obvious lines in the raw data - presumably reflecting groups of one land use, or biome, or taxonomic group? This could do with explanation and these figures could be tidier in order to allow us to interpret and know whether this artifact is a problem or not.

RESPONSE: We are pleased that the reviewer thinks our analyses are sensible and our justifications and conclusions are robust, assuming the underlying data is reliable (which we hope has been established through the extra validation analyses and explanations above). Regarding

Extended Data Figure 2, these lines reflect ecoregional differences in species composition. Species composition influences what subset of expert-elicited intactness scores contribute to the BII in a given pixel, and these compositions are determined at ecoregion level (see Methods: Species richness across regions). We have converted these scatterplots into boxplots to make them tidier and easier to detect differences in BII, also based on Reviewer 2's comment that the BII values we report in the text for the lowest vs highest land use intensities are not clear on the figure. Please see new draft plots in response to Reviewer 2 below.

Referee #1 (Remarks on code availability):

Will happily do it if they come back with sensible further analyses convincing me the data are worth looking at...

RESPONSE: Thank you, we would be very happy to get feedback on this. The feedback above on the analysis has been extremely helpful in clarifying and validating our approach.

Referee #2 (Remarks to the Author):

General Comments:

Clements et al. really did tremendous work on “The biodiversity intactness of Africa assessed through local knowledge” with about 200 experts to address the current knowledge gap on less-known and rarely well-represented sub-Saharan biodiversity in global biodiversity maps and data. Hence, the study is timely, crucial for biodiversity conservation in sub-Saharan Africa, and worth publishing in this journal. The data presented, which was used in estimating the biodiversity intactness index, are original and were collected through a clear and reproducible expert elicitation method. Further, the methods, estimates, and statistics in the main manuscript, supplementary, R scripts and the previously published bii4africa dataset paper are very clear enough and reproducible.

RESPONSE: Many thanks for the positive feedback and for recognizing the timeliness of this study and its crucial importance for biodiversity conservation.

However, there are some concerns that should be addressed. First, given the disproportionately low representation of experts that are resident in nearly all the sub-Saharan Africa countries (except South Africa, as shown by the affiliations of the experts/authors on the published bii4africa dataset paper and website), I have concerns that the non-resident experts may not really know much about the history or current intactness of biodiversity in each country, ecoregion, or species group they represent during BII scoring, thereby weakening the “bottom-up” approach. For instance, an expert that had a long-term study or lived in a country over several years (like 10 years or more) but has long left the country or ecoregion or stopped working there for say 10 years or more before this data collection may overestimate or underestimate the BII. Hence, authors should add texts on how they, to a large extent, accurately identify the country/region of experience of every expert to avoid underestimating or overestimating BIIs of most countries (other than South Africa that had several people). Or authors, should add caveats to the manuscript about the challenges faced or potential weaknesses of the expert recruitment method despite their efforts to mitigate such challenges, as they did in Lines 403-406 and 635-636 for limited access to data from protected areas.

RESPONSE: Please see our above response to Reviewer 1's comment starting “*A similar test of internal consistency in assessments by global vs local experts would also be reassuring*”, which speaks to this concern. Expertise was defined according to relevant place-based knowledge (as opposed to the need for experts to originate from these places or currently live in them). We

focused on recruiting experts that were actively working in the region wherever this was possible. Many experts also worked across multiple places/countries. Experts were asked to estimate how functional groups of species are impacted by characteristic land uses, using their place-based knowledge. We then translated those estimates into a map based on the current distribution of such land uses. Thus, the experts themselves were not asked to consider how prevalent a given land use was; this was determined through mapping. It was therefore less of a concern if an expert had not worked in an area for some time, provided they had in-depth knowledge of the region, since how different land use types affect different types of species is relatively stable over time. We have clarified the process and criteria for selection of experts in the methods section. As the reviewer suggests, we will also add a caveats section to the manuscript to ensure limitations are acknowledged and their implications considered, including reflections on the limitations of the expert selection process.

Please also see the additional validation we performed as suggested by Reviewer 1 above, which demonstrates strong congruence between our expert-based estimates of intactness for 4887 vertebrate species across their sub-Saharan African ranges, and their IUCN threat statuses. This finding suggests that our expert-estimation approach was robust.

Our response to Reviewer 3's comment starting "*The lead experts were also asked to identify subsequent experts*" addresses our efforts to include experts from across the region, and the challenges that we encountered. We provide examples for birds and primates.

Second, I do strongly believe that the title can only explicitly include the "local knowledge" if the intactness scoring truly involved the "local non-expert residents or citizens" of sub-Saharan Africa" that live and depend on the ecosystems that house these species. The traditional ecological knowledge of locals are often passed from generation to generation and it would have added more values to the study. So, given that "non-expert locals" were missing in the intactness scoring (unless I missed it), I suggest that the title be revised to: "The biodiversity intactness of Africa assessed through experts' local knowledge"

RESPONSE: In light of this comment, together with Reviewer 1's second comment (please see our detailed response to that comment), Reviewer 3's request that we specify 'sub-Saharan', and in keeping to 75 characters, we have changed the title to "*Sub-Saharan Africa's biodiversity intactness using place-based knowledge*". We have also changed our use of the term 'local knowledge' to 'place-based knowledge' throughout the paper as outlined in the responses to Reviewer 1.

Generally, all aspects of the manuscript are well written except for the necessary edits, some rearrangements (and suggested analyses) in the result section, and, more importantly, the improvement of the discussion section that I mention in the specific comments below. I therefore recommend the manuscript for publication after the revisions I suggested here and below.

RESPONSE: We thank the reviewer for their positive feedback. We address the specific comments below.

Specific comments:

Line 55: rewrite as "Grasslands and fynbos (mediterranean-type ecosystems) suffered the greatest declines, mostly from land transformation into cropland, while deserts and shrublands have been least impacted."

RESPONSE: We have made this small suggested edit

Line 61: “decision-makers”

RESPONSE: We have made this small suggested edit

Line 62: Two heavy words there. Replace “credible, legitimate” with “real”

RESPONSE: We have replaced these two words with ‘appropriate’ rather than ‘real’

Line 70: replace “it” with “biodiversity”

RESPONSE: We have made this small suggested edit

Line 72: delete “to”

RESPONSE: We have made this small suggested edit

Line 85: Before the IPBES sentence, please add one more sentence here, probably as an example, to shed more light on the sentence in Line 82-85.

RESPONSE: We have added the following example: ‘For example, models estimating mean species abundance across the globe rely on extrapolation from patchy data that underrepresents data-poor bioregions, taxa, and threats¹³’

Line 89: “We apply this approach to assess the biodiversity intactness of ecosystems in sub-Saharan Africa, ...”

RESPONSE: We have made this small suggested edit

Line 107: how about replacing “sub-Saharan Africa’s” with “African”?

RESPONSE: We think ‘Africa’s’ works better than ‘African’, but have removed ‘sub-Saharan’ as suggested

Line 109: how about replacing “Africa’s” with “African”?

RESPONSE: We have made this small suggested edit

Line 109: be specific with the “remaining vegetation”, please. Were you referring to the “vegetation in other continents” or “other vegetation types in Africa”?

RESPONSE: We have added ‘for these rangelands’ to clarify

Line 121-122: “the impact of sub-Saharan Africa’s predominant land uses and their intensities on ~50,000 species of terrestrial vertebrates and vascular plants.”

RESPONSE: We have made this small suggested edit

Line 124: “experts’ estimates...”

RESPONSE: We have made this small suggested edit

Line 125, 151 and other similar places like the text in Fig. 1, title of Fig. 3: Wherever you refer to regions within sub-Saharan Africa, which you already regarded as a region from the abstract, please consistently write “ecoregions(s)” or “biome(s)” accordingly not the ambiguous “region(s)”. For example, in Line 151, replace “vegetation types or regions” with “biomes or ecoregions”. Doing this align all texts and sections of the paper with the terms in Figures (both main and extended) and in results.

RESPONSE: Thank you for this suggestion. We have amended throughout to be more specific: sometimes we are referring to biogeographical areas, or political areas (e.g. countries), or ecoregions, and have now specified in each instance. We have left the term ‘region’ only when referring to sub-Saharan Africa which, as the reviewer says, we define as the region in the abstract.

Line 129: delete “that”

RESPONSE: We have made this small suggested edit

Line 141: is there a mistake in “magnitude more experts”?

RESPONSE: There is not a mistake, but we have rephrased to make meaning clearer: “by including over ten times the original number of experts (200 versus 16)”

Line 161-163: This is true, as Africans engage in hunting and consume those animals as “bush meat”. It would be good that you mention this in your discussion, and what can be done (as replacement though) to stop this hunting of non-domesticated animals in the “wild” for food.

RESPONSE: We plan to add this to the new discussion paragraphs reflecting on key findings (see our response to the reviewer’s later comment on the discussion section)

Figure 1: A very cool and interesting presentation here. Please edit the “Low-High” under the “Intensity” in the land use class legend to “Low  High” in Figure 1a: i.e., use arrow instead of hyphen, and let the “Low” and “High” be at the extreme left and right part of the box, respectively.

RESPONSE: We are glad to hear the reviewer likes the figure! We have implemented the suggested change.

In the third to the last line on the first page of “Scientific Data File”: “unavailability” should replace “availability”

RESPONSE: We are unfortunately not sure what page the reviewer is referring to here and would appreciate clarification.

Line 195: I think the bii4africa dataset should be available as soon as the paper is accepted (by this journal or elsewhere) for reproducibility and further studies for interested.

RESPONSE: The data are now published open-access in a paper in *Scientific Data* (Clements et al.¹ <https://www.nature.com/articles/s41597-023-02832-6>), and the reference has been updated to reflect this. The BII map will be made available open access as well once the current paper is published.

Line 198: add “(Fig. 2a)” just after the word “intactness”.

RESPONSE: We have made this small suggested edit

Line 198: since the paragraph in Line 211-222 show the relative impacts of the predominant land uses in sub-Saharan Africa, please add either “absolute”, “independent” or “individual” in between “Considering the ...” and “... impact of the region’s predominant land uses on biodiversity intactness, we” in Line 198.

RESPONSE: We have added ‘absolute’ as suggested

Line 199: do you really need to add “strictly” here?

RESPONSE: Yes we think we do need this adjective, since only strictly protected areas are included in this land use category (IUCN PA categories I-III); please see explanation in Appendix 2.

Line 203-204: Consider replacing this sentence (i.e., “Depending on species composition and land use intensity, the BII of a given land use pixel varies (Fig. 2a)”) with the following more direct sentence: “Our estimates also show that the BII of each of the major species groups varies between the six land use classes (Fig. 2b).”

RESPONSE: In this sentence we were aiming to explain that there is of course variation within these land use classes (as seen by the spread of data in the boxplots in Fig. 2a), and that this is

caused by variation in species composition and land use intensity. We have rephrased the sentence to try make this point more clearly: *“The variation in BII scores within these landscapes (Fig. 2a) is caused by variation in both species composition and land use intensity across pixels”*

Line 207: with the Fig.2b results/boxplots, the “while the opposite holds true for reptiles” appears not really a true opposite of the first part of the sentence. Please check it and reframe for clarity.

RESPONSE: We have rephrased to *“Plants tend to have retained higher intactness in croplands compared with tree croplands, while reptiles tend to have retained higher intactness in tree croplands than croplands (Fig. 2b).”*

Line 212: “its total extent”. Please use exact word to replace “its”. It makes the sentence ambiguous.

RESPONSE: *“its total extent”* replaced with *“the total extent of that land use”*

Line 211-222: While it is very nice that the BII results, which are the main focus of this study, occupies the 1st paragraph of the results section, Fig. 1a only shows land use intensity without the BII numbers quoted in Lines 214, 217 and 219 and it is not really related to the story in Fig. 1b&c and paragraphs 1&2 of the results section. Hence, for a better presentation and easy-to-follow flow of the results, please detach Fig. 1a from Fig. 1 and make it Fig. 2a; create a new Fig. (a box plot like the current Fig. 2a as Fig. 2b) to show the “relative” proportion of the remaining BII value (76%) that each of the 6 classes contribute. The new fig. 2b will further explain the new Fig. 2a (former 1a). With that adjustment, the current Fig. 2a becomes 2c, current 2b becomes 2d. And more importantly, please let the whole paragraph of Line 211-222 come before that of 198-209, which would show that you move from general to specific estimates.

RESPONSE: We included the land use map in Figure 1 because we had intended the figure to broadly visualise the approach we took (since in Nature the methods section only comes at the end of the paper). But we also agree with your points about the readability and link to the text and will make the suggested changes to the figures and the order of the results.

Line 231: Move this sub-topic (Land use intensity impacts intactness) to the line preceding the contents in Line 211-222.

RESPONSE: We have made this small suggested edit

Line 233-234: please write “(Fig. 2a)” (i.e., the current Fig. 1a which I suggested above that you should make 2a) after the phrase “Considering the two most extensive land uses (croplands and unprotected untransformed lands)”.

RESPONSE: We have made this small suggested edit

Line 235: please add “two” before “land uses”

RESPONSE: We have made this small suggested edit

Line 236-239: This isn’t clear- something is missing in Line 238. Seems you wanted to add “compared with 54% BII in” before the word “smallholder”? If choose my guess, remember to remove the (54%) at the of the sentence.

RESPONSE: We have rephrased to *“which is notably less than in lowest intensity croplands (smallholder agriculture in small fields with limited inputs, and patches of remaining vegetation) where BII is 54%”*.

Line 236-244: Please consider adding extended figure(s) on the percentages quoted here. Also, consider reporting the “relative” percentages (not the absolute ones) such that the BII comparison between the land use intensities within each of the two land uses adds up to 100, as the title of extended figure 2 and histogram imply.

RESPONSE: These percentages refer to the average BII values of the lower and upper ends of the land use intensity continuum plotted in Extended Figure 2 (i.e., the BII values in pixels with land use intensities of 0 to 0.1 and 0.9 to 1, respectively). Based on the reviewer’s comment that this is unclear, we have converted these scatterplots into boxplots (see draft plots below) so that the different BII values we report in the text are clearer on the figure. We have rephrased the legend accordingly and converted the histogram into relative percentages as suggested.

Extended Data Fig. 2. The relative intensity of land use across sub-Saharan Africa and its influence on the Biodiversity Intactness Index (BII). The two most extensive land uses (a) untransformed lands and (b) croplands are shown. Histograms (top) show the relative number of pixels characterised by different land use intensities, reflecting the skew towards lower intensity land use in the region’s untransformed lands and croplands. Boxplots (bottom) show the distribution of BII values for pixels characterised by different land use intensities from lowest intensity (0 to 0.1) to highest intensity (0.9 to 1).

Line 253: “Differences in intactness between countries, biomes, and ecoregions”

RESPONSE: We have made this small suggested edit

Line 254-257: And please make a sentence about those remaining 15 countries with 70-80% BII too.

RESPONSE: We have added the following sentence: *“The remaining 15 countries have retained intermediate levels of BII (70-80%), with Ethiopia being middle of the range with a BII of 73%.”*

Given that those with less than <70% BII are relatively small in number, consider dwelling more on the likely drivers of such critically low BII score in the discussion section (and possibly with some supplementary data analysis of countries like Nigeria and Rwanda that would nail such huge biodiversity loss and trigger government and NGO’s attention).

RESPONSE: We will add this to the discussion as suggested, please see our more detailed response to the reviewer’s later comment on the discussion section.

Line 260-261: please rewrite as “Biodiversity intactness across (a) countries, (b) biomes, and (c) ecoregions30 of sub-Saharan Africa. Average BII scores are shown in each map.”

RESPONSE: We have made this small suggested edit

Given that the ecoregions are many and you couldn't insert them in Fig. 3c, I suggest you delete Fig. 3c from the manuscript, and make the "Extended Data Fig. 4" to be the main Fig. 3c.

RESPONSE: Great suggestion, thank you! We have implemented this change.

Figure 3b: is there any biome particularly called "Mosaic"?

RESPONSE: There is no biome called 'Mosaic'; this term refers to areas that are a mosaic of two biomes. We have revised this figure to rather label those areas according to the specific mosaics that they represent (e.g., 'Humid savanna - forest mosaic') rather than simply labelling them all 'mosaic'. We have also stated in the figure legend that (b) shows "Biomes and biome mosaics".

Line 266: "remaining BII are made by each biome in unprotected untransformed lands: ..."

RESPONSE: We have made this small suggested edit

Line 268-269: "The largest contributions of desert and fynbos to the remaining BII are in protected lands" Not so sure if my suggested revisions in Line 266, 268-269 are so clear, but I just want the authors to use direct, unambiguous sentences to convey Figure 4 as a follow-up to 3b in Line 265-269.

RESPONSE: We have made this small suggested edit

Line 278-279: "likely because crop production is generally unviable in these biomes (Fig. 4)." - so lack of cultivation could drive such poor BII in deserts and shrublands? I think that is a very unlikely speculation, unless you have evidences.

RESPONSE: We did not mean to speculate, but rather to point out that cropland extent was limited in these arid biomes. We have reworded to improve clarity: "In the more arid biomes where cropland extent is limited, the greatest losses of total biodiversity intactness occur across near-natural lands (desert and shrubland biomes) and rangelands (shrubland biome)"

Line 280: list the grassy biomes in parentheses just after the word grassy biomes.

RESPONSE: We have added: "(grassland, acacia savanna and humid savanna)"

Line 283-284: this ", together with deforestation for rangelands, croplands and timber plantations" does not complement to the preceding clause in the sentence.

RESPONSE: We have deleted this text and replaced with a standalone sentence: "Deforestation for rangelands, croplands and timber plantations also contribute to losses in the forest biome"

Line 285: list the five biomes in parentheses for clarity

RESPONSE: We have made this small suggested edit

Line 294: see the comment on this Fig. 3c above

RESPONSE: We have addressed this in our earlier response (we have replaced Fig. 3c with Extended Data Fig. 4)

Line 310: delete "species"

RESPONSE: We have made this small suggested edit

Line 317: "make use of here" "use here"

RESPONSE: We have made this small suggested edit

Line 319-320: Not so clear.

RESPONSE: We have revised this sentence to clarify: "In contrast to a decontextualised modelling approach to estimating BII, our approach was based on a structured expert elicitation that

included a step where experts came together to critically discuss the validity of anonymised, aggregated estimates of intactness for different groups of species and thereafter (anonymously) revise their independent estimates where they deemed it necessary. A form of validity checking is therefore embedded in the process of producing this large-scale map."

Line 333: do you really meant to write "average mean"?

RESPONSE: This repetition was a mistake, we have removed 'mean'

Line 349: Consider adding comma sign before and after the "for example"

RESPONSE: We have made this small suggested edit

Line 352-355: This information appears not related with the preceding sentences and the story on "The validity of this biodiversity assessment" and Extended Fig. 6a-c. Maybe you need to create a premise for it or relocate it to a more pleasing spot in this paragraph.

RESPONSE: We have added the following premise: "Another criticism of the globally modelled BII was that biodiversity hotspots (priority areas of exceptional endemism that have lost $\geq 70\%$ of their primary vegetation) had unexpectedly high scores compared with non-hotspots. By contrast, ..."

Line 367: why don't you just say "200 experts" instead of "several hundred"?

RESPONSE: We have made this small suggested edit

Line 373: it would be nice that you briefly recap here that the remaining overall BII of sub-Saharan Africa is 76%.

RESPONSE: We have added the following sentence: "Sub-Saharan Africa has lost just under a quarter of its pre-industrial biodiversity intactness."

Line 375: Based on my comment on Fig. 1 and on Line 211-222, the referenced "(Fig. 1) may not be Fig. 1 (alone or again).

RESPONSE: We will ensure all figure references are amended accordingly

Line 375-377: Absolutely! That's very important.

RESPONSE: We are pleased that the reviewer agrees with this key takeaway message of the paper!

Discussion section: Despite that the results section was full of exhaustive and interrelated data analyses and that the discussion section should mirror the structure of the results section in flow for ease of understanding and linkage of the findings, it is strange that about 80% or more of the discussion section (especially from 412-439) repetitively (in multiple, different sentences) only summarily speculate (without data comparisons between the current findings and the previous) the potential benefits of the approaches and data used for this study. Hence, a vast portion of the intriguing findings in the study was left undiscussed — especially in relation with existing studies' reported BII values for sub-Saharan Africa; the potential (literature-supported) drivers of the BII values reported for across and within biomes, ecoregions, and countries in sub-Saharan Africa; whether there were over estimation or underestimation, and why this study potentially provide a more accurate BIIs etc. For example, authors should discuss the underlying drivers of the huge ($\geq \sim 50\%$) decline in BII in West Africa and South-eastern Africa countries.

RESPONSE: Thank you for this helpful suggestion, we agree we have missed an opportunity to dig deeper into some of the results. We plan to add two paragraphs to the discussion to address this comment.

In the first new paragraph we will reflect on the similarities and differences between this study and the only two previously published BII maps: one done for southern Africa only and another

done for the globe. Notably, we will reflect that our BII shows some declines in southern Africa compared to the map produced for the region two decades ago (which is to be expected). By contrast, our map shows quite markedly different results to the global BII, particularly in the more open/arid ecosystems where the global map shows BII to be unusually low compared with our map, the literature and other human pressure maps. We suggest that including place-based knowledge has produced a more accurate large-scale map than currently available and represents a significant improvement on the current global map for the African region in particular.

In the second additional paragraph we will discuss where the most notable losses have occurred and why (e.g., Nigeria - extensive croplands having large impact on forest-dependent biodiversity; Rwanda - extensive eucalypt plantations interspersed with smallholder croplands). We will also reflect on surprising findings (e.g., relatively high BII in Mozambique, Angola, South Sudan and Somalia despite histories of extended conflict, possibly because conflict depletes larger, well-studied species but actually insulates the majority of plants and smaller vertebrates from large-scale land use change, though acknowledging context-specific complexities of conflict's impacts on biodiversity).

To make space for these new paragraphs, we plan to reduce the current discussion paragraphs 5 and 6 into one paragraph. This reduction will eliminate repetition as suggested by the reviewer.

Line 410: "megfaunal" -> "megafaunal".

RESPONSE: It is 'megafaunal' in the manuscript

Line 465-466: "in such cases" or "in all cases"? I looked through the methods but couldn't find if any other BII formula apart from Equation 1 exist for cases without data scarcity. Reading through 499-505, I later found that the "in such cases" only apply to other than large mammals species. Please show and illustrate the equation for the "ideal situation" first (i.e., the situation you expressed in Line 502-505) before the "data scarcity situation", which the current equation 1 displays. If I am wrong, please make the texts here clearer for readers like me.

RESPONSE: We now clarify by first showing the basic BII formula (1), applicable in all cases, and then two equations for the pixel-level BII scores that feed into that equation: (2) the 'ideal situation' (not applicable here) and (3) the data-scarcity situation, applicable here. We also make note of the large mammal variation for equation 3:

"The BII for a particular area (e.g., continent, ecoregion, biome or country) is calculated by averaging the BII scores for each unit of land ('pixel'), such that each pixel counts equally.

$$BII = \frac{\sum BII_{pixel}}{n} \quad (1)$$

The BII for a unit of land (BII_{pixel}) is calculated by averaging the intactness scores I for each present species s under land use activities u such that each species counts equally:

$$BII_{pixel} = \frac{\sum I_{su}}{n} * 100 \quad (2)$$

The index can accommodate data scarcity—in the absence of intactness scores per species per pixel, the BII can use intactness scores that represent groups of species that are expected to respond similarly to human impacts ('response groups' or 'functional groups'); categories of human land use activities (as opposed to a continuous scale); and species richness information for the broader area (e.g., ecoregion). In such cases, the BII for a pixel is determined as follows:

$$BII_{pixel} = \frac{\sum_i R_i I_{ik}}{\sum_i R_i} * 100 \quad (3)$$

where I is the intactness score for species group i under land use activities k , weighted by the richness R of species group i (number of species or proportion of total species) in the relevant area (commonly, the ecoregion).

BII can then be averaged across pixels in an area of interest to provide a single average BII score for that area that accounts for the composition of both species and land uses (equation 1). BII scores can be calculated for all species or a subset (e.g., vertebrates, amphibians, etc). Here we make use of equations 1 and 3, to accommodate data scarcity. As in the original BII approach, we work at a base pixel scale of 1 km²."

Line 465,469: "region (commonly, the ecoregion)"- please directly and simply say "ecoregion" whenever you meant "ecoregion" and limit the use of "region" to "sub-Saharan Africa" throughout the manuscript.

RESPONSE: We have implemented this suggestion throughout (see our above response to the reviewer's similar comment)

Line 512: how finer can a patch or scale would be for it to be regarded as "finer"? Please state the specific spatial scale that was regarded as being finer.

RESPONSE: We have added "(i.e., less than a square kilometre)"

Line 525: simply replace "biogeographical vegetation types" with "biomes" as you did name them in next sentence and in Fig. 3b.

RESPONSE: We have implemented this small suggested edit

Line 544: replace "regions" with "ecoregions and biomes"

RESPONSE: We have implemented this small suggested edit

Line 603: Please cite the "Extended Data Table 2" after the nitrogen input inside the parentheses.

RESPONSE: We have implemented this small suggested edit

Line 617, 618, 619: "herbivore biomass capacity" and "herbivore biomass" are not that correct to be used as terminologies. You meant "herbaceous biomass", right? If yes, in Line 617, change "herbivore biomass capacity" to "herbaceous biomass (i.e., forage for herbivores)"; in Line 618, change "sustainable herbivore biomass" to "biomass declines", and in Line 619 just say "biomass". Replace ". In addition" with "and" in Line 618 to merge the 2 sentences.

RESPONSE: We did not mean 'herbaceous biomass', we meant 'potential African herbivore biomass'. We have rephrased to make this clearer: "A unimodal relationship has been shown between potential African mammal herbivore biomass and mean annual rainfall (MAR), peaking at ~1700 kg/km² and ~700 mm MAR¹⁴. In addition, potential African mammal herbivore biomass is higher in areas with higher nutrient soils¹⁵."

Line 619: Please add "Hence," before "we clustered..."

RESPONSE: We have implemented this small suggested edit

Line 637: The latter were considered ...

RESPONSE: We have implemented this small suggested edit

Line 642: use multiplication sign for dimensions not x

RESPONSE: We have implemented this small suggested edit

Line 650: "sub-Saharan Africa region"

RESPONSE: We have implemented this small suggested edit

Line 649-650: Think this

RESPONSE: It is unfortunately unclear to us what is being suggested here

Line 656: replace # with No. throughout the table

RESPONSE: We have implemented this small suggested edit

Line 681: change x to multiplication sign

RESPONSE: We have implemented this small suggested edit

Line 700-710: Not sure this is different from the preceding sentence. Moreso, “reasonable correlation” is subjective.

RESPONSE: We have merged these sentences and removed the comment about ‘reasonable correlation’, which is not necessary: “However, we would expect that areas with a higher BII to have a lower HFI, lower BMI and higher BHI, and vice versa.”

Line 703: Extended Data Fig. 6. shows “across ecoregions” correlations not “per ecoregion”

RESPONSE: We have replaced ‘per’ with ‘across’ as suggested

Line 826: But there are references mentioned in the Methods with serial numbers before this 51st reference.

RESPONSE: We will ensure all reference numbers are accurate

Referee #2 (Remarks on code availability):

Yes, the code provided via the <https://figshare.com/s/00168eb685728c0be9d3>, guided by a self-explanatory code lines and README run well.

RESPONSE: Thank you for the positive feedback.

Referee #3 (Remarks to the Author):

Thank you for your submitted article, which involves providing biodiversity indices for policy in contexts of data paucity. Specifically, the use of expert input to evaluate sub-Saharan Africa’s biodiversity intactness. Since my expertise is in expert elicitation and sustainable development, I will focus my review of those aspects (methods and claims made). This also corresponds with what the editor asked me to focus on.

RESPONSE: We thank the reviewer for their expert input in flagging the key aspects of our elicitation that require elaboration, justification and reflection on limitations.

First, I should point out that my initial submitted review did not account for the fact that most of the expert elicitation is presented in an accompanying article in Scientific Data. I am not used to having methods described in accompanying articles. Personally I lament that some of the most widely read journals tend to de-emphasize the presentation of methods (such as at the end of articles), but regardless I would recommend that the authors more clearly identify that the elicitation methods can be found in the accompanying article. When I read this piece it read to me as if the authors point to the other article as a different study using a similar methodology.

RESPONSE: Thank you for the feedback on the confusion you experienced. Based on this feedback, we have made it clearer that the expert-elicited dataset that we use has been described in a

stand-alone published paper (Clements et al.¹). Specifically, we have amended the fifth paragraph of the introduction to start *“Our novel approach overcomes these limitations of top-down biodiversity models by quantifying the Biodiversity Intactness Index (BII)¹¹ using the ‘bii4africa’ expert-elicited dataset that we previously published based on the knowledge of 200 experts in African fauna and flora¹”*. In the methods we state that: *“...we made use of a published dataset¹ of intactness scores that were estimated as part of our ‘Biodiversity Intactness Index for Africa’ project (<https://bii4africa.org/>) that involved a structured expert elicitation process involving 200 experts in African flora and fauna”*. We have also included an additional sentence at the end of this paragraph stating that *“A detailed description of our expert elicitation approach and resulting bii4africa dataset can be found in a previously published paper, which recognises the contributing experts as co-authors. Here we provide a brief summary of the key elements of the expert elicitation.”*

We decided to publish the expert elicitation methods and intactness score dataset as a standalone data paper because (1) we felt it was important that the contributions of all experts to this ‘bii4africa’ dataset were fairly recognised through co-authorship; and (2) the dataset has many uses beyond mapping BII (as summarised in Table 3 of the paper: <https://www.nature.com/articles/s41597-023-02832-6/tables/3>). Our intention was not to de-emphasize the elicitation method but rather the opposite: to give it full attention in its own paper so that there would be sufficient space to describe the method in detail. However, we take note of the fact that a more detailed summary is clearly needed in the current paper, and hope our additions address that shortcoming.

In fact, I believe that since expert elicitation is so prominent in the presentation of this study, the authors should provide much more details of the methods in the article itself (or at least in the supplement).

RESPONSE: We have expanded the ‘Intactness scores’ section of the methods to provide more details on the elicitation process, aimed particularly at addressing the specific comments laid out below and by the other reviewers. We will also add a Supplementary Methods document that provides an example of the process, as requested by Reviewer 1.

I think the topic is a very important and difficult effort, especially to do well. However, I have some concerns about the elicitation done, and also the framing of the piece in relation to development.

RESPONSE: Thank you for the positive feedback. We have addressed your specific concerns on the elicitation and framing in our point-by-point responses below.

Expert elicitation, when done credibly, is a very meticulous process of identifying expertise and regulating the kinds of information and intuition you want to elicit while keeping the various biases and cognitive issues that can degrade data quality at bay. You also want to make sure that information acquired from experts reflects the expertise you seek, and is not ultimately controlled and shaped by the analysts that ask the questions and synthesize the results. I think at broad levels the authors adopt a good structure, but I have some reservations in the specifics in the identification, training and structure of the elicitation process. I also provided important references (which are a brief selection of articles in entire literatures on the topics).

RESPONSE: We adopted the IDEA protocol for structured expert elicitation because it is a published approach¹⁶ that draws on a large body of literature on expert elicitation (including many of the authors and useful references that the reviewer shares with us below), aimed at mitigating potential shortcomings of expert elicitation. We have added text to the ‘Intactness scores’ section

of the methods to clarify the use of this protocol and the reasons for selecting it. We are pleased to hear that the reviewer thinks we adopted a good expert elicitation structure, and we respond to the specific reservations below.

I appreciate the description of experts chosen for the elicitation. When I first read the paper boast 200 experts my immediate reaction was “that’s too many to handle effectively”. However, I think the substructuring of expert teams into groups of 20 is well done (though I would say 20 is on the high end of good elicitation methods). The authors also demonstrate relevant information showing some dimensions of variation in experts involved in the process.

RESPONSE: We are pleased to hear that the reviewer thinks our sub structuring of experts into smaller groups was a good approach.

However, I have some hesitation with the treatment of “lead experts” and their role in the further selection of experts I’m hoping the authors can clarify, expand on, or reflect on in their draft. Selection of lead experts is vaguely, “identified based on their relevant expertise”. I will note that expert status (as determined by credentials and markers of prestige) often does not have any bearing on performance or accuracy of expert input, and can even be counterproductive.

RESPONSE: We agree that status alone is not a good way to identify experts, as reflected by our statement in Clements et al.¹, which we have now included in this paper’s methods section as well: “A broad definition of expertise was used to identify experts, centred on experience of how sub-Saharan species are impacted by human land uses. Diverse types of people can have such experience (e.g., researchers, field or tour guides, park rangers, conservation practitioners, museum curators, and consultants), and inclusion was thus not limited to specific qualifications or institutional affiliations.” Lead experts needed to meet this definition of relevant expertise (which was not based on prestige), as well as being willing to take on this task and having access to an existing network of experts, or else the willingness to develop such a network. We now state the following in the methods section: “An expert in each broad taxonomic group was invited to lead the elicitation for that taxonomic group. Each ‘lead expert’ was identified based on their relevant expertise (meeting the above definition), existing network across the continent or willingness to develop such a network, and willingness to take on the task of assisting with the identification and recruitment of an appropriate diversity of additional experts.” Such a network was important given the challenging task of identifying experts across the continent and convincing them to give of their time to participate. For example, several lead experts serve in the IUCN Species Survival Commission working groups and other regional networks (e.g., <https://ascaris.org/>; <https://www.birdlife.org/our-partners-africa/>), noting that these are not all exclusive scientist networks (e.g., Birdlife partners).

The lead experts were also asked to identify subsequent experts. This kind of snowball approach to identifying experts is not necessarily problematic but can introduce issues of similarity biases (“groupthink”). Notably, focusing on experts from scientific initiatives is likely to leave out experts with local knowledge not often valorized in official and scientific working groups. Did the authors do any calibration to see if they collected a sufficiently broad range of expertise? The total number of experts may not represent knowledge breadth if experts are from similar training and collegial working relationships.

RESPONSE: We put significant effort into including a diverse group of experts for each taxonomic group. The experts did not need to be trained in a certain way to meet our definition of an expert, and lead experts were briefed to find and reach out to people that met the definition of expertise quoted above, focusing on diversity of expertise geographically, taxonomically, professionally, as

well as aiming to include as many African experts as possible (these different dimensions of diversity within the expert group are depicted in Clements et al.¹ Figure 2 which we have also inserted below for easy reference). We also explain in Clements et al.¹ that *“The lead expert identified individuals known to have relevant expertise. If this activity did not achieve the target of 20 individuals, additional experts were identified through relevant publications (using appropriate search terms on Google Scholar) and websites (e.g., specialist nature guides or tours, conservation organisations)”* and that *“In some cases, participating experts were asked to recommend other experts (snowball sampling)”* when the lead did not manage to recruit a sufficient number of participants or experts noted in the introductory phase of the process that key dimensions of diversity were missing. We needed to balance the need for multiple dimensions of diversity with the need for an appropriate number of experts to ensure each elicitation worked well (as the reviewer mentions, even 20 is on the upper end of what is recommended).

In reality, finding experts and securing their participation was a challenging task. For example, our lead bird expert sent 118 invitation emails to people and organisations across 36 of sub-Saharan Africa’s 42 countries. Bird clubs, societies and NGOs were contacted (e.g., 21 Bird Clubs and 10 Birdlife country offices), as well as researchers, extension officers, local bird guides and tour companies. This resulted in 21 participating bird experts. This 80% non-response rate may be due to digital inequities, outdated information on websites, lack of incentive for non-researchers to participate in research, language barriers (particularly in Francophone countries), etc. Similarly, our lead primate expert sent out 65 invitation emails to people and organisations across 15 sub-Saharan African countries, with 24 ultimately participating. We can include these numbers in our new Supplementary Methods that will provide an example of the process.

We have added text to the methods section to clarify the role of the lead expert in identifying diverse experts and reflecting on the challenges of recruiting experts: *“Lead experts were briefed to find and reach out to people that met our definition of expertise, focusing on diversity of expertise geographically, taxonomically and professionally, as well as aiming to include as many African experts as possible. Their role was intended to reduce potential biases towards well-published experts, taxa and geographies. These persistent biases were challenging to completely overcome, however. Many but not all of the participating experts were scientists (107 of the 200 have university affiliations), and more experts were knowledgeable about Southern and East African biodiversity than West or Central African biodiversity. Most of the experts who participated reside in Africa (72%) and/or are originally from Africa (59%). Importantly, those not from or currently residing in the region still met our criteria of having place-based knowledge as a result of working in the region (mostly as scientists but also for example as tour operators or guides, for conservation organisations, etc). This place-based knowledge informed the creation of a quantitative dataset through an iterative, context-based and interactive expert elicitation process.”*

Figure 2 in Clements et al. 2024¹. Attributes of the 200 participating experts. All values in white font (and black font on the cord plot) represent the number of experts. Numbers do not add up to 200 when categories are not mutually exclusive (region, taxonomic group, employment sector) or when experts did not report a certain attribute (unk = unknown; Mamm = Mammals; org. = organisation).

Choosing diverse experts is analogous to increasing the “degrees of freedom” in a dataset – and asking similarly trained experts and colleagues will severely degrade the independence between experts and be akin to pseudosampling. See the following references for discussions on expert selection and elicitation processes

Burgman, M. 2005. Risks and decisions for conservation and environmental management. Cambridge University Press.

Burgman, M. A., M. McBride, R. Ashton, A. Speirs-Bridge, L. Flander, B. Wintle, F. Fidler, L. Rumpff, and C. Twardy. 2011b. Expert status and performance. PLoS One 6:e22998.

Singh GG, Sinner J, Ellis J, Kandlikar M, Halpern BS, et al. (2017) Correction: Group elicitations yield more consistent, yet more uncertain experts in understanding risks to ecosystem services in New Zealand bays. PLOS ONE 12(12): e0190326

Tetlock, P. E. (2005). Expert Political Judgment How Good Is It? How Can We Know?, Princeton University Press.

-While this book covers experts of social science, it is one of the best works documenting the limitations of expert status. It also largely corroborates the work Mark Burgman showed in conservation scientists.

RESPONSE: Thank you for sharing these references. Burgman et al. 2011 and Singh et al. 2017 both find that (to quote Burgman) *“if experts are given the opportunity to listen to one another, assess other judgements, and cross examine reasoning and data within a structured process, their average performance improves substantially. Additionally, the averages of a group's independent best guesses following discussion generally perform at least as well as, and often much better than the estimates of the best-regarded person in the group”*. The IDEA protocol that we used incorporates this research aimed at improving expert performance into its design, with experts given the opportunity to listen, assess and cross-examine during a discussion meeting, followed by a second round of independent estimation. Thus, in addition to aiming to select a diverse group of experts as outlined above, our elicitation process was guided by the evidence on how to improve the performance of the expert group. We have added an additional paragraph to the methods to elaborate on this protocol and how we addressed the potential issues that the reviewer notes.

I would suggest that one of the best ways to utilize a snowball approach to address knowledge breadth is to snowball across experts (and not just the “lead” expert) and track the “network” of experts to see if the expert pool saturates (or includes the majority) of potential experts. This process is probably modeled best in the following paper:

Ban, S. S., R. L. Pressey and N. A. J. Graham (2014). "Assessing interactions of multiple stressors when data are limited: A Bayesian belief network applied to coral reefs." *Global Environmental Change* 27: 64-72.

RESPONSE: We note that Ban et al. 2014 used scientific publications and ‘top-cited’ papers to identify experts and then snowballed based on those experts. By contrast, in line with what the reviewer recommends, our approach as explained above aimed to look beyond ‘published’ experts to include a wider diversity of field-based experts, for example working as tour guides or rangers. As mentioned above, we now clarify the role of lead experts in the paper and that they *“...facilitated a snowballing approach among experts where needed to increase diversity and numbers. Their role was intended to reduce potential biases towards well-published experts, taxa and geographies.”*

While we understand the value of reaching saturation, it would have been impractical for this study, as it would have been very difficult to know how large the total pool of experts was (in contrast to the cited paper where the pool was defined according to the published literature, making it possible to identify saturation). It was (a) extremely challenging to find all these experts (the lead author and experts spent several weeks each trying to track down experts, find current emails, etc - see bird and primate examples above) and (b) we would then have potentially ended up with a larger number of experts than can be handled effectively in an elicitation. In addition, some taxonomic groups have many more experts than others, and for many taxonomic groups the total pool of experts is heavily biased towards South Africa. Instead, we opted for an approach where the lead expert was specifically tasked with sourcing a diverse set of ~20 experts, across geographic regions, professions and taxa. We contend this 1) created consistency across the approaches used by the different leads, and 2) most likely led to a more diverse group of experts than selecting a random sample from a saturated group of experts.

If the authors did not take this quality assurance step in their methods, I would encourage them to consider the implications of their results. Are there hidden uncertainties in their results? At worst, if

there are significant selection biases in their experts the uncertainties would be systematic (biased) and the large sample of experts may actually reinforce the bias rather than reduce uncertainty.

RESPONSE: We hope that we have addressed the reviewer's concerns about our expert selection process in response to the earlier comments. We have expanded the methods section to explain how we aimed to mitigate potential biases in our expert selection by focusing on selecting a diversity of experts and through the elicitation process using the IDEA protocol. We will add a sentence stating that "The attributes of the 200 participating experts along key dimensions of diversity are summarised in Clements et al.¹ Figure 2". We also plan to add a caveats section to the paper where we will reflect on the potential limitations and biases of the set of experts we engaged drawing on the literature that the reviewer provides.

Please also see our new independent validation test, presented in response to Reviewer 1 above. We believe that the results of this test also help show that limitations in our expert selection process did not lead to systematic biases that compromised the robustness of our results.

There is also very little information on how experts were elicited to quantify their BII scores. There is much research to indicate that experts can more reliably provide ranges of estimates than point scores, and even then it is worth "nudging" experts by questioning their certainty to see what scores they provide.

Morgan, M. G. (2014). "Use (and abuse) of expert elicitation in support of decision making for public policy." *Proceedings of the National Academy of Sciences* 111(20): 7176-7184.

Speirs-Bridge A, Fidler F, McBride M, Flander L, Cumming G, Burgman M. Reducing overconfidence in the interval judgments of experts. *Risk Analysis*. 2009;30(3):512–23. Pmid:20030766

RESPONSE: We considered (and developed questionnaires) for a range estimate approach but ultimately opted for the point estimate approach after trialling the approaches on lead experts, since it was found to be cumbersome and confusing for experts to provide ranges for multiple species groups and land uses. Importantly, however, we did nudge experts by questioning their certainty both in the estimation process (as highlighted in Clements et al.¹: "Experts were encouraged to provide any comments relevant to each estimate (e.g., land use characteristics that could influence their score, assumptions that they made, uncertainties, the likely score range across species in a group, and any other explanatory information)" and in the discussion meeting (where the facilitated discussion aimed to "interrogate sources of variability, improve the consistency with which experts were interpreting the species response groups and land uses, and cross-examine reasoning, assumptions and evidence, thereby sharing insights between experts to promote learning"). We have added these details to the methods section of the paper and explained why we adopted a point-based approach instead of a range-based approach. We will also note the limitations of the point-based approach in the new caveats section in the paper methods, reflecting on the risk that it led to expert overconfidence, and how we mitigated this through nudging.

Please also see the results of our new validity test in response to Reviewer 1, which suggests that our eliciting of point estimates, combined with nudging, led to robust results.

I would encourage the authors to justify how species groupings were done relative to expert judgement. I understand the ecological justification for species groups, but it would speak more to data quality to relate these decisions to expert familiarity. I understand the pragmatic necessity to group species given the vast numbers of species that are being addressed, but how species are

groups will affect expert scores. If the groups are made in such a way that experts are not familiar with a grouping or thinking about a group of species in the way identified, the quality of the scores will likely be questionable. I would extend the same comment to the grouping by biomes. Again it may make ecological sense, but the authors should relate the groupings to expert familiarity to help assess data quality.

RESPONSE: We agree that it was imperative that the experts understood the groupings. This was part of the reason why the grouping exercise also included the participating experts (as explained in Clements et al.¹: “...draft species response groups were presented to the participating experts for that taxonomic group during an introductory planning meeting (see Structured expert elicitation section below) and revised based on experts’ feedback”). Lead experts presented the draft groups to the experts during this meeting and facilitated a discussion where experts gave input into these groupings and collaboratively revised them during the meeting. Experts were then provided with details of the groupings in the survey spreadsheet where they entered their estimates (e.g., including example species or full lists of species, species attributes such as fossorial or rupicolous, etc; see ‘Sp_Groups’ tab of the bii4africa dataset published in Clements et al.¹). Experts were also provided with a map of the biomes in the introductory meeting, and one biome name was revised based on expert feedback during the meeting (‘humid’ was added to Caesalpinoid-miombo savanna). To clarify these points in the current paper we have added the following to the methods section: “These groups were decided on by the participating experts during the introductory phase of the elicitation process and are summarised in the ‘Sp_Groups’ tab of the bii4africa dataset¹”. We also added: “the intactness scores of distinct plant species response groups in different land uses were estimated within each of eight biomes (forest, Caesalpinoid-miombo humid savanna, mixed-acacia savanna, grassland, shrubland, thicket, desert and fynbos, with a map of these biomes provided to experts during the elicitation)¹.”

Please also see the results of our validity test in response to Reviewer 1. The robustness of the results across species suggest that species groups and biomes were consistently understood by experts.

The author description of identifying and describing land uses shows built-in training in the elicitation process, which is great, but I found the writing of the section confusing because of the constant switch between what was done and why other scale considerations would not be as good. First, in asking experts to think of “average” landscapes based on visual cues of land uses and descriptions – people are prone to the “availability bias” when trying to imagine representatives, which may vary considerably across people based on their experiences. Perhaps that was part of the design and the aggregation stage was meant to address this, but I think in communicating the methods the authors should address how they dealt with these cognitive biases.

RESPONSE: We are glad that the reviewer supports our approach. The reviewer is correct that it was part of the design to ask experts to base their estimates of land use impacts on their experience in representative landscapes. These estimates were then averaged across experts to address the availability bias of each individual expert. Importantly, our use of confidence intervals reflects the uncertainty/variation caused by availability bias. Availability bias would only be a problem if we expected all experts to hold knowledge for the same subset of representative landscapes. Given the diversity of experts’ experiences across different geographies (as outlined in response to your earlier comment), we have no reason to expect this to be the case. We have added the following sentence to our methods section in the landscapes paragraph to clarify this: “Experts based their estimates on their experience in representative landscapes, with the averaging of estimates across experts addressing the availability bias of individual experts” to address how we dealt with these cognitive biases. We also elaborate in our sentence on

confidence intervals: *“The variability in scores between experts (expressed as 95% confidence intervals) were used to reflect the degree of uncertainty around BII estimates, arising for example from differences in experts’ place-based knowledge (and associated availability biases), unknowns or disagreements around how some species groups are impacted by human land uses, and variability in the number of experts contributing to each estimate.”*

Please also see the results of the additional validity test, in response to Reviewer 1. The robustness of the results across species (including rare and localised species and species where there are few experts) suggest that availability biases did not substantively bias the results.

My more serious concern here is that while experts were asked to consider integrated impact of all characteristics of the landscape, it seems that there is potential for experts to be considering different problems (because different experts may be identifying different impacts). This so-called “linguistic uncertainty” whereby experts may translate “all characteristics” differently can lead to underappreciated uncertainty and instability between experts.

Regan, H.M., Colyvan, M. and Burgman, M.A. (2002), A TAXONOMY AND TREATMENT OF UNCERTAINTY FOR ECOLOGY AND CONSERVATION BIOLOGY. *Ecological Applications*, 12: 618-628. [https://doi.org/10.1890/1051-0761\(2002\)012\[0618:ATATOU\]2.0.CO;2](https://doi.org/10.1890/1051-0761(2002)012[0618:ATATOU]2.0.CO;2)

RESPONSE: We agree there is room for variability in interpretation of landscape impacts. We aimed to reduce this through the detailed descriptions and representative images for each land use, as the reviewer mentions above, in an effort for experts to have as clear a picture as possible of the land use in question. In addition, the introduction and discussion meetings and comments section for each estimate in the survey were intended to identify and reduce this uncertainty. Regarding the comments, we explain in Clements et al.¹ that *“This qualitative information was useful when aggregating the data (step 2), to gain insight into the reasoning behind experts’ scoring, and detect potential inconsistencies between experts.”* Regarding the discussion meeting, *“Project and expert leads reflected on key trends and sources of variability, and any insights or discrepancies (e.g., when it was apparent from experts’ spreadsheet comments that they were interpreting a given land use in different ways). The project lead then facilitated a discussion among the experts, where they were encouraged to share their experiences from Round 1 (e.g., with what did they struggle; what helped them) and to reflect on the aggregated results (e.g., their insights for the species response groups they know well, or any results that surprised them). They were encouraged to discuss outlying (anonymous) expert estimates and why they may have occurred.”* These steps were aimed at addressing linguistic uncertainty, and we will add these details to the methods section of the current paper to clarify this. While this process reduced uncertainty, reflected by the reduced standard errors around expert estimates in the second, compared with the first, round of the expert elicitation (see validation section in Clements et al.¹), we could not completely eliminate this kind of uncertainty. In contrast, some variability among experts’ estimates demonstrates that they are applying their place-based knowledge and helps capture the variation in particular land use categories across the region, which is an important element of the approach. This variability is captured in the confidence intervals that we report in the results, as now elaborated on in our methods: *“The variability in scores between experts (expressed as 95% confidence intervals) were used to reflect the degree of uncertainty around BII estimates, arising for example from differences in experts’ place-based knowledge (and associated availability biases), unknowns or disagreements around how some species groups are impacted by human land uses, and variability in the number of experts contributing to each estimate.”*

Please also see our new validity test, in response to Reviewer 1’s request for assurance that this approach led to robust results, suggesting linguistic uncertainty did not invalidate the process.

I think the IDEA process is a good structure for elicitation, especially with iterative feedback. I do however have some comments on some of the details within the process presented. There is much research to indicate that experts can more reliably provide ranges of estimates than point scores, and even then it is worth “nudging” experts by questioning their certainty to see what scores they provide. The issues I present on the (lack of) details presented in how experts were elicited extends to the construction of the decision-tree used for the mapping – who was involved and how was this done? Finally, I note the extreme range of the numbers of experts involved in different groups of species. I understand the need for pragmatism (again) but the authors need to consider the range in reliability and robustness of responses as a result of this.

Morgan, M. G. (2014). "Use (and abuse) of expert elicitation in support of decision making for public policy." *Proceedings of the National Academy of Sciences* 111(20): 7176-7184.

Speirs-Bridge A, Fidler F, McBride M, Flander L, Cumming G, Burgman M. Reducing overconfidence in the interval judgments of experts. *Risk Analysis*. 2009;30(3):512–23. pmid:20030766

RESPONSE: We have addressed the query on ranges and nudging in response to the reviewer’s earlier comment, clarifying the additional details we have added to the methods section regarding the elicitation approach and reasoning behind it. We will also reflect on the limitations to our approach in a new caveats section, as highlighted above.

Regarding the decision-tree used for the mapping, we made use of an established, widely cited classification algorithm (e.g., Ellis et al.^{17,18}; Klein Goldewijk et al.¹⁹) that uses population density and land use area to assign pixels to categorical land use classes, reflecting the differences between the land uses described to experts. The assigning of intensity estimates within these land uses also used a protocol comparable with other highly cited human intensity indices (see review by Watson et al.²⁰), selecting variables appropriate to the regional context and aligning with the land use descriptions given the experts (e.g., of relevance for croplands is whether they are smallholder or commercial, together with the extent of croplands and nitrogen input intensities). The identification of intensity variables and thresholds was done by the author team (which includes all lead experts), through a series of online discussions and pilot maps.

We have elaborated on this in the ‘Mapping land use’ section of the methods as follows: *“We used an established decision tree classification algorithm^{17,18} built on (area) standardised thresholds of human population density (>1,000 people/km²) and/or land cover (>20%) to allocate each pixel in sub-Saharan Africa into one of six broad land use classes, reflecting the differences between the land uses described to experts” and “...we then scored each pixel based on its intensity, using a protocol comparable with other highly cited human intensity indices²⁰. Proxies of intensity relevant to each class were selected to reflect the land use descriptions for which experts estimated intactness scores through a series of pilot maps and online discussions between the lead experts and other paper authors”.*

Regarding variability in expert numbers per elicitation, as the reviewer mentions there was a need for pragmatism, and it was difficult to land on identical numbers of experts providing estimates for each group of species in each land use. We make use of confidence intervals to account for this variability/uncertainty, since sample size is embedded in these intervals (with lower confidence / large intervals for estimates that had fewer contributing experts). In the methods we state, *“The variability in scores between experts (expressed as 95% confidence intervals) were used to reflect the degree of uncertainty around BII estimates.”* We have added *“...arising for example from differences in experts’ place-based knowledge (and associated availability biases), unknowns or disagreements around how some species groups are impacted by human land uses, and variability in the number of experts contributing to each estimate.”*

I have made some mention of this already, but I am concerned about the extent to which results actually reflect experts and not the analysts. The expert input feed into a pre-defined notion of what biodiversity intactness is and is only part of equations to calculate this. Was there any work to engage experts to see if the definitions, standards, and reference points of the analysis correspond with how the experts think of the problem? Because of the bounding of the analysis, have the authors considered that the results are more affected by the pre-defined equations rather than the expert input? For a study showcasing expert input, I find the lack of engagement with these very fundamental issues troubling.

Aven, T. and S. Guikema. 2011. Whose uncertainty assessments (probability distributions) does a risk assessment report: the analysts' or the experts'? *Reliability Engineering & System Safety* 96:1257-1262.

RESPONSE: The reviewer is correct that we made use of the Biodiversity Intactness Index, which has a published definition and equation¹¹, and is bounded in nature. The reviewer is also correct that the expert-elicited intactness scores are only part of the BII equation, but they are the major part, the other parts are to enable the mapping of these scores: maps of species composition (based on well-established IUCN range maps) and land uses (which align with the land uses discussed with experts and are mapped based on published approaches). While we appreciate that the bounded nature of the assessment had limitations, we also think that such bounding was necessary for our aim, which was not to co-produce a definition of biodiversity intactness for the region, but rather to use and embed place-based knowledge in a well-established index that has been recognized for its theoretical and practical strengths (Mace et al.²¹).

We agree fully with the reviewer that critical to the validity of this approach was that experts understood and accepted the predefined notion of biodiversity intactness and approach to estimating it. In the invitation email and introductory meeting, we explained the index (both conceptually and how we would calculate it informed by expert-elicited intactness scores), and experts had the opportunity to ask questions or voice concerns. Experts generally asked clarifying questions, and no major concerns were raised. As noted in the paper, the introductory meeting of the large mammal elicitation did actually result in a change to the approach (to elicit at species level instead of response group level), based on expert feedback that they had the necessary knowledge at this level (and experts then suggests additional experts to plug species-level gaps). The discussion meeting provided another opportunity to voice challenges and concerns, with these largely focused on expert uncertainties around certain estimates they had given and wanting to talk through those.

Experts also had the option to choose not to participate should they feel that such an approach did not resonate with them and/or that they were not able to contribute meaningfully. Of the 266 experts that agreed to participate in the elicitation, 66 (25%) did not see the process through to completion. Of those that gave a reason, the vast majority stated time limitations, while a handful stated that they did not feel they had the necessary expertise. We will clarify these steps taken to ensure that experts understood the wider process they were contributing to, and were comfortable with that, in the methods section.

We will also add an Extended Data Table to the methods section that clarifies which aspects of the results are informed by (1) the literature (the BII definition and equation, land use mapping algorithm); (2) iterative discussions between lead experts (the reference state, the land use class descriptions and variables, the biomes); (3) inputs from the full group of experts (the species functional response groups); and (4) the structured expert elicitation (intactness scores). We will

reflect on the limitations of this bounded approach in the new caveats section of the paper methods, drawing on the literature that the reviewer has suggested.

Beyond the expert elicitation, I also have issues with the framing of the paper and some more quantitative analysis. I go into the specifics below

Title – since this assessment is for sub-Saharan Africa, the title should reflect that and not suggest that it is for the whole continent. This is to address both accuracy issues as well as avoid the often-made conflation of “Africa” as some homogenous place instead of diverse continent.

In light of this comment, together with Reviewer 1’s first comment (please see our detailed response to that comment) and Reviewer 2’s related comment, and in keeping to 75 characters, we suggest changing the title to “Sub-Saharan Africa’s biodiversity intactness using placed-based knowledge”

Line 66 – The links between biodiversity, and any specific status of the environment, with global sustainable development, are tenuous and not at all settled. The author refer to literature on the Planetary Boundaries, but this framework has been heavily criticized throughout its history, and perhaps most aggressively on issues of biodiversity. I know that the planetary boundaries authors have replied to many challenges, but in my view (and many others in development communities, and throughout many science disciplines) the responses do not address the severe challenges and problems raised. I am not suggesting the authors retract their statements, but they must make them in context of the unsettled nature of the debates if raised.

Biermann F, Kim RE. The boundaries of the planetary boundary framework: a critical appraisal of approaches to define a “safe operating space” for humanity. *Annu Rev Env Resour.* 2020;45:497–521.

Montoya, J. M., I. Donohue and S. L. Pimm (2018). "Planetary Boundaries for Biodiversity: Implausible Science, Pernicious Policies." *Trends in Ecology & Evolution* 33(2): 71-73.

You do not need to overextend the importance of biodiversity to make the case for this study. Or if you do, it should be done fairly, while pointing to the variety of evidence that disagrees with these kinds of statements. Highlighting the local and regional importance of biodiversity is likely enough to point out, and this will be more accurate and less contentious

RESPONSE: We have rephrased Line 66 (and similar statements in the manuscript) to rather emphasise that retaining biodiversity has become an integral part of sustainable development agendas (as opposed to stating that it is the bedrock of sustainable development itself): “Biodiversity is increasingly considered to be an integral part of sustainable development agendas...”.

We have also added text noting the contested nature of the planetary boundaries where we mention this concept in the introduction, citing the references provided by the reviewer: “..., though both conceptual and practical challenges with the concept have been raised^{22,23}.” It is worth also noting that Mace et al.²⁴ suggest some ways to address the issues associated with the biodiversity planetary boundary, with our approach aligned with these suggestions. We have added the following sentence to follow that on planetary boundaries: “Notably, our approach aligns with suggestions to address some of the challenges associated with the biosphere integrity boundary²⁴, including assessing biome integrity and BII at regional (as opposed to global) levels.”

Lines 113 -The authors make the point that there are limitations of top-down approaches to understanding ecosystems and points to the novelty of their approach in consulting local experts. However, it's not clear to what degree the authors have considered that the structure of their analysis is very top-down: thinking of biodiversity intactness in reference to pre-industrial society is a subjective standard that historically largely follows European notions of nature. Have the authors looked to see if the experts they consult follow these notions? These kinds of questions are important because if the experts do not think on these terms, then the responses they provide may not be as robust. That is, if questions are posed in ways that experts are not familiar with, they may not be accurately considered experts.

RESPONSE: We appreciate that the bounded nature of our approach is top-down to some extent, but part of the challenge we aimed to overcome was how to integrate more 'bottom-up' (place-based) knowledge into a large-scale regional product that speaks to the needs of national and international decision making, as opposed to the dominant approach of using outputs from global data-driven models that are far less appreciative of context. If we had not adopted a structured and bounded approach based on an established index with a defined reference state, we think such a task would have been unmanageable. We now elaborate in the introduction: "*Here we demonstrate how place-based knowledge can be mobilised in a 'bottom-up' yet large-scale assessment of biodiversity intactness—a key component of ecosystem condition²⁵. Our approach aligns with recent calls for the integration of place-based knowledge into global sustainability policies and actions, and the focus on 'region' as a bridge between local and global sustainability initiatives to overcome cross-scale integration challenges² through 'contextualised generalisations⁶."* As stated in response to the reviewer's earlier comment, we will reflect on the bounded and structured nature of our approach in a new caveats section in the paper.

A measure of intactness/integrity is by nature a relative measure. The Global Biodiversity Framework defines the notion as "*The degree to which the ecosystem's composition, structure and function resemble those characteristic of its natural range of variation, which may be defined from historical or minimally disturbed reference states, replicated contemporary samples, ecosystem models and/or expert judgement*". We agree with the reviewer that this is a particular way of thinking about the state of biodiversity, and that science in general and ecology in particular are dominated by Western ways of thinking²⁶. Here we are countering this Western dominance by bringing more African expertise into ecology and adding more insight around important aspects of local and regional context that are often ignored (e.g., the difference between smallholder vs commercial croplands), while still using notions and measures that are embedded in international agreements such as the Global Biodiversity Framework that countries are required to report to. We recognise the limitations of this approach, but believe it is an important first step. We note in the paper that the reference state we made use of was "*before alteration by modern (industrialised, colonial and post-colonial) society*" and "*In most parts of Africa, this corresponds to populations before the substantial alteration of the landscape triggered by colonial settlement*". Such a reference, importantly, includes Indigenous people as part of this intact state (pushing back against Western notions of wilderness), and recognises the impact of both colonisation and industrialisation on this state.

The index and its reference were presented to experts in the introductory meetings. We have added text to the 'Intactness scores' section of the methods to clarify this. While no major objections to either the chosen reference or the concept of a reference were raised, neither would have eroded the quality of the assessment provided experts understood the concept and the reference to use. We endeavoured to provide this understanding in the introductory meeting and

written instructions. The consistency in expert scoring, and the robustness of the results compared with the IUCN Red List indicate that this understanding was sufficient. A reflection on reference points and their limitations will be added to the new caveats section of the paper.

Results –

Some results seems rather tautological. Since the BII is defined as intact nature relative to preindustrial settings, this measure shares the same information with indices about human activity and human footprint. I understand these relationships are partially meant to validate the measure, but here I have some questions as well. First, given the shared information, it's not really surprising that the best relationship seems to be between the BII and the Human Modification Index compared with the biomass modification and habitat indices. Perhaps this shows some evidence of predictive validity of the measure. However, it speaks to questions about the potential construct validity of the measure: since it does a better job tracking human footprint than it does biomass and other ecological variables, is it tracking what we want it to? I think questions about the tautological nature of the validation, and a broader discussion of validity is needed.

RESPONSE: We agree it is not surprising that BII and other measures of human activity and habitat/biomass are correlated. What IS surprising is that the global BII model does not show this expected correlation: *“Unexpected relationships between a BII map produced by a global modelling approach²⁷ and other maps of human pressure, remaining habitat and biodiversity hotspots has led to concerns about the index²⁸”* (in the validation section of results). As such, the one objective here was to demonstrate that our map, unlike the global BII one²⁷, does correlate broadly with these other indices as expected. This broad ‘sense-check’ is not completely tautological, since *“while these expected correlations corroborate our BII assessment, the variability is also important, since the BII provides insights into how specific groups of species respond to different human pressures and is thus expected to vary from measures than simply quantify aggregated human pressure (proxied by HFI) and the impacts of human pressures on habitat (proxied by BMI and BHI)”* (presented in the validation section of results). Recognising that biomass and habitat metrics are also linked to human activity, it is perhaps unsurprising that BII better ‘tracks’ human footprint than other derivatives of human footprint and we have added this discussion point to the validation section.

Importantly, the BII is not merely an extension of human activity metrics. Even when two regions have the same land use intensity, one may have a more vulnerable suite of species due to biogeographical factors. For example, we might expect savanna species, adapted to disturbance, to be more resilient to land use change than forest species, which are not disturbance adapted. This has major policy implications for how conservationists design protected area networks and can inform e.g. the land-sparing versus land-sharing strategies. A pure human footprint map captures none of this biogeographical complexity. Furthermore, while the BII itself does not depict changes in biomass, it can be combined with other variables to do exactly that - another paper currently in review at Nature²⁹ combines the BII with species population density datasets to quantify changes to biomass and energy flows through species, and ultimately to track changes in animal-mediated ecosystem functions. This is one of the reasons the BII is a useful metric compared to others such as HFI, BMI, etc — its high resolution in terms of both species and space means it can also be used as a foundation for other novel metrics that track changes to the biosphere.

We have also added an additional validation test to this section, as presented in response to Reviewer 1, which is based on the IUCN Red List and does not have a similar ‘human pressure’ spatial input, though of course human activity also impacts species threat statuses.

Second, the plotted figures only show the “average” indices of human modification and habitat indices in each ecoregion. To better represent the data, the variance or some measure of spread should also be illustrated. This will help understand how closely the BII corresponds to the range of measures used.

RESPONSE: Thank you for this suggestion. We have added standard deviations to these plots—see new plots below.

Extended Data Fig. 6. Relationships between the Biodiversity Intactness Index (BII) and other human pressure indicators. These indicators include the (a) Human Footprint Index³⁰ (HFI), (b) Biomass Modification Index³¹ (BMI) and (c) Biodiversity Habitat Index³² (BHI). Points are average values for each of sub-Saharan Africa’s 89 ecoregions, with lines reflecting standard deviations around these means. Ecoregions with ≥50% of their area inside a biodiversity hotspot³³ are shown by red points and lines. The boxplot (d) shows the spread of BII scores across hotspot and other ecoregions.

Quantitative analysis

-since mean \pm 95% CI was calculated across pixels, the authors need to be cautious of spatial autocorrelation effects potentially inflating measures of precision, and affecting statistics calculated. I encourage the authors to estimate sampling distances needed to eliminate spatial autocorrelation, then randomly sample pixels given this distance to calculate the mean and CI estimates.

RESPONSE: Spatial autocorrelation can be an issue when validating a map based on reference data for a subset of locations. These sample estimates serve as an approximation of population values. In our case, however, we make use of population values and not sample values in both our analyses and the validation. Thus, spatial autocorrelation is not an issue. These confidence intervals reflect confidence around expert scores (based on the variability of scores between experts), not variance attributed to cross validation or bootstrapped sampling.

Did authors account for test assumptions when conducting correlations and t-test? Independence is already questionable, but do other assumptions hold? Did the authors do any diagnostics? Some of the relationships look potentially heteroskedastic, for example, meaning the error structure varies (which violates a lot of statistical model assumptions). Perhaps nonparametric tests are more relevant.

RESPONSE: We tested that the assumptions held and can confirm that all relevant assumptions were met.

References

1. Clements, H. S. *et al.* The bii4africa dataset of faunal and floral population intactness estimates across Africa's major land uses. *Sci. Data* **11**, 191 (2024).
2. Balvanera, P. *et al.* Interconnected place-based social-ecological research can inform global sustainability. *Curr. Opin. Environ. Sustain.* **29**, 1–7 (2017).
3. Sievers, E., Spierenburg, M., Jhagroe, S. S. & van Oudenhoven, A. P. E. Place-based knowledge transfer in a local-to-global and knowledge-to-action context: key steps and facilitative factors. *Ecol. Soc.* **29**, (2024).
4. Norström, A. V. *et al.* Principles for knowledge co-production in sustainability research. *Nat. Sustain.* **3**, 182–190 (2020).
5. Martín-López, B., Balvanera, P., Manson, R., Mwampamba, T. H. & Norström, A. Contributions of place-based social-ecological research to address global sustainability challenges. *Glob. Sustain.* **3**, e21 (2020).
6. Reyers, B., Moore, M. L., Haider, L. J. & Schlüter, M. The contributions of resilience to reshaping sustainable development. *Nat. Sustain.* **5**, 657–664 (2022).
7. Buschke, F. T. *et al.* Make global biodiversity information useful to national decision-makers. *Nat. Ecol. Evol.* **7**, 1953–1956 (2023).
8. Robinson, T. P. *et al.* *Global Livestock Production Systems*. (2011).
9. Ryan, C. M. *et al.* Ecosystem services from southern African woodlands and their future under global change. *Philos. Trans. R. Soc. B Biol. Sci.* **371**, 20150312 (2016).
10. Slooten, E., Jordaan, E., White, J. D. M., Archibald, S. & Siebert, F. South African grasslands and ploughing: Outlook for agricultural expansion in Africa. *S. Afr. J. Sci.* **119**, 3–6 (2023).
11. Scholes, R. J. & Biggs, R. A biodiversity intactness index. *Nature* **434**, 45–49 (2005).
12. Palacio, R. D. *et al.* The global influence of the IUCN Red List can hinder species conservation efforts. *Authorea Prepr.* 1–22 (2023).
13. Hawkins, F. *et al.* Bottom-up global biodiversity metrics needed for businesses to assess and manage their impact. *Conserv. Biol.* **38**, e14183 (2024).
14. Hempton, G. P., Archibald, S. & Bond, W. J. A continent-wide assessment of the form and intensity of large mammal herbivory in Africa. *Science* **350**, 1056–1061 (2015).
15. Fritz, H. & Duncan, P. On the carrying-capacity for large ungulates of African savanna ecosystems. *Proc. R. Soc. London B* **256**, 77–82 (1994).
16. Hemming, V., Burgman, M. A., Hanea, A. M., McBride, M. F. & Wintle, B. C. A practical guide to structured expert elicitation using the IDEA protocol. *Methods Ecol. Evol.* **9**, 169–180 (2018).
17. Ellis, E. C., Goldewijk, K. K., Siebert, S. & Lightman, D. Anthropogenic transformation of the biomes,

- 1700 to 2000. *Glob. Ecol. Biogeogr.* **19**, 589–606 (2010).
18. Ellis, E. C., Beusen, A. H. W. & Goldewijk, K. K. Anthropogenic biomes: 10,000 BCE to 2015 CE. *Land* **9**, 8–10 (2020).
 19. Klein Goldewijk, K., Beusen, A., Van Drecht, G. & De Vos, M. The HYDE 3.1 spatially explicit database of human-induced global land-use change over the past 12,000 years. *Glob. Ecol. Biogeogr.* **20**, 73–86 (2011).
 20. Watson, J. E. M. & Venter, O. Mapping the continuum of humanity’s footprint on land. *One Earth* **1**, 175–180 (2019).
 21. Mace, G. M. Biodiversity: An index of intactness. *Nature* **434**, 32–33 (2005).
 22. Biermann, F. & Kim, R. E. The boundaries of the planetary boundary framework: A critical appraisal of approaches to define a ‘safe operating space’ for humanity. *Annu. Rev. Environ. Resour.* **45**, 497–521 (2020).
 23. Montoya, J. M., Donohue, I. & Pimm, S. L. Planetary Boundaries for Biodiversity: Implausible Science, Pernicious Policies. *Trends Ecol. Evol.* **33**, 71–73 (2018).
 24. Mace, G. M. *et al.* Approaches to defining a planetary boundary for biodiversity. *Glob. Environ. Chang.* **28**, 289–297 (2014).
 25. Nicholson, E. *et al.* Scientific foundations for an ecosystem goal, milestones and indicators for the post-2020 global biodiversity framework. *Nat. Ecol. Evol.* **5**, 1338–1349 (2021).
 26. Trisos, C. H., Auerbach, J. & Katti, M. Decoloniality and anti-oppressive practices for a more ethical ecology. *Nat. Ecol. Evol.* **5**, 1205–1212 (2021).
 27. Newbold, T. *et al.* Has land use pushed terrestrial biodiversity beyond the planetary boundary? A global assessment. *Science* **351**, 600–604 (2016).
 28. Martin, P. A., Green, R. E. & Balmford, A. The biodiversity intactness index may underestimate losses. *Nat. Ecol. Evol.* **3**, 862–863 (2019).
 29. Loft, T. *et al.* Energy flows reveal declining ecosystem functions by animals across Africa. *Nature preprint*, 1–25 (2024).
 30. Venter, O. *et al.* Sixteen years of change in the global terrestrial human footprint and implications for biodiversity conservation. *Nat. Commun.* **7**, 12558 (2016).
 31. Erb, K. H. *et al.* Unexpectedly large impact of forest management and grazing on global vegetation biomass. *Nature* **553**, 73–76 (2018).
 32. Harwood, T. *et al.* *BHI v2: Biodiversity Habitat Index: 30s global time series v1.* (2022).
 33. Olson, D. M. *et al.* Terrestrial Ecoregions of the World: A New Map of Life on Earth. *Bioscience* **51**, 933 (2001).

Referees' comments:

Referee #1 (Remarks to the Author):

This is a really interesting paper describing a first proper attempt to estimate the biodiversity intactness across the African continent. Recognising the challenge involved in direct measurement of this the authors opt for an expert-based assessment, identifying a cohort of over 200 taxonomic experts to help in the process. They find that relatively small proportions of current biodiversity are across the continent fall in protected sites, with much the larger part remaining in unprotected but untransformed landscapes, co-occurring alongside human activities. These results are important and broadly in line with what I would have expected - I'd love to have seen some comparisons with other continents to put this into context, but as a top level result this seems plausible.

RESPONSE: Thank you for the positive feedback; we are pleased that the reviewer thinks that the paper is interesting and the results important. Comparisons with other continents are difficult to do meaningfully at this stage since no other regional maps of BII are available (and in the paper we highlight the concerns raised about the validity of the global maps of BII). Hopefully, our work will inspire similar research in other regions, allowing for such comparisons that we agree would be most interesting.

I have to admit I was excited about reading this paper - Local and Indigenous Knowledge is a hot topic in current biodiversity discussions. IPBES has been praised for delivering a better job on this than many previous attempts, and the K-M GBF identifies improving its involvement as a key target. So I was slightly disappointed that despite the title and the talk of co-developing knowledge, which made me think there was going to be a new method of harnessing indigenous knowledge, this isn't really the case at all. Rather, the underlying data are the result of a fairly standard Delphi process involving experts from across the world (though in reality a disappointingly small fraction of experts are not from N. America, Europe or South Africa). I struggle to really see this as 'local knowledge' when, for example, only two of the ornithologists are West African (both Nigerian), despite the existence of an active ornithological community in the region. To suggest that Southern African ecologists are more local to East or West Africa than geographically much closer European ecologists is not aligned with the CBD or IPBES use of the term 'local knowledge'. Nor do I see any sense of 'co-production' here in the usual usage of this term - I suggest the authors considerably revise their use of local and co-production.

RESPONSE: In light of this feedback and after consulting IPBES and recent literature, we agree that our use of the term 'local' is not aligned with current use of the term. We believe the term 'place-based knowledge'¹ is more appropriate for this paper and we have changed the term accordingly throughout the paper. While 'local' 'and 'place-based' both refer to knowledge that is contextual, the former is held by local people while the latter encompasses diverse forms of knowledge—including scientific, experiential, and local—rooted in particular landscapes or regions^{1,2}.

In our previously published data paper (Clements et al.³) describing the expert elicitation process and resulting 'bii4africa' dataset, we explain that "A broad definition of expertise was used to identify experts, centred on experience of how sub-Saharan species are impacted by human land uses. Diverse types of people can have such experience (e.g., researchers, field or tour guides, park rangers, conservation practitioners, museum curators, and consultants), and inclusion was thus not limited to specific qualifications or institutional affiliations." We have now added this information to the methods of the current paper (lines 642-645) and elaborated on the lead expert's role to identify diverse experts and the difficulty in overcoming persistent biases towards well-published experts, taxa and geographies (lines 660-668).

While we agree with the reviewer that a standard expert survey does not qualify as knowledge co-production, we think that our iterative, context-based and interactive expert elicitation process does meet the requirements of a knowledge co-production process. Knowledge co-production is defined as “*Iterative and collaborative processes involving diverse types of expertise, knowledge and actors to produce context-specific knowledge and pathways towards a sustainable future*”⁴).

Our process aligns with the four principles of co-production laid out by Norstrom et al.⁴ as follows:

1. *Context-based. Co-production processes should be considered and situated within the particular social, economic and ecological contexts in which they are embedded:* Our process specifically focused on place-based knowledge for particular land uses in the sub-Saharan African context.
2. *Pluralistic. Co-production of knowledge must explicitly recognize the multiple ways of knowing and doing.* As quoted above, our definition of expertise recognised experienced-based knowledge held by diverse types of people. We also elaborate in response to Reviewer 3 on how we aimed to capture diversity in expertise in our expert selection process and provide a summary of the different dimensions of diversity across the expert group (lines 641-669).
3. *Goal-oriented. Knowledge co-production for sustainability is problem-focused and benefits from clearly defined and meaningful goals shared among participants.* We had the specific goal of mobilising expert knowledge to produce intactness scores that could be used to produce a regional assessment of biodiversity intactness to feed into national and international decision-making. This goal was introduced to all contributing experts in the invitation email and introductory meeting, as detailed in response to Reviewer 3 and now added to the methods (lines 676-680). We also had the more immediate goal of making the co-produced *bi4africa* dataset of intactness scores widely accessible and this has been achieved through the publication of an open-access paper in *Scientific Data* that includes contributing experts as co-authors (Clements et al.³).
4. *Interactive. High-quality co-production requires frequent interactions among participants to occur throughout the process, extending from collaboratively framing and designing the research agenda, to conducting the research, and jointly using and disseminating the knowledge generated.* We elaborate in response to Reviewer 3 that our expert-elicitation process was iterative, and that during two meetings experts collaborated to produce the species functional groupings, as well as engaging in critical discussion around the estimates that emerged from the elicitation. Experts then contributed as co-authors to the publication of the resulting dataset, where they were given the opportunity to contribute to two drafts of the paper, as well as the revised paper subsequent to review. The lead author addressed 193 comments on aspects of the paper from contributing experts. As explained in response to Reviewer 3, lead experts were involved from the start of the project, including assisting in conceptualising the process; deciding on the land use class descriptions and variables, and the biomes; mapping and assessment; and results interpretation. This interactive process is now elaborated on in the paper methods (lines 676-680, 697-755, 762-763, 835-838).

Place-based expert assessment (though not always through a structured, iterative elicitation as done here) is the backbone of widely accepted biodiversity indicators and assessments including the IUCN Red List, IPBES and IPCC and an important form of knowledge production. The place-based knowledge we make use of in this paper is knowledge of how different functional groupings of species in the sub-Saharan African region respond to land uses characteristic of the region. We estimate biodiversity intactness across the region by generalising this place-based knowledge across similar land use contexts in a bounded way. Our approach aligns with recent research emphasising the need to integrate place-based research and knowledge into global sustainability policies and actions^{1,2,5}. Specifically, our approach speaks to “*the concept of region as a bridge*

between local and global sustainability initiatives” to overcome cross-scale integration challenges (Balvanera et al.¹ pg. 1) through ‘contextualised generalisations’⁶. We now explain this important contribution of our approach and work in the revised introduction (lines 89-101). We hope this also clarifies why our approach is not dependent on the extent to which experts are local knowledge holders, but rather the extent to which they hold place-based knowledge within the region.

Importantly, our approach presents an alternative to commonly used global mapping approaches that do not account for regional context⁷. In the paper introduction (lines 103-117) we give the examples of croplands and rangelands: global maps typically do not differentiate between smallholder vs large-scale commercial croplands, nor pasture vs rangelands, but these distinctions are critical in the sub-Saharan context for a wide range of impacts, including on biodiversity, livelihoods and ecosystem services. Our approach helps differentiate these impacts in ways that are often ignored in large-scale mapping exercises. We trust that our new validation against the IUCN dataset (lines 304-310) also confirms that our contextualised generalisations are robust.

Despite this slight disappointment caused by what might be considered some of the hype in the writing, the paper stands or falls on the basis of the work these 200+ experts undertook. For them to even attempt such a comprehensive review is highly commendable.

RESPONSE: Thank you for your commendation of the enormous effort this involved.

The difficulties of this are clearly acknowledged, and have resulted in grouping of species and geographies in order to make the problem tractable. It is therefore crucial that we trust this data reduction. If I have understood the methods correctly, each expert identified their particular expertise and were then asked to estimate an average score for the change in population of functional groups within their taxonomic speciality by landuse and biome. Thus an ornithologist would be asked to provide estimates for each of 17 groups of birds in each of 8 biomes, within which there may be up to 9 land uses. This would be (up to) 1224 total estimates for an ornithologist. (NB this summary wasn't easy to piece together involving both the paper for review and a cited paper and searching around the project website too and might be wrong, it would be great to see an example like this clearly laid out somewhere.)

RESPONSE: To address the reviewer’s important comment that it was not easy to piece together what was done, we have considerably expanded the ‘Intactness scores’ section of the methods (lines 625-782) to more fully describe the expert elicitation process, and also include a new extended figure (Extended Data Fig. 1) with a schematic of the process that includes a summary of the expert elicitation approach that is described in detail in our recently published data paper (Clements et al.³). This new figure also includes an example of the process and outputs for birds. Importantly, experts only estimated for the land uses and functional groups (and biomes, for plants) that they knew well. As reported by Clements et al.³: “On average, each expert provided 155 intactness scores”. We have also added this clarification to the methods section (lines 724-725).

Hidden in here is the idea that an individual ornithologist from Nigeria, say, could reasonably estimate the average % decline in aerial feeders (swallows and swifts) in all the savanna areas of Africa across the 9 land use tasks. I am deeply skeptical that any expert could do this. While it might be possible within one's own country where one's local knowledge is valuable, to estimate it across all the Acacia savannas in the continent as a whole seems implausible. Moreover, the reader is left to trust that the groupings are reasonable (I couldn't really work out what they were fully) and that the implicit assumption they respond similarly within a biome to land use change is reasonable. I

would like to see this tested at least at some level. Perhaps a quick evaluation of the current trends for those species included in the IUCN Red List would be useful - if we are to trust these data I would like to see agreement between the average red list category for each functional group and the expert-based estimates of this, and I'd like to see that the IUCN status of species within each taxonomic grouping is more similar than between functional groups.

RESPONSE: The reviewer is correct that our approach assumes that the population abundances of species within a functional response group are similarly affected by different land use changes. This published approach⁸ of focusing at the level of functional groups instead of individual species to make the problem tractable is what makes it possible to translate experts' place-based knowledge into a regional map. To ensure rigour in these functional groupings, the groupings were themselves informed by the knowledge of the expert group through a structured, interactive, and iterative process. We now explain this process in more detail in the paper methods (lines 684-702). We have also provided an example of the groups, for birds, in the new Extended Data Fig. 1.

We concur with the reviewer's concern that we need to provide additional evidence that our approach led to robust results and thank the reviewer for suggesting we use the IUCN Red List to do so. In the paper we now include a validation a test akin to the first comparison suggested by the reviewer (i.e., testing "*agreement between the average red list category for each functional group and the expert-based estimates of this*"), with the difference being that because our map can be disaggregated into its component parts we could actually do this comparison at species' level. In other words, we could assess whether a species' IUCN Red List category was a good predictor of its remaining BII (i.e., the average BII of its functional group across that species' particular range). However, we disagree on the usefulness of the second comparison that the reviewer suggests (i.e., demonstrating that species within functional groups have similar IUCN threat statuses, compared to species between functional groups). This comparison would not hold up for species' IUCN threat status, since the threat level for a species within a functional group depends strongly on the species' range and the land uses it is exposed to within that range. The threat level for species within a functional group can therefore vary widely, and comparing some kind of average threat status between functional groups is not informative. Our approach to estimating BII accounts for individual species' ranges and does not assume that all species within a functional group have the same range.

While we would expect to see general congruence between the predicted intactness of a species within its range and its IUCN threat category, it is important to note *a priori* that the relationship is unlikely to be perfect for several reasons. Firstly, intactness is estimated relative to pre-industrial populations while Red Listing is based in part on the proportional population decline over the past 10 years/3 generations (or projected into the future). Thus, if most population decline occurred before this period, the species may have lower intactness than expected based on its IUCN threat status. Similarly, if large future declines are projected, the species may have higher intactness than expected based on its IUCN threat status. Secondly, a species with a smaller absolute population size and/or smaller range size will be considered more threatened according to the Red List threat classification process, meaning two species with similar intactness scores could have different threat statuses because of differences in their population or range size. We can account for the effect of range size in our comparison, but we cannot account for shifted baselines or absolute population sizes. In addition to these issues, we also cannot assume that the IUCN Red List is a perfect reflection of reality, given that it suffers from outdated assessments and various methodological and data-availability drawbacks (reviewed by Palacio et al.⁹).

With these caveats acknowledged, we now include in the paper an additional validation analysis comparing each species' average BII across its range and its IUCN threat category. We find that

IUCN threat category is a significant predictor of intactness, with Critically Endangered species having significantly lower remaining intactness than moderately threatened (Endangered, Vulnerable) and Near Threatened species, which, in turn, have significantly lower remaining intactness than Least Concern species (Extended Data Fig. 6e).

This comparison of two largely independent datasets demonstrates that our approach of functionally grouping species, asking experts to estimate how those groups would be impacted by characteristic land uses, and mapping those estimates based on land uses and species ranges across the region, led to an intactness index that corresponds as expected with an independent assessment of the threat categories of individual species. We have added this validation to the methods (lines 945-970) and results (lines 304-310). We have also uploaded the relevant data and code.

A similar test of internal consistency in assessments by global vs local experts would also be reassuring. For example, I would love to have seen some evaluation of the differences in assessment by more or less local contributors. At least for the more geographically restricted biomes it would be interesting to see if (truly) local experts had different average assessments of change than experts from elsewhere in the continent, vs those based elsewhere in the world. Similarly, it might be interesting to check whether the large national differences in estimated BII are reflected in the average scores of experts from those countries - do, for example, experts from the relatively intact Mozambique on average estimate smaller declines than those from the much more heavily impacted regions of Nigeria or Rwanda? For the method to work well this needs NOT to be the case. These sorts of analysis could not only help us identify the real value of true local knowledge, and reassure us that the expert group did have the detailed knowledge required to make these very large-scale assessments.

RESPONSE: As explained in our response to the reviewer's helpful earlier comment on our inappropriate use of the term 'local knowledge,' in this paper we identified experts based on their knowledge of how sub-Saharan species are impacted by land uses characteristic of the region (lines 641-645). Since expertise was defined according to relevant place-based knowledge (as opposed to the need for experts to originate from these places), we do not think it is relevant to test whether 'more or less local' contributors differed in their estimates. It would also be difficult to make such a differentiation: country of nationality or residence is not a good proxy since many experts work across multiple countries. Please see our response to Reviewer 3's questions on our expert selection process and the additional information that we have included in the 'Intactness scores' section of the methods to elaborate on the process for selecting experts to promote diversity in place-based expertise (lines 650-669). As one important aspect of diversity, we specifically aimed to have a good representation of expertise from across the region, asking experts to note their subregion(s) of expertise (i.e., Central, East, Southern and/or West Africa). In response to the reviewer's suggestion, we have now tested whether an expert's sub-region of expertise impacted their intactness estimates published in the bii4africa dataset.

As the reviewer mentions, for our approach to work well we should not find an expert's sub-region of expertise to be a significant predictor of intactness scores for a given functional group of species in a given land use (e.g., experts working in more intact Central Africa should have provided similar scores for near-natural lands as those from less intact West Africa, etc). A critical aspect for our process to work well was to ensure that land uses were well differentiated, and land uses that have different effects were not lumped together into one category. Different experts should then be able to provide consistent estimates for these land uses, whether they hold 'very local' knowledge (e.g., for just one example of species in a given functional group being impacted by a given land use), or more regional knowledge (e.g., from working across different

species within a functional group and across different examples of a characteristic land use across the region). Similarly, estimates for functional group responses to specific land uses should be consistent between experts with knowledge from different areas. Even low intactness countries (e.g., Rwanda) maintain some high intactness land uses (e.g., Akagera National Park, Nyungwe forest), although these will make up a much smaller proportion of the country's total area than in highly intact countries. We therefore expected that experts with place-based knowledge from low intactness countries would be familiar with a range of land use classes, and able to estimate intactness for each. For example, Rwandan vs Mozambican experts should be able to consistently estimate impacts of dense urban areas vs near-natural lands on the populations of aerial feeding birds. The difference in BII between these countries lies in the relative extent of land use types and intensities, which is calculated from the mapping process. Experts only scored for the land uses (and species functional groups) they knew, so if a given land use was not present in the area an expert knew, they would have not provided an estimate. Our replacement of 'local' terminology with 'place-based' and the addition of details on what is meant by this in the revised introduction (lines 91-101), along with additional method details (lines 684-693) hopefully clarifies this approach of contextualised generalisation.

We tested our assumption that sub-region of expertise should not influence experts' intactness estimates by running a GLMM, assessing whether sub-region (as a fixed effect) is a significant predictor of expert's intactness estimates (as a response variable), including taxonomic group as a nested random effect (as explained in the new methods for our IUCN validation), as well as including land use as a random effect (since land use affects intactness but in this case we simply want to control for this effect). We find that sub-region is not a significant predictor of experts' intactness estimates ($F = 1.56, df = 4, p = 0.19$; Fig. R1 below), as expected. This result hopefully helps assure the reviewer that this method of contextualised generalisation works.

Fig. R1. The distribution of intactness score estimates provided by 200 experts, published in the bii4africa dataset (Clements et al.³). Boxplots show the spread of scores provided by experts with expertise in one or multiple sub-regions of sub-Saharan Africa. Boxes show median and interquartile range, lines show maxima and minima, and means are also shown with a cross. We find no significant difference in intactness scores according to expert's sub-region of expertise, controlling for taxonomic group, land use and expert ($F = 1.56, df = 4, p = 0.19$).

Our new IUCN validation (described above) provides additional internal consistency checks, as suggested by the reviewer. If experts' knowledge was indeed only locally applicable, and not

generalisable across the region, then we would expect that our approach of contextualised generalisation would have resulted in inaccurate estimates of the intactness of species within their regional range. By contrast, we show that these estimates consistently relate as expected with species' IUCN threat categories (lines 304-310). Notably, we find this relationship holds when controlling for species' range size, meaning it holds for both wide-ranging species and very 'local' (small range) species that few (if any) experts would have had specific place-based knowledge of. This consistency suggests that our approach of contextualised generalisation led to robust estimates. We flag this in the new validation results in the paper (lines 308-310). We hope that this validation will reassure the reviewer that the process we used was robust to variations in the expert group's extent and diversity of knowledge, as we only used estimates that linked directly to their expertise and experience.

Without total confidence in the underlying evaluation, the rest of the analyses are harder to review. I do think they are a sensible way of cutting the data up for analysis, and (provided the underlying data are trustworthy) the justifications and conclusions are robust with most analyses adequately explained and apparently sensibly undertaken. The only real question I raised here concerns the figures in Extended Data Fig 2, where I struggled to understand why there are such obvious lines in the raw data - presumably reflecting groups of one land use, or biome, or taxonomic group? This could do with explanation and these figures could be tidier in order to allow us to interpret and know whether this artifact is a problem or not.

RESPONSE: We are pleased that the reviewer thinks our analyses are sensible and our justifications and conclusions are robust, assuming the underlying data is reliable (which we hope has been established through the additional validation analyses and explanations above). Regarding Extended Data Fig. 2 these lines reflect ecoregional differences in species composition. Species composition influences what subset of expert-elicited intactness scores contribute to the BII in a given pixel, and these compositions are determined at ecoregion level (lines 606-614, 785-809). We have converted these scatterplots into boxplots to make them tidier and easier to detect differences in BII, also based on Reviewer 2's comment that the BII values we report in the text for the lowest vs highest land use intensities are not clear on the figure (note that this figure has become Extended Data Fig. 5 in the revised manuscript, due to a reordering of the results).

Referee #1 (Remarks on code availability):

Will happily do it if they come back with sensible further analyses convincing me the data are worth looking at...

RESPONSE: Thank you, we would be very happy to get feedback on this. The feedback above on the analysis has been extremely helpful in clarifying and validating our approach.

Referee #2 (Remarks to the Author):

General Comments:

Clements et al. really did tremendous work on "The biodiversity intactness of Africa assessed through local knowledge" with about 200 experts to address the current knowledge gap on less-known and rarely well-represented sub-Saharan biodiversity in global biodiversity maps and data. Hence, the study is timely, crucial for biodiversity conservation in sub-Saharan Africa, and worth publishing in this journal. The data presented, which was used in estimating the biodiversity intactness index, are original and were collected through a clear and reproducible expert elicitation

method. Further, the methods, estimates, and statistics in the main manuscript, supplementary, R scripts and the previously published bii4africa dataset paper are very clear enough and reproducible.

RESPONSE: Many thanks for the positive feedback and for recognising the timeliness of this study and its crucial importance for biodiversity conservation.

However, there are some concerns that should be addressed. First, given the disproportionately low representation of experts that are resident in nearly all the sub-Saharan Africa countries (except South Africa, as shown by the affiliations of the experts/authors on the published bii4africa dataset paper and website), I have concerns that the non-resident experts may not really know much about the history or current intactness of biodiversity in each country, ecoregion, or species group they represent during BII scoring, thereby weakening the “bottom-up” approach. For instance, an expert that had a long-term study or lived in a country over several years (like 10 years or more) but has long left the country or ecoregion or stopped working there for say 10 years or more before this data collection may overestimate or underestimate the BII. Hence, authors should add texts on how they, to a large extent, accurately identify the country/region of experience of every expert to avoid underestimating or overestimating BIIs of most countries (other than South Africa that had several people). Or authors, should add caveats to the manuscript about the challenges faced or potential weaknesses of the expert recruitment method despite their efforts to mitigate such challenges, as they did in Lines 403-406 and 635-636 for limited access to data from protected areas.

RESPONSE: Please see our above response to Reviewer 1’s comment starting “*A similar test of internal consistency in assessments by global vs local experts would also be reassuring*”, which speaks to this concern. Expertise was defined according to relevant place-based knowledge (as opposed to the need for experts to originate from these places or currently live in them). We focused on recruiting experts that were actively working in the region wherever this was possible. Many experts also worked across multiple places/countries. Experts were asked to estimate how functional groups of species are impacted by characteristic land uses, using their place-based knowledge. We then translated those estimates into a map based on the current distribution of such land uses. Thus, the experts themselves were not asked to consider how prevalent a given land use was; this was determined through mapping. It was therefore less of a concern if an expert had not worked in an area for some time, provided they had in-depth knowledge of the region, since how different land use types affect different types of species is relatively stable over time (what changes is the extend of those land uses). We have now clarified the process and criteria for selection of experts in the methods (lines 641-669). As the reviewer helpfully suggests, we have also added a caveats section to the manuscript (lines 977-1031) to ensure limitations are acknowledged and their implications considered, including reflections on the limitations of the expert selection process.

Please also see the additional validation we performed as suggested by Reviewer 1 above (lines 304-310 and Extended Data Fig. 6e in the revised paper), which demonstrates strong congruence between our expert-based estimates of intactness for 4887 vertebrate species across their sub-Saharan African ranges, and their IUCN threat statuses. This finding suggests that our expert-estimation approach was robust.

Our response to Reviewer 3’s comment starting “*The lead experts were also asked to identify subsequent experts*” addresses our efforts to include experts from across the region, and the challenges that we encountered. We provide examples for birds and primates.

Second, I do strongly believe that the title can only explicitly include the “local knowledge” if the intactness scoring truly involved the “local non-expert residents or citizens” of sub-Saharan Africa”

that live and depend on the ecosystems that house these species. The traditional ecological knowledge of locals are often passed from generation to generation and it would have added more values to the study. So, given that “non-expert locals” were missing in the intactness scoring (unless I missed it), I suggest that the title be revised to: “The biodiversity intactness of Africa assessed through experts’ local knowledge”

RESPONSE: In light of this comment, together with Reviewer 1’s second comment (please see our detailed response to that comment), Reviewer 3’s request that we specify ‘sub-Saharan’, and in keeping within character limits, we have changed the title to “*Sub-Saharan Africa’s biodiversity intactness assessed with place-based knowledge*”. We have also changed our use of the term ‘local knowledge’ to ‘place-based knowledge’ throughout the paper as outlined in the responses to Reviewer 1.

Generally, all aspects of the manuscript are well written except for the necessary edits, some rearrangements (and suggested analyses) in the result section, and, more importantly, the improvement of the discussion section that I mention in the specific comments below. I therefore recommend the manuscript for publication after the revisions I suggested here and below.

RESPONSE: We thank the reviewer for their positive feedback. We address the specific comments below.

Specific comments:

Line 55: rewrite as “Grasslands and fynbos (mediterranean-type ecosystems) suffered the greatest declines, mostly from land transformation into cropland, while deserts and shrublands have been least impacted.”

RESPONSE: We have reworded to “fynbos (mediterranean-type ecosystems)” as suggested (as well as rewording the sentence more generally to improve clarity) (lines 56-59).

Line 61: “decision-makers”

RESPONSE: We have made this suggested edit (line 59).

Line 62: Two heavy words there. Replace “credible, legitimate” with “real”

RESPONSE: We have replaced these two words with ‘appropriate’ rather than ‘real’ (line 60).

Line 70: replace “it” with “biodiversity”

RESPONSE: We have removed this part of the sentence (line 70).

Line 72: delete “to”

RESPONSE: We have made this suggested edit (line 72).

Line 85: Before the IPBES sentence, please add one more sentence here, probably as an example, to shed more light on the sentence in Line 82-85.

RESPONSE: We have added an example: ‘For example, models estimating species abundance globally often rely on extrapolation from sparse data that underrepresents data-poor bioregions, taxa, and threats’ (lines 87-89).

Line 89: “We apply this approach to assess the biodiversity intactness of ecosystems in sub-Saharan Africa, ...”

RESPONSE: We have made this suggested edit and also merged this sentence with the previous sentence (lines 96-99).

Line 107: how about replacing “sub-Saharan Africa’s” with “African”?

RESPONSE: We have revised this sentence (lines 106-110).

Line 109: how about replacing “Africa’s” with “African”?

RESPONSE: We have now removed this sentence to reduce wordcount.

Line 109: be specific with the “remaining vegetation”, please. Were you referring to the “vegetation in other continents” or “other vegetation types in Africa”?

RESPONSE: We have removed this part of the sentence to improve its clarity and succinctness (line 112).

Line 121-122: “the impact of sub-Saharan Africa’s predominant land uses and their intensities on ~50,000 species of terrestrial vertebrates and vascular plants.”

RESPONSE: We have removed “and their intensities” to improve the clarity of this sentence (line 127).

Line 124: “experts’ estimates...”

RESPONSE: We have replaced this phrase with “bii4africa expert data” (lines 130-131).

Line 125, 151 and other similar places like the text in Fig. 1, title of Fig. 3: Wherever you refer to regions within sub-Saharan Africa, which you already regarded as a region from the abstract, please consistently write “ecoregions(s)” or “biome(s)” accordingly not the ambiguous “region(s)”. For example, in Line 151, replace “vegetation types or regions” with “biomes or ecoregions”. Doing this align all texts and sections of the paper with the terms in Figures (both main and extended) and in results.

RESPONSE: Thank you for this suggestion. We have amended throughout to be more specific: sometimes we are referring to biogeographical areas, or political areas (e.g. countries), or ecoregions, and have now specified in each instance. We have used the term ‘region’ only when referring to sub-Saharan Africa which, as the reviewer says, we define as the region in the abstract.

Line 129: delete “that”

RESPONSE: We have made this suggested edit (line 135).

Line 141: is there a mistake in “magnitude more experts”?

RESPONSE: There is not a mistake, but we have rephrased to make meaning clearer (line 147).

Line 161-163: This is true, as Africans engage in hunting and consume those animals as “bush meat”. It would be good that you mention this in your discussion, and what can be done (as replacement though) to stop this hunting of non-domesticated animals in the “wild” for food.

RESPONSE: We now include this point in two paragraphs in the discussion (lines 347-353 and 393-398).

Figure 1: A very cool and interesting presentation here. Please edit the “Low-High” under the “Intensity” in the land use class legend to “Low  High” in Figure 1a: i.e., use arrow instead of hyphen, and let the “Low” and “High” be at the extreme left and right part of the box, respectively.

RESPONSE: We are glad to hear the reviewer likes the figure! We have implemented the suggested change, noting that Fig. 1a has been moved to Fig. 3a (the old Fig. 2), as suggested by the reviewer.

In the third to the last line on the first page of “Scientific Data File”: “unavailability” should replace “availability”

RESPONSE: We are unfortunately not sure what page the reviewer is referring to here. We did a search for “Scientific Data File” in our submitted manuscript and could not find this term. We would appreciate clarification so we can address this comment.

Line 195: I think the bii4africa dataset should be available as soon as the paper is accepted (by this journal or elsewhere) for reproducibility and further studies for interested.

RESPONSE: The expert-elicited data are now published open-access in a paper in *Scientific Data* (Clements et al.³ <https://www.nature.com/articles/s41597-023-02832-6>), and the reference has been updated to reflect this. The BII map will be made available open access as well once the current paper is published.

Line 198: add “(Fig. 2a)” just after the word “intactness”.

RESPONSE: We removed the first part of this sentence to reduce word count, and have included reference to the figure (which has become Fig. 3b) at the end of the sentence (lines 217-220).

Line 198: since the paragraph in Line 211-222 show the relative impacts of the predominant land uses in sub-Saharan Africa, please add either “absolute”, “independent” or individual” in between “Considering the ...” and “... impact of the region’s predominant land uses on biodiversity intactness, we” in Line 198.

RESPONSE: We have now removed this part of the sentence as part of our result reordering (lines 217-220). Based on the reviewer’s feedback that we need to be clearer on absolute vs relative impacts, we have now reordered the results to first present all the ‘absolute’ impacts (overall, per species group, per country, ecoregion, biome and land use; lines 161-238), and thereafter included a new sub-heading “Opportunities for directing conservation efforts” where we include the relative impacts (lines 241-275).

Line 199: do you really need to add “strictly” here?

RESPONSE: Yes we think that we do need this adjective (line 217), since only strictly protected areas are included in this land use category (IUCN PA categories I-III); please see explanation in the Supplementary Methods.

Line 203-204: Consider replacing this sentence (i.e., “Depending on species composition and land use intensity, the BII of a given land use pixel varies (Fig. 2a)”) with the following more direct sentence: “Our estimates also show that the BII of each of the major species groups varies between the six land use classes (Fig. 2b).”

RESPONSE: In this sentence we were aiming to explain that there is variation in BII within these land use classes (as seen by the spread of data in the boxplots in Fig. 2a – which has become Fig. 3b in the revised manuscript), and that this is caused by variation in species composition and land use intensity. We have rephrased the sentence to try make this point more clearly (lines 220-222).

Line 207: with the Fig.2b results/boxplots, the “while the opposite holds true for reptiles” appears not really a true opposite of the first part of the sentence. Please check it and reframe for clarity.

RESPONSE: We have rephrased to improve clarity (lines 223-226).

Line 212: “its total extent”. Please use exact word to replace “its”. It makes the sentence ambiguous.

RESPONSE: We have now deleted this sentence as part of our results reordering (please see our above response to your comment about Line 198).

Line 211-222: While it is very nice that the BII results, which are the main focus of this study, occupies the 1st paragraph of the results section, Fig. 1a only shows land use intensity without the BII numbers quoted in Lines 214, 217 and 219 and it is not really related to the story in Fig. 1b&c and

paragraphs 1&2 of the results section. Hence, for a better presentation and easy-to-follow flow of the results, please detach Fig. 1a from Fig. 1 and make it Fig. 2a; create a new Fig. (a box plot like the current Fig. 2a as Fig. 2b) to show the “relative” proportion of the remaining BII value (76%) that each of the 6 classes contribute. The new fig. 2b will further explain the new Fig. 2a (former 1a). With that adjustment, the current Fig. 2a becomes 2c, current 2b becomes 2d. And more importantly, please let the whole paragraph of Line 211-222 come before that of 198-209, which would show that you move from general to specific estimates.

RESPONSE: Thank you for this helpful suggestion which we have implemented in terms of both figures and text order (noting that Fig. 2 is now Fig. 3), with the exception that we do not make a new boxplot as suggested. Rather, we present this information as a scatterplot in the new Fig. 4a. Regarding your suggestion to move the whole of paragraph lines 211-222 up, please see our above response to your comment about Line 198, where we explain that we now order the results to move from absolute impacts (lines 161-238) to relative contributions (lines 241-275). This restructure is also aligned with your helpful suggestion here that we move from general to specific estimates.

Line 231: Move this sub-topic (Land use intensity impacts intactness) to the line preceding the contents in Line 211-222.

RESPONSE: We have moved this sub-heading up as suggested (line 216). Please also see our response to your above comment regarding Line 198, where we explain how we have reordered the results.

Line 233-234: please write “(Fig. 2a)” (i.e., the current Fig. 1a which I suggested above that you should make 2a) after the phrase “Considering the two most extensive land uses (croplands and unprotected untransformed lands)”.

RESPONSE: We have made this suggested edit (lines 229-231, noting that this figure is now Fig. 3a).

Line 235: please add “two” before “land uses”

RESPONSE: We have made this suggested edit (line 237).

Line 236-239: This isn’t clear- something is missing in Line 238. Seems you wanted to add “compared with 54% BII in” before the word “smallholder”? If choose my guess, remember to remove the (54%) at the of the sentence.

RESPONSE: We have rephrased to improve clarity (lines 232-234).

Line 236-244: Please consider adding extended figure(s) on the percentages quoted here. Also, consider reporting the “relative” percentages (not the absolute ones) such that the BII comparison between the land use intensities within each of the two land uses adds up to 100, as the title of extended figure 2 and histogram imply.

RESPONSE: These percentages (lines 231-234) refer to the average BII values of the lower and upper ends of the land use intensity continuum plotted in the old Extended Data Fig. 2 (now Extended Data Fig. 5) (i.e., the BII values in pixels with land use intensities of 0 to 0.1 and 0.9 to 1, respectively). Based on the reviewer’s comment that this is unclear, we have converted these scatterplots into boxplots (see Extended Data Fig. 5a-b) so that the different BII values we report in the text are clearer on the figure. We think it is important to state absolute percentages, however, given the threat of intensifying land use across the region. We have reworded the figure legend to clarify (lines 1189-1196).

Line 253: “Differences in intactness between countries, biomes, and ecoregions”

RESPONSE: This suggestion would make the sub-heading longer than that allowed by Nature, so we have rather edited to “Differences across countries, ecoregions and biomes” (line 193).

Line 254-257: And please make a sentence about those remaining 15 countries with 70-80% BII too.
RESPONSE: We have added a sentence (lines 197-198).

Given that those with less than <70% BII are relatively small in number, consider dwelling more on the likely drivers of such critically low BII score in the discussion section (and possibly with some supplementary data analysis of countries like Nigeria and Rwanda that would nail such huge biodiversity loss and trigger government and NGO's attention).

RESPONSE: We have substantially revised our discussion based on the reviewer's helpful feedback, including additional discussion of the drivers on low vs high BII scores across the region. Paragraphs 3-6 in the revised discussion address these drivers and resulting differences between countries. Notably, see lines 373-378 and 395-398.

Line 260-261: please rewrite as "Biodiversity intactness across (a) countries, (b) biomes, and (c) ecoregions³⁰ of sub-Saharan Africa. Average BII scores are shown in each map."

RESPONSE: We have made this suggested edit (lines 557-588), though we have now integrated the biome estimates into the ecoregion figure (the new Fig. 2b), based on your below suggestion for this ecoregion figure, which we have also applied to the country figure (Fig. 2a).

Given that the ecoregions are many and you couldn't insert them in Fig. 3c, I suggest you delete Fig. 3c from the manuscript, and make the "Extended Data Fig. 4" to be the main Fig. 3c.

RESPONSE: Great suggestion, thank you! We have implemented this change (in the new Fig. 2b). We think this suggestion also applies well for the country results, and have thus changed the old country map to a confidence interval plot (the new Fig. 2a). We have removed the biome map to make space for this, since the biome results are also reflected in the Extended Data Fig. 4.

Figure 3b: is there any biome particularly called "Mosaic"?

RESPONSE: There is no biome called 'Mosaic'; this term refers to areas that are a mosaic of two biomes. We now state in the figure legend (new Fig. 2) that this figure includes BII scores per "biome or biome mosaic" (line 559).

Line 266: "remaining BII are made by each biome in unprotected untransformed lands: ..."

RESPONSE: We have rephrased this sentence to improve clarity (lines 255-257).

Line 268-269: "The largest contributions of desert and fynbos to the remaining BII are in protected lands" Not so sure if my suggested revisions in Line 266, 268-269 are so clear, but I just want the authors to use direct, unambiguous sentences to convey Figure 4 as a follow-up to 3b in Line 265-269.

RESPONSE: We have rephrased to improve clarity (lines 257-260).

Line 278-279: "likely because crop production is generally unviable in these biomes (Fig. 4)."- so lack of cultivation could drive such poor BII in deserts and shrublands? I think that is a very unlikely speculation, unless you have evidences.

RESPONSE: We did not mean to speculate, but rather to point out that cropland extent was limited in these arid biomes. We have now deleted this sentence to reduce wordcount.

Line 280: list the grassy biomes in parentheses just after the word grassy biomes.

RESPONSE: We have added: "(grassland, acacia savanna and humid savanna)" (line 261).

Line 283-284: this ", together with deforestation for rangelands, croplands and timber plantations" does not complement to the preceding clause in the sentence.

RESPONSE: We have deleted this text and replaced with a standalone sentence: *“Deforestation for rangelands and croplands also contribute to losses in the forest biome”* (lines 265-266).

Line 285: list the five biomes in parentheses for clarity

RESPONSE: We have made this suggested edit (line 212).

Line 294: see the comment on this Fig. 3c above

RESPONSE: We have addressed this in our earlier response (we have replaced the old Fig. 3c – now Fig. 2b - with what was previously Extended Data Fig. 4).

Line 310: delete “species”

RESPONSE: We have rephrased as *“Percentage plant contribution (as opposed to vertebrates) to total species richness per ecoregion”* (lines 1178-1179).

Line 317: “make use of here” “use here”

RESPONSE: We have revised this sentence to improve clarity (lines 279-281).

Line 319-320: Not so clear.

RESPONSE: We have revised this sentence to improve clarity (lines 279-281).

Line 333: do you really meant to write “average mean”?

RESPONSE: This repetition was a mistake; we have reworded this figure heading (lines 1230-1233).

Line 349: Consider adding comma sign before and after the “for example”

RESPONSE: We have made this suggested edit (line 295).

Line 352-355: This information appears not related with the preceding sentences and the story on “The validity of this biodiversity assessment” and Extended Fig. 6a-c. Maybe you need to create a premise for it or relocate it to a more pleasing spot in this paragraph.

RESPONSE: We have added a premise (lines 298-300).

Line 367: why don't you just say “200 experts” instead of “several hundred”?

RESPONSE: We have made this suggested edit (line 325).

Line 373: it would be nice that you briefly recap here that the remaining overall BII of sub-Saharan Africa is 76%.

RESPONSE: We have added the following sentence: *“Our results show that Sub-Saharan Africa has lost just under a quarter of its pre-industrial biodiversity intactness.”* (lines 332-333).

Line 375: Based on my comment on Fig. 1 and on Line 211-222, the referenced “(Fig. 1) may not be Fig. 1 (alone or again).

RESPONSE: We have ensured all figure references were revised appropriately.

Line 375-377: Absolutely! That's very important.

RESPONSE: We are pleased that the reviewer agrees with this key takeaway message of the paper!

Discussion section: Despite that the results section was full of exhaustive and interrelated data analyses and that the discussion section should mirror the structure of the results section in flow for ease of understanding and linkage of the findings, it is strange that about 80% or more of the discussion section (especially from 412-439) repetitively (in multiple, different sentences) only summarily speculate (without data comparisons between the current findings and the previous) the

potential benefits of the approaches and data used for this study. Hence, a vast portion of the intriguing findings in the study was left undiscussed — especially in relation with existing studies' reported BII values for sub-Saharan Africa; the potential (literature-supported) drivers of the BII values reported for across and within biomes, ecoregions, and countries in sub-Saharan Africa; whether there were over estimation or underestimation, and why this study potentially provide a more accurate BIIs etc. For example, authors should discuss the underlying drivers of the huge (\geq ~50%) decline in BII in West Africa and South-eastern Africa countries.

RESPONSE: Thank you for this helpful suggestion, we agree we have missed an opportunity to dig deeper into some of the results. We have substantially revised our discussion section. In summary, we have expanded our discussion of the major drivers of BII trends across the region, drawing on previous literature (lines 345-398). We comment on countries with particularly low vs high BII (and what we can learn from the latter) (lines 346-347, 359-361, 374-378, 395-398). We also compare our results with those of previous BII estimates for the region (highlighting findings from the global model that are incongruent with our findings and previous literature) (Figure 7, lines 312-321, 353-356, 374-378). We have articulated the relevance of our findings to policy (lines 339-343, 352-353, 363-371, 384-388, 393-414). To make space for these new paragraphs, we have eliminated repetition as suggested by the reviewer.

Line 410: “megfaunal” -> “megafaunal”.

RESPONSE: This sentence has been removed in the new discussion.

Line 465-466: “in such cases” or “in all cases”? I looked through the methods but couldn't find if any other BII formular apart from Equation 1 exist for cases without data scarcity. Reading through 499-505, I later found that the “in such cases” only apply to other than large mammals species. Please show and illustrate the equation for the “ideal situation” first (i.e., the situation you expressed in Line 502-505) before the “data scarcity situation”, which the current equation 1 displays. If I am wrong, please make the texts here clearer for readers like me.

RESPONSE: We now clarify in this first section of the methods (lines 602-623) by presenting two equations for the pixel-level BII scores: the ‘ideal situation’ (new equation 1; not applicable here) and the data-scarcity situation (new equation 2; applicable here). We also make note of the large mammal variation for equation 2 (lines 704-707).

Line 465,469: “region (commonly, the ecoregion)”- please directly and simply say “ecoregion” whenever you meant “ecoregion” and limit the use of “region” to “sub-Saharan Africa” throughout the manuscript.

RESPONSE: We have implemented this suggestion throughout (see our above response to the reviewer's similar comment).

Line 512: how finer can a patch or scale would be for it to be regarded as “finer”? Please state the specific spatial scale that was regarded as being finer.

RESPONSE: We have added “(<1 km²)” (line 717).

Line 525: simply replace “biogeographical vegetation types” with “biomes” as you did name them in next sentence and in Fig. 3b.

RESPONSE: We have implemented this suggested edit (line 760).

Line 544: replace “regions” with “ecoregions and biomes”

RESPONSE: We have implemented this suggested edit (line 784).

Line 603: Please cite the “Extended Data Table 2” after the nitrogen input inside the parentheses.

RESPONSE: We have implemented this suggested edit (line 847).

Line 617, 618, 619: “herbivore biomass capacity” and “herbivore biomass” are not that correct to be used as terminologies. You meant “herbaceous biomass”, right? If yes, in Line 617, change “herbivore biomass capacity” to “herbaceous biomass (i.e., forage for herbivores)”; in Line 618, change “sustainable herbivore biomass” to “biomass declines”, and in Line 619 just say “biomass”. Replace “. In addition” with “and” in Line 618 to merge the 2 sentences.

RESPONSE: We did not mean ‘herbaceous biomass’, we meant ‘potential African herbivore biomass’. We have rephrased to make this clearer (lines 851-855).

Line 619: Please add “Hence,” before “we clustered...”

RESPONSE: We have implemented this suggested edit (line 855).

Line 637: The latter were considered

RESPONSE: We have implemented this suggested edit (line 873).

Line 642: use multiplication sign for dimensions not x

RESPONSE: We have implemented this suggested edit (line 878).

Line 650: “sub-Saharan Africa region”

RESPONSE: We have implemented this suggested edit (line 886).

Line 649-650: Think this

RESPONSE: It is unfortunately unclear to us what is being suggested here.

Line 656: replace # with No. throughout the table

RESPONSE: We have implemented this suggested edit (Extended Data Table 2).

Line 681: change x to multiplication sign

RESPONSE: We have implemented this suggested edit (line 916).

Line 700-710: Not sure this is different from the preceding sentence. Moreso, “reasonable correlation” is subjective.

RESPONSE: We have merged these sentences and removed the comment about ‘reasonable correlation’, which is not necessary (lines 935-936).

Line 703: Extended Data Fig. 6. shows “across ecoregions” correlations not “per ecoregion”

RESPONSE: We have replaced ‘per’ with ‘across’ as suggested (line 938).

Line 826: But there are references mentioned in the Methods with serial numbers before this 51st reference.

RESPONSE: We have ensured referencing follows Nature’s formatting guidelines.

Referee #2 (Remarks on code availability):

Yes, the code provided via the <https://figshare.com/s/00168eb685728c0be9d3>, guided by a self-explanatory code lines and README run well.

RESPONSE: Thank you for the positive feedback.

Referee #3 (Remarks to the Author):

Thank you for your submitted article, which involves providing biodiversity indices for policy in contexts of data paucity. Specifically, the use of expert input to evaluate sub-Saharan Africa's biodiversity intactness. Since my expertise is in expert elicitation and sustainable development, I will focus my review of those aspects (methods and claims made). This also corresponds with what the editor asked me to focus on.

RESPONSE: We thank the reviewer for their expert input in flagging the key aspects of our elicitation that require elaboration, justification and reflection on limitations.

First, I should point out that my initial submitted review did not account for the fact that most of the expert elicitation is presented in an accompanying article in Scientific Data. I am not used to having methods described in accompanying articles. Personally I lament that some of the most widely read journals tend to de-emphasize the presentation of methods (such as at the end of articles), but regardless I would recommend that the authors more clearly identify that the elicitation methods can be found in the accompanying article. When I read this piece it read to me as if the authors point to the other article as a different study using a similar methodology.

RESPONSE: Thank you for the feedback on the confusion you experienced. Based on this feedback, we have made it clearer that the expert-elicited dataset that we use has been described in a stand-alone published paper (Clements et al.³) (lines 119-122, 628-632 and 636-369).

We decided to publish the expert elicitation methods and intactness score dataset as a standalone data paper because (1) we felt it was important that the contributions of all experts to this 'bii4africa' dataset were fairly recognised through co-authorship; and (2) the dataset has many uses beyond mapping BII (as summarised in Table 3 of the paper: <https://www.nature.com/articles/s41597-023-02832-6/tables/3>). Our intention was not to de-emphasize the elicitation method but rather the opposite: to give it full attention in its own paper so that there would be sufficient space to describe the method in detail. However, we take note of the fact that a more detailed summary is clearly needed in the current paper, and hope our additions address that shortcoming.

In fact, I believe that since expert elicitation is so prominent in the presentation of this study, the authors should provide much more details of the methods in the article itself (or at least in the supplement).

RESPONSE: We have considerably expanded the 'Intactness scores' section of the methods to provide more details on the elicitation process, aimed particularly at addressing the specific comments laid out below and by the other reviewers. We have also added an extended figure (the new Extended Data Fig. 1) that summarises our approach, including the expert elicitation process (with an example for birds), as requested by Reviewer 1.

I think the topic is a very important and difficult effort, especially to do well. However, I have some concerns about the elicitation done, and also the framing of the piece in relation to development.

RESPONSE: Thank you for the positive feedback. We have addressed your specific concerns on the elicitation and framing in our point-by-point responses below.

Expert elicitation, when done credibly, is a very meticulous process of identifying expertise and regulating the kinds of information and intuition you want to elicit while keeping the various biases and cognitive issues that can degrade data quality at bay. You also want to make sure that information acquired from experts reflects the expertise you seek, and is not ultimately controlled

and shaped by the analysts that ask the questions and synthesize the results. I think at broad levels the authors adopt a good structure, but I have some reservations in the specifics in the identification, training and structure of the elicitation process. I also provided important references (which are a brief selection of articles in entire literatures on the topics).

RESPONSE: We adopted the IDEA protocol for structured expert elicitation because it is a published approach¹⁰ that draws on a large body of literature on expert elicitation (including many of the authors and useful references that the reviewer shares with us below), aimed at mitigating potential shortcomings of expert elicitation. We have added text to the ‘Intactness scores’ section of the methods to elaborate on this protocol (lines 673-748). We are pleased to hear that the reviewer thinks we adopted a good expert elicitation structure, and we respond to the specific reservations below.

I appreciate the description of experts chosen for the elicitation. When I first read the paper boast 200 experts my immediate reaction was “that’s too many to handle effectively”. However, I think the substructuring of expert teams into groups of 20 is well done (though I would say 20 is on the high end of good elicitation methods). The authors also demonstrate relevant information showing some dimensions of variation in experts involved in the process.

RESPONSE: We are pleased to hear that the reviewer thinks our sub-structuring of experts into smaller groups was a good approach.

However, I have some hesitation with the treatment of “lead experts” and their role in the further selection of experts I’m hoping the authors can clarify, expand on, or reflect on in their draft. Selection of lead experts is vaguely, “identified based on their relevant expertise”. I will note that expert status (as determined by credentials and markers of prestige) often does not have any bearing on performance or accuracy of expert input, and can even be counterproductive.

RESPONSE: We agree that status alone is not a good way to identify experts, as reflected by our methods in Clements et al.³: “A broad definition of expertise was used to identify experts, centred on experience of how sub-Saharan species are impacted by human land uses. Diverse types of people can have such experience (e.g., researchers, field or tour guides, park rangers, conservation practitioners, museum curators, and consultants), and inclusion was thus not limited to specific qualifications or institutional affiliations.” Lead experts needed to meet this definition of relevant expertise (which was not based on prestige), as well as being willing to take on the task of recruiting experts and have access to an existing network of experts, or else the willingness to develop such a network. We now explain this in the paper methods (lines 641-645 and 648-650). Such a network was important given the challenging task of identifying experts across the continent and convincing them to give of their time to participate. For example, several lead experts serve in the IUCN Species Survival Commission working groups and other regional networks (e.g., <https://ascaris.org/>; <https://www.birdlife.org/our-partners-africa/>), noting that these are not all exclusive scientist networks (e.g., Birdlife partners).

The lead experts were also asked to identify subsequent experts. This kind of snowball approach to identifying experts is not necessarily problematic but can introduce issues of similarity biases (“groupthink”). Notably, focusing on experts from scientific initiatives is likely to leave out experts with local knowledge not often valorized in official and scientific working groups. Did the authors do any calibration to see if they collected a sufficiently broad range of expertise? The total number of experts may not represent knowledge breadth if experts are from similar training and collegial working relationships.

RESPONSE: We put significant effort into including a diverse group of experts for each taxonomic group. The experts did not need to be trained in a certain way to meet our definition of an expert, and lead experts were briefed to find and reach out to people that met the definition of expertise quoted above, focusing on diversity of expertise geographically, taxonomically, professionally, as well as aiming to include as many African experts as possible (these different dimensions of diversity within the expert group are depicted in Clements et al.³ Figure 2 <https://www.nature.com/articles/s41597-023-02832-6/figures/2>). We had to balance the need for multiple dimensions of diversity with the need for an appropriate number of experts to ensure each elicitation worked well (as the reviewer mentions, even 20 is on the upper end of what is recommended). We now elaborate on the role of the lead expert in the methods (lines 650-658).

In reality, finding experts and securing their participation was a challenging task. For example, our lead bird expert sent 118 invitation emails to people and organisations across 36 of sub-Saharan Africa's 42 countries. Bird clubs, societies and NGOs were contacted (e.g., 21 Bird Clubs and 10 Birdlife country offices), as well as researchers, extension officers, local bird guides and tour companies. This resulted in 21 participating bird experts. This 80% non-response rate may be due to digital inequities, outdated information on websites, lack of incentive for non-researchers to participate in research, language barriers (particularly in Francophone countries), etc. Similarly, our lead primate expert sent out 65 invitation emails to people and organisations across 15 sub-Saharan African countries, with 24 ultimately participating. We have included these numbers for birds in our new extended figure (Extended Data Fig. 1b) that provides an example of the process.

We have also added text to the methods section reflecting on the challenges of recruiting experts (lines 660-669). We have also added a new 'Caveats' section to the end of the methods, where we reflect on the limitations of our expert selection (lines 978-999).

Choosing diverse experts is analogous to increasing the "degrees of freedom" in a dataset – and asking similarly trained experts and colleagues will severely degrade the independence between experts and be akin to pseudosampling. See the following references for discussions on expert selection and elicitation processes

Burgman, M. 2005. Risks and decisions for conservation and environmental management. Cambridge University Press.

Burgman, M. A., M. McBride, R. Ashton, A. Speirs-Bridge, L. Flander, B. Wintle, F. Fidler, L. Rumpff, and C. Twardy. 2011b. Expert status and performance. PLoS One 6:e22998.

Singh GG, Sinner J, Ellis J, Kandlikar M, Halpern BS, et al. (2017) Correction: Group elicitation yields more consistent, yet more uncertain experts in understanding risks to ecosystem services in New Zealand bays. PLOS ONE 12(12): e0190326

Tetlock, P. E. (2005). Expert Political Judgment How Good Is It? How Can We Know?, Princeton University Press.

-While this book covers experts of social science, it is one of the best works documenting the limitations of expert status. It also largely corroborates the work Mark Burgman showed in conservation scientists.

RESPONSE: Thank you for sharing these references. Burgman et al. 2011 and Singh et al. 2017 both find that (to quote Burgman) *"if experts are given the opportunity to listen to one another, assess other judgements, and cross examine reasoning and data within a structured process, their average performance improves substantially. Additionally, the averages of a group's independent best guesses following discussion generally perform at least as well as, and often much better than the estimates of the best-regarded person in the group."* The IDEA protocol¹⁰ that we used incorporates this research aimed at improving expert performance into its design, with experts

given the opportunity to listen, assess and cross-examine during a discussion meeting, followed by a second round of independent estimation. Thus, in addition to aiming to select a diverse group of experts as outlined above, our elicitation process was guided by the evidence on how to improve the performance of the expert group. We have notably expanded the 'Intactness score' section of the methods to elaborate on this protocol and how it aims to promote independence between experts through independent estimation prior (and subsequent) to group discussion, as recommended by Burgman et al.¹¹, Singh et al.¹², and Morgan¹³ (lines 673-682, 722-748).

I would suggest that one of the best ways to utilize a snowball approach to address knowledge breadth is to snowball across experts (and not just the "lead" expert) and track the "network" of experts to see if the expert pool saturates (or includes the majority) of potential experts. This process is probably modeled best in the following paper:

Ban, S. S., R. L. Pressey and N. A. J. Graham (2014). "Assessing interactions of multiple stressors when data are limited: A Bayesian belief network applied to coral reefs." *Global Environmental Change* 27: 64-72.

RESPONSE: We note that Ban et al. 2014 used scientific publications and 'top-cited' papers to identify experts and then snowballed based on those experts. By contrast, in line with what the reviewer recommends, our approach as explained above aimed to look beyond 'published' experts to include a wider diversity of field-based experts, for example working as tour guides or rangers. As mentioned above, we now clarify the role of lead experts in the paper and that they *"...facilitated a snowballing approach among experts where needed to increase diversity and numbers."* (lines 654-661).

While we understand the value of reaching saturation, it would have been impractical for this study, as it would have been very difficult to know how large the total pool of experts was (in contrast to the cited paper where the pool was defined according to the published literature, making it possible to identify saturation). It was (a) extremely challenging to find all these experts (the lead author and lead experts spent several weeks each trying to track down experts, find current emails, etc - see bird and primate examples above) and (b) we would then have potentially ended up with a larger number of experts than can be handled effectively in an elicitation. In addition, some taxonomic groups have many more experts than others, and for many taxonomic groups the total pool of experts is heavily biased towards South Africa. Instead, we opted for an approach where the lead expert was specifically tasked with sourcing a diverse set of ~20 experts, across geographic regions, professions and taxa. We contend this 1) created consistency across the approaches used by the different leads, and 2) most likely led to a more diverse group of experts than selecting a random sample from a saturated group of experts. As explained in our response to the reviewer's above comment, we now include details of the expert selection process and rationale in the paper methods (lines 641-669).

If the authors did not take this quality assurance step in their methods, I would encourage them to consider the implications of their results. Are there hidden uncertainties in their results? At worst, if there are significant selection biases in their experts the uncertainties would be systematic (biased) and the large sample of experts may actually reinforce the bias rather than reduce uncertainty.

RESPONSE: We hope that we have addressed the reviewer's concerns about our expert selection process in response to the earlier comments. We have expanded the methods 'Intactness scores' section to explain how we aimed to mitigate potential biases in our expert selection by focusing on selecting a diversity of experts (lines 641-669) and in our elicitation process by using the IDEA protocol (lines 673-682, 722-748). We have also added a 'Caveats' section to the paper where we reflect on the risk of systemic biases arising from our expert group (lines 978-999).

Please also see our new independent validation test using the IUCN Red List (Extended Data Fig. 6e, lines 304-310), in response to Reviewer 1. We believe that the results of this test also help show that limitations in our expert selection process did not lead to systematic biases that compromised the robustness of our results.

There is also very little information on how experts were elicited to quantify their BII scores. There is much research to indicate that experts can more reliably provide ranges of estimates than point scores, and even then it is worth “nudging” experts by questioning their certainty to see what scores they provide.

Morgan, M. G. (2014). "Use (and abuse) of expert elicitation in support of decision making for public policy." *Proceedings of the National Academy of Sciences* 111(20): 7176-7184.

Speirs-Bridge A, Fidler F, McBride M, Flander L, Cumming G, Burgman M. Reducing overconfidence in the interval judgments of experts. *Risk Analysis*. 2009;30(3):512–23. Pmid:20030766

RESPONSE: We considered (and developed questionnaires) for a range estimate approach but ultimately opted for the point estimate approach after trialling the approaches on lead experts, since it was found to be cumbersome and confusing for experts to provide ranges for numerous species groups and land uses. Importantly, however, we did nudge experts by questioning their certainty both in the estimation process and in the discussion meeting. We have added these details to the methods (lines 725-728 and 733-745). We have also noted the limitations of the point-based approach in the new ‘Caveats’ section, reflecting on the risk that it led to expert overconfidence, and how we mitigated this through nudging (lines 991-997).

Please also see the results of our new independent validation test (Extended Data Fig. 6e, lines 304-310), which suggests that our eliciting of point estimates, combined with nudging, led to robust results.

I would encourage the authors to justify how species groupings were done relative to expert judgement. I understand the ecological justification for species groups, but it would speak more to data quality to relate these decisions to expert familiarity. I understand the pragmatic necessity to group species given the vast numbers of species that are being addressed, but how species are grouped will affect expert scores. If the groups are made in such a way that experts are not familiar with a grouping or thinking about a group of species in the way identified, the quality of the scores will likely be questionable. I would extend the same comment to the grouping by biomes. Again it may make ecological sense, but the authors should relate the groupings to expert familiarity to help assess data quality.

RESPONSE: We agree that it was imperative that the experts understood the groupings. This was part of the reason why the species grouping exercise also included the participating experts. Lead experts presented the draft groups to the experts during the introductory meeting and facilitated a discussion where experts gave input into these groupings and collaboratively revised them during the meeting. We now explain this process in the paper methods (lines 700-701). We also show this step on our new Extended Data Fig. 1. Experts were then provided with details of the groupings in the survey spreadsheet where they entered their estimates (e.g., including example species or full lists of species, species attributes such as fossorial or rupicolous, etc; we now clarify in the paper methods that “*The final groups are summarised in the ‘Sp_Groups’ tab of the bii4africa dataset*” (lines 701-702). Experts were also provided with a map of the biomes in the introductory meeting, and one biome name was revised based on expert feedback during the

meeting ('humid' was added to Caesalpinoid-miombo savanna). We now clarify in the paper methods that *“These biomes were proposed by the lead plant experts and refined based on feedback from participating experts during the elicitation introductory meeting”* (lines 762-763).

Please also see the results of our new independent validation test (Extended Data Fig. 6e, lines 304-310). The robustness of the results across species suggest that groupings were consistently understood by experts.

The author description of identifying and describing land uses shows built-in training in the elicitation process, which is great, but I found the writing of the section confusing because of the constant switch between what was done and why other scale considerations would not be as good. First, in asking experts to think of “average” landscapes based on visual cues of land uses and descriptions – people are prone to the “availability bias” when trying to imagine representatives, which may vary considerably across people based on their experiences. Perhaps that was part of the design and the aggregation stage was meant to address this, but I think in communicating the methods the authors should address how they dealt with these cognitive biases.

RESPONSE: We are glad that the reviewer supports our approach. The reviewer is correct that it was part of the design to ask experts to base their estimates of land use impacts on their experience in representative landscapes. These estimates were then averaged across experts to address the availability bias of each individual expert. Importantly, our use of confidence intervals reflects the uncertainty/variation caused by availability bias. Availability bias would only be a problem if we expected all experts to hold knowledge for the same subset of representative landscapes. Given the diversity of experts’ experiences across different geographies (as outlined in response to the reviewer’s earlier comment), we have no reason to expect this to be the case. We have added these details to our methods (lines 728-731 and 774-782).

Please also see the results of our new independent validation test (Extended Data Fig. 6e, lines 304-310). The robustness of the results across species (including rare and localised species and species where there are few experts) suggest that availability biases did not substantively bias the results.

My more serious concern here is that while experts were asked to consider integrated impact of all characteristics of the landscape, it seems that there is potential for experts to be considering different problems (because different experts may be identifying different impacts). This so-called “linguistic uncertainty” whereby experts may translate “all characteristics” differently can lead to underappreciated uncertainty and instability between experts.

Regan, H.M., Colyvan, M. and Burgman, M.A. (2002), A TAXONOMY AND TREATMENT OF UNCERTAINTY FOR ECOLOGY AND CONSERVATION BIOLOGY. *Ecological Applications*, 12: 618-628. [https://doi.org/10.1890/1051-0761\(2002\)012\[0618:ATATOU\]2.0.CO;2](https://doi.org/10.1890/1051-0761(2002)012[0618:ATATOU]2.0.CO;2)

RESPONSE: We agree there is room for variability in interpretation of landscape impacts. We aimed to reduce this through the detailed descriptions and representative images for each land use, as the reviewer mentions above, in an effort for experts to have as clear a picture as possible of the land use in question. In addition, the introduction and discussion meetings and comments section for each estimate in the survey were intended to identify and reduce this uncertainty, as we now explain in the methods (lines 725-728 and 735-737).

While this process reduced uncertainty, reflected by the reduced standard errors around expert estimates in the second, compared with the first, round of the expert elicitation (see validation

section in Clements et al.³), we could not completely eliminate this kind of uncertainty. In contrast, some variability among experts' estimates demonstrates that they are applying their place-based knowledge and helps capture the variation in particular land use categories across the region, which is an important element of the approach. This variability is captured in the confidence intervals that we report in the results, as now elaborated on in our methods (lines 777-782).

Please also see our new independent validation test (Extended Data Fig. 6e, lines 304-310), suggesting linguistic uncertainty did not invalidate the process.

I think the IDEA process is a good structure for elicitation, especially with iterative feedback. I do however have some comments on some of the details within the process presented. There is much research to indicate that experts can more reliably provide ranges of estimates than point scores, and even then it is worth "nudging" experts by questioning their certainty to see what scores they provide. The issues I present on the (lack of) details presented in how experts were elicited extends to the construction of the decision-tree used for the mapping – who was involved and how was this done? Finally, I note the extreme range of the numbers of experts involved in different groups of species. I understand the need for pragmatism (again) but the authors need to consider the range in reliability and robustness of responses as a result of this.

Morgan, M. G. (2014). "Use (and abuse) of expert elicitation in support of decision making for public policy." *Proceedings of the National Academy of Sciences* 111(20): 7176-7184.

Speirs-Bridge A, Fidler F, McBride M, Flander L, Cumming G, Burgman M. Reducing overconfidence in the interval judgments of experts. *Risk Analysis*. 2009;30(3):512–23. pmid:20030766

RESPONSE: We have addressed the query on ranges and nudging in response to the reviewer's earlier comment, clarifying the additional details we have added to the methods section regarding the elicitation approach and reasoning behind it. We have also reflected on the limitations to our approach in a new 'Caveats' section of the methods, as highlighted above.

Regarding the decision-tree used for the mapping, we made use of an established, widely cited classification algorithm (e.g., Ellis et al.^{14,15}; Klein Goldewijk et al.¹⁶) that uses population density and land use area to assign pixels to categorical land use classes, reflecting the differences between the land uses described to experts. The assigning of intensity estimates within these land uses also used a protocol comparable with other highly cited human intensity indices (see review by Watson et al.¹⁷), selecting variables appropriate to the regional context and aligning with the land use descriptions given the experts (e.g., of relevance for croplands is whether they are smallholder or commercial, together with the extent of croplands and nitrogen input intensities). The identification of intensity variables and thresholds was done by the author team (which includes all lead experts), through a series of online discussions and pilot maps.

We have now elaborated on this in the 'Mapping land uses' section of the methods (lines 823-826 and 834-838).

Regarding variability in expert numbers per elicitation, as the reviewer mentions there was a need for pragmatism, and it was difficult to land on identical numbers of experts providing estimates for each group of species in each land use. We make use of confidence intervals to account for this variability/uncertainty, since sample size is embedded in these intervals (with lower confidence / large intervals for estimates that had fewer contributing experts). In the methods 'Intactness scores' section we now explain this (lines 777-782).

I have made some mention of this already, but I am concerned about the extent to which results actually reflect experts and not the analysts. The expert input feed into a pre-defined notion of what

biodiversity intactness is and is only part of equations to calculate this. Was there any work to engage experts to see if the definitions, standards, and reference points of the analysis correspond with how the experts think of the problem? Because of the bounding of the analysis, have the authors considered that the results are more affected by the pre-defined equations rather than the expert input? For a study showcasing expert input, I find the lack of engagement with these very fundamental issues troubling.

Aven, T. and S. Guikema. 2011. Whose uncertainty assessments (probability distributions) does a risk assessment report: the analysts' or the experts'? Reliability Engineering & System Safety 96:1257-1262.

RESPONSE: The reviewer is correct that we made use of the Biodiversity Intactness Index, which has a published definition and equation⁸, and is bounded in nature. The reviewer is also correct that the expert-elicited intactness scores are only part of the BII equation, but they are the major part; the other parts are to enable the mapping of these scores: maps of species composition (based on well-established IUCN range maps) and land uses (which align with the land uses discussed with experts and are mapped based on published approaches). While we appreciate that the bounded nature of the assessment had limitations, we also think that such bounding was necessary for our aim, which was not to co-produce a definition of biodiversity intactness for the region, but rather to use and embed place-based knowledge in a well-established index that has been recognized for its theoretical and practical strengths.

We agree fully with the reviewer that critical to the validity of this approach was that experts understood the predefined notion of biodiversity intactness and approach to estimating it. In the invitation email and introductory meeting, we explained the index (both conceptually and how we would calculate it informed by expert-elicited intactness scores), and experts had the opportunity to ask questions or voice concerns. Experts generally asked clarifying questions, and no major concerns were raised. As noted in the paper methods (lines 704-707), the introductory meeting of the large mammal elicitation did actually result in a change to the approach (to elicit at species level instead of response group level), based on expert feedback that they had the necessary knowledge at this level (and experts then suggested additional experts to plug species-level gaps). The discussion meeting provided another opportunity to voice challenges and concerns, with these largely focused on expert uncertainties around certain estimates they had given and wanting to talk through those. Experts also had the option to choose not to participate should they feel that such an approach did not resonate with them and/or that they were not able to contribute meaningfully. Of 266 experts that agreed to participate in the elicitation, 66 (25%) did not see the process through to completion (lines 680-682). Of those that gave a reason, the vast majority stated time limitations, while a handful stated that they did not feel they had the necessary expertise. We now clarify these steps taken to ensure that experts understood the wider process they were contributing to, and were comfortable with that, in the methods (lines 676-680 and 737-739).

We also now clarify in the methods section which aspects of the process are informed by (1) the literature (the BII definition and equation – lines 585-600, land use mapping algorithm – line 823); (2) consultation with lead experts (the land use class descriptions and variables – lines 709-711 and 835-838); (3) consultation with participating experts (the species functional response groups – lines 700-701, the biomes – lines 762-763); and (4) the structured expert elicitation (intactness scores – lines 626-639). We reflect on the limitations of this bounded approach in the new 'Caveats' section of the paper, drawing on the literature that the reviewer has suggested (lines 1001-1010).

Beyond the expert elicitation, I also have issues with the framing of the paper and some more quantitative analysis. I go into the specifics below

Title – since this assessment is for sub-Saharan Africa, the title should reflect that and not suggest that it is for the whole continent. This is to address both accuracy issues as well as avoid the often-made conflation of “Africa” as some homogenous place instead of diverse continent.

In light of this comment, together with Reviewer 1’s first comment (please see our detailed response to that comment) and Reviewer 2’s related comment, and in keeping within Nature’s character limit for titles, we suggest changing the title to “*Sub-Saharan Africa’s biodiversity intactness assessed with place-based knowledge*”

Line 66 – The links between biodiversity, and any specific status of the environment, with global sustainable development, are tenuous and not at all settled. The author refer to literature on the Planetary Boundaries, but this framework has been heavily criticized throughout its history, and perhaps most aggressively on issues of biodiversity. I know that the planetary boundaries authors have replied to many challenges, but in my view (and many others in development communities, and throughout many science disciplines) the responses do not address the severe challenges and problems raised. I am not suggesting the authors retract their statements, but they must make them in context of the unsettled nature of the debates if raised.

Biermann F, Kim RE. The boundaries of the planetary boundary framework: a critical appraisal of approaches to define a “safe operating space” for humanity. *Annu Rev Env Resour.* 2020;45:497–521.

Montoya, J. M., I. Donohue and S. L. Pimm (2018). "Planetary Boundaries for Biodiversity: Implausible Science, Pernicious Policies." *Trends in Ecology & Evolution* 33(2): 71-73.

You do not need to overextend the importance of biodiversity to make the case for this study. Or if you do, it should be done fairly, while pointing to the variety of evidence that disagrees with these kinds of statements. Highlighting the local and regional importance of biodiversity is likely enough to point out, and this will be more accurate and less contentious

RESPONSE: We have rephrased this introductory sentence (line 66) and similar statements in the manuscript to rather emphasise that retaining biodiversity has become an integral part of sustainable development agendas (as opposed to stating that it is the bedrock of sustainable development itself).

We have added text noting the contested nature of the planetary boundaries (lines 143-144), citing the references provided by the reviewer. We also now note that our approach aligns with suggestions to address some of the challenges with the biosphere integrity boundary (lines 144-146).

Lines 113 -The authors make the point that there are limitations of top-down approaches to understanding ecosystems and points to the novelty of their approach in consulting local experts. However, it’s not clear to what degree the authors have considered that the structure of their analysis is very top-down: thinking of biodiversity intactness in reference to pre-industrial society is a subjective standard that historically largely follows European notions of nature. Have the authors looked to see if the experts they consult follow these notions? These kinds of questions are important because if the experts do not think on these terms, then the responses they provide may

not be as robust. That is, if questions are posed in ways that experts are not familiar with, they may not be accurately considered experts.

RESPONSE: We appreciate that the bounded nature of our approach is top-down to some extent, but part of the challenge we aimed to overcome was how to integrate more ‘bottom-up’ (place-based) knowledge into a large-scale regional product that speaks to the needs of national and international decision-making, as opposed to the dominant approach of using outputs from global data-driven models that are far less appreciative of context. If we had not adopted a structured and bounded approach based on an established index with a defined reference state, we think such a task would have been unmanageable. We now elaborate in the introduction (lines 77-82 and 96-101). As stated in response to the reviewer’s earlier comment, we have reflected on the bounded and structured nature of our approach in a new ‘Caveats’ section (lines 1001-1010).

A measure of intactness/integrity is by nature a relative measure. The Global Biodiversity Framework defines the notion as *“The degree to which the ecosystem’s composition, structure and function resemble those characteristic of its natural range of variation, which may be defined from historical or minimally disturbed reference states, replicated contemporary samples, ecosystem models and/or expert judgement.”* We agree with the reviewer that this is a particular way of thinking about the state of biodiversity, and that science in general and ecology in particular are dominated by Western ways of thinking¹⁸. Here we are countering this Western dominance by bringing more African expertise into ecology and adding more insight around important aspects of local and regional context that are often ignored (e.g., the difference between smallholder vs commercial croplands), while still using notions and measures that are embedded in international agreements such as the Global Biodiversity Framework that countries are required to report to. We recognise the limitations of this approach, but believe it is an important first step. We note in the paper that the reference state we made use of was *“before alteration by modern (industrialised, colonial and post-colonial) society”* and *“In most parts of sub-Saharan Africa, this corresponds to populations before the substantial alteration of the landscape triggered by colonial settlement”* (lines 593-596). Such a reference, importantly, includes Indigenous people as part of this intact state (pushing back against Western notions of wilderness), and recognises the impact of both colonisation and industrialisation on this state.

The index and its reference were presented to experts in the introductory meetings. We have added text to the methods to clarify this (lines 676-679). While no major objections to either the chosen reference or the concept of a reference were raised, neither would have eroded the quality of the assessment provided experts understood the concept and the reference to use. We endeavoured to provide this understanding in the introductory meeting and written instructions. The consistency in expert scoring, and the robustness of the results compared with the IUCN Red List indicate that this understanding was sufficient.

Results –

Some results seems rather tautological. Since the BII is defined as intact nature relative to preindustrial settings, this measure shares the same information with indices about human activity and human footprint. I understand these relationships are partially meant to validate the measure, but here I have some questions as well. First, given the shared information, it’s not really surprising that the best relationship seems to be between the BII and the Human Modification Index compared with the biomass modification and habitat indices. Perhaps this shows some evidence of predictive validity of the measure. However, it speaks to questions about the potential construct validity of the measure: since it does a better job tracking human footprint than it does biomass and other

ecological variables, is it tracking what we want it to? I think questions about the tautological nature of the validation, and a broader discussion of validity is needed.

RESPONSE: We agree it is not surprising that BII and other measures of human activity and habitat/biomass are correlated. What IS surprising is that the global BII model does not show this expected correlation. As such, the one objective here was to demonstrate that our map, unlike the global BII map, does correlate broadly with these other indices as expected (lines 288-290). This broad 'sense-check' is not completely tautological, since *"While expected correlations corroborate our BII assessment, the variability is also important, since the BII provides insights into how diverse species groups respond to different human pressures. It is thus expected to vary from measures that simply quantify aggregated human pressure"* (lines 290-294). Recognising that biomass (BMI) and habitat (BHI) metrics are also derived from human activity (notably, land use), it is perhaps unsurprising that BII correlates slightly better with the human activity metric (HFI) than other derivatives of human activity.

Importantly, the BII is not merely an extension of human activity metrics. Even when two regions have the same land use intensity, one may have a more vulnerable suite of species due to biogeographical factors. For example, we might expect savanna species, adapted to disturbance, to be more resilient to land use change than forest species, which are not disturbance adapted. This has major policy implications for how conservationists design protected area networks and can inform e.g. the land-sparing versus land-sharing strategies. A pure human footprint map captures none of this biogeographical complexity. Furthermore, while the BII itself does not depict changes in biomass, it can be combined with other variables to do exactly that - another paper currently in review at Nature¹⁹ combines the BII with species population density datasets to quantify changes to biomass and energy flows through species, and ultimately to track changes in animal-mediated ecosystem functions. This is one of the reasons the BII is a useful metric compared to others such as HFI, BMI, etc — its high resolution in terms of both species and space means it can also be used as a foundation for other novel metrics that track changes to the biosphere.

We have also added an additional validation test (Extended Data Fig. 6e, lines 304-310), which is based on the IUCN Red List and does not have a similar 'human pressure' spatial input, though of course human activity also impacts species threat statuses.

Second, the plotted figures only show the "average" indices of human modification and habitat indices in each ecoregion. To better represent the data, the variance or some measure of spread should also be illustrated. This will help understand how closely the BII corresponds to the range of measures used.

RESPONSE: Thank you for this suggestion. We have added standard deviations to these plots (new Extended Data Fig. 6a-c).

Quantitative analysis

-since mean \pm 95% CI was calculated across pixels, the authors need to be cautious of spatial autocorrelation effects potentially inflating measures of precision, and affecting statistics calculated. I encourage the authors to estimate sampling distances needed to eliminate spatial autocorrelation, then randomly sample pixels given this distance to calculate the mean and CI estimates.

RESPONSE: Spatial autocorrelation can be an issue when validating a map based on reference data for a subset of locations. These sample estimates serve as an approximation of population values. In our case, however, we make use of population values and not sample values in both our

analyses and the validation. Thus, spatial autocorrelation is not an issue. These confidence intervals reflect confidence around expert scores (based on the variability of scores between experts; lines 281-283 and 901-903), not variance attributed to cross validation or bootstrapped sampling.

Did authors account for test assumptions when conducting correlations and t-test? Independence is already questionable, but do other assumptions hold? Did the authors do any diagnostics? Some of the relationships look potentially heteroskedastic, for example, meaning the error structure varies (which violates a lot of statistical model assumptions). Perhaps nonparametric tests are more relevant.

RESPONSE: Since we are aiming to assess correlations (as opposed to causal relationships) between BII and other measures of human pressure, we have decided it is more appropriate to perform Spearman's correlation tests instead of linear regressions (noting that the trends and their significance remain the same) (lines 936-938). We have also replaced the t-test with a non-parametric equivalent Mann Whitney U-test (again noting that it does not change the result; lines 941-943).

References

1. Balvanera, P. *et al.* Interconnected place-based social–ecological research can inform global sustainability. *Curr. Opin. Environ. Sustain.* **29**, 1–7 (2017).
2. Sievers, E., Spierenburg, M., Jhagroe, S. S. & van Oudenhoven, A. P. E. Place-based knowledge transfer in a local-to-global and knowledge-to-action context: key steps and facilitative factors. *Ecol. Soc.* **29**, (2024).
3. Clements, H. S. *et al.* The bii4africa dataset of faunal and floral population intactness estimates across Africa's major land uses. *Sci. Data* **11**, 191 (2024).
4. Norström, A. V. *et al.* Principles for knowledge co-production in sustainability research. *Nat. Sustain.* **3**, 182–190 (2020).
5. Martín-López, B., Balvanera, P., Manson, R., Mwampamba, T. H. & Norström, A. Contributions of place-based social-ecological research to address global sustainability challenges. *Glob. Sustain.* **3**, e21 (2020).
6. Reyers, B., Moore, M. L., Haider, L. J. & Schlüter, M. The contributions of resilience to reshaping sustainable development. *Nat. Sustain.* **5**, 657–664 (2022).
7. Buschke, F. T. *et al.* Make global biodiversity information useful to national decision-makers. *Nat. Ecol. Evol.* **7**, 1953–1956 (2023).
8. Scholes, R. J. & Biggs, R. A biodiversity intactness index. *Nature* **434**, 45–49 (2005).
9. Palacio, R. D. *et al.* The global influence of the IUCN Red List can hinder species conservation efforts. *Authorea Prepr.* 1–22 (2023).
10. Hemming, V., Burgman, M. A., Hanea, A. M., McBride, M. F. & Wintle, B. C. A practical guide to structured expert elicitation using the IDEA protocol. *Methods Ecol. Evol.* **9**, 169–180 (2018).
11. Burgman, M. A. *et al.* Expert status and performance. *PLoS One* **6**, 1–7 (2011).
12. Singh, G. G. *et al.* Group elicitation yields more consistent, yet more uncertain experts in understanding risks to ecosystem services in New Zealand bays. *PLoS One* **12**, e0182233 (2017).
13. Morgan, M. G. Use (and abuse) of expert elicitation in support of decision making for public policy. *Proc. Natl. Acad. Sci. U. S. A.* **111**, 7176–7184 (2014).
14. Ellis, E. C., Goldewijk, K. K., Siebert, S. & Lightman, D. Anthropogenic transformation of the biomes, 1700 to 2000. *Glob. Ecol. Biogeogr.* **19**, 589–606 (2010).
15. Ellis, E. C., Beusen, A. H. W. & Goldewijk, K. K. Anthropogenic biomes: 10,000 BCE to 2015 CE. *Land* **9**, 8–10 (2020).
16. Klein Goldewijk, K., Beusen, A., Van Drecht, G. & De Vos, M. The HYDE 3.1 spatially explicit

- database of human-induced global land-use change over the past 12,000 years. *Glob. Ecol. Biogeogr.* **20**, 73–86 (2011).
17. Watson, J. E. M. & Venter, O. Mapping the continuum of humanity's footprint on land. *One Earth* **1**, 175–180 (2019).
 18. Trisos, C. H., Auerbach, J. & Katti, M. Decoloniality and anti-oppressive practices for a more ethical ecology. *Nat. Ecol. Evol.* **5**, 1205–1212 (2021).
 19. Loft, T. *et al.* Energy flows reveal declining ecosystem functions by animals across Africa. *Nature preprint*, 1–25 (2024).

Editor's comments

Dear Hayley

Your manuscript, "Sub-Saharan Africa's biodiversity intactness assessed with place-based knowledge", has now been seen by the original three referees. You will see from their comments below that while they continue to find your work of interest, some important points are raised. We are interested in the possibility of publishing your study in Nature, but would like to consider your response to these concerns in the form of a revised manuscript before we make a final decision on publication.

We therefore invite you to revise your manuscript taking into account the referee points. For publication in Nature, we will need you to better document potential error and bias in the results, as requested by referee 3, and we will need you to do this in the main text (as opposed to the Methods section). We will also need you to assess whether spatial autocorrelation is at play and if so consider the consequences for confidence intervals, as also requested by referee 3. And we will need you better present the variance in the results and explore some of the less expected outcomes, as requested by referee 1. We will also need you to document all of the code and annotate it.

You will also need to make some editorial changes to your paper so that it is as brief as possible and complies with our Guide to Authors. We also strongly suggest that your revised manuscript has tracked changes, which is increasingly requested by referees to aid in their re-review.

RESPONSE: Thank you for the opportunity to respond to the reviewers' remaining concerns, which we address point-by-point below.

In summary, we have revised the 'Validation and uncertainties' section of the results in the main text to better document the potential that bias in our approach, particularly systematic bias in our expert group, could lead to error in the overall assessment, as requested by Reviewer 3. We explain the importance of thus critically evaluating our assessment in numerous ways, and that the resulting corroboration with diverse other datasets suggests our assessment was robust to potential bias.

We have also added detail in the results section to better present the variance in the IUCN Red List comparison, as requested by Reviewer 1. We summarise insights into some of the less expected outcomes, where species have outlying BII values relative to their level of threat, which we explore in a new Supplementary Table. We find that these 'outliers' are driven by poorly known species (particularly when localised to protected areas of questionable efficacy), as well as differences in the purpose of a BII vs risk-of-extinction assessment.

Finally, we consulted two geospatial data scientists to interrogate whether spatial autocorrelation was influencing our reported confidence intervals and concluded that this is not the case, providing a detailed explanation in response to Reviewer 3. We believe this concern arose from us not being sufficiently clear in the paper how we calculated the reported confidence intervals. We have added text to clarify what the reported uncertainty in our assessment represents, in the results of the main paper as well as in the methods.

We have made some editorial changes to the paper to improve conciseness. All changes are track changed. We have also uploaded all our BII assessment code to *Figshare*, including for plotting, and annotated it. We have also submitted our source data.

Referees' comments:

Referee #1 (Remarks to the Author):

Firstly, to avoid confusion, I was reviewer 1 in the first round. I would like to thank the authors for their comprehensive approach to responding to my and other reviewer's comments. I believe the changes they have made have significantly improved the manuscript. I am aware that my suggestions for validation steps has required quite a lot of new work to undertake, and I believe have helped inform the analysis. I am boardly happy with this, and the manuscript remains easy to read throughout.

RESPONSE: Thank you for the positive feedback. We felt the changes suggested by the reviewer in the first round significantly improved the paper, and are happy to hear that the reviewer concurs. Thank you again for the thoughtful suggestions.

One thing the new analysis (and Figure S6 in particular) highlights is the considerable noise in these species-level assessments. I agree with the authors that it is reassuring that IUCN Red-list categories can be differentiated statistically in the BII scores, and this gives me some confidence in the remaining results. However, I think the authors need to be rather more transparent that this statistical significance is actually based on very tiny average differences in BII between groups - and there seem to be species IUCN list as LC that are apparently 80% depleted, and CR species that have only lost 5% of BII. Given that IUCN Red listing is also an expert based process, and despite the author's arguments in their response letter that there may not be be congruence between BII and RL status, I simply don't buy that all the species that make this possible are a result of those processes. Maybe the SM could pick some of those examples and transparently explain what is going on for a few idenitified species? The existence of such wide spread of BII scores within RL categories suggests to me that while on average overall patterns may be identified from these methods, we'd be really unwise to read anything into the details of each species. I think this uncertainty and the species-level errors that I'm sure are here should be given a bit more prominence. I know the authors want to present their results in the best possible light, but I think the overall aptterns identified will hold while they remain transparent about what can't be read into these data.

RESPONSE: We agree with the reviewer that the strength of the BII is in understanding patterns of intactness based on large numbers of species (hence being a biodiversity index), and that caution should be applied when reading into species-specific results, particularly for lesser-known species where knowledge gaps would have impacted the robustness of both BII and IUCN assessments. This is, to our knowledge, a first attempt to disaggregate the index to species-level and it was undertaken purely as a validation exercise, which we agree with the reviewer was a very useful additional analysis to provide confidence in the robustness of our aggregated index.

Thank you for the new suggestion to interrogate some of the 'outlier' species. This was a very informative exercise, which we present in a new supplement (Supplementary Table 1). In summary, we randomly selected 10 Critically Endangered (CR) species with surprisingly high (upper

10th percentile) BII values and 10 Least Concern (LC) species with surprisingly low (bottom 10th percentile) BII values. We reviewed each species' IUCN Red List assessments, and the human activities that our BII assessment documents across each species' range. Unexpected BII results for these outlier species appear to have resulted from (1) knowledge gaps (e.g., half of these species are poorly known; several are localised to protected areas where we may overestimate effectiveness) and (2) differences in the purpose of assessments of intactness versus threat of extinction (e.g., a species can have lost >50% of their pre-industrial populations while still remaining abundant within many protected areas and thus not considered threatened).

We summarise these results in the main paper as follows: *“While these broad trends demonstrate the robustness of our approach, large within-category variation results in relatively small absolute differences in mean BII between threat categories. A review of Red List assessments for a random sample of outlier species suggests this variation arises from both knowledge gaps and differences in the purpose of assessments of intactness versus threat of extinction (Supplementary Table 1). Importantly, the BII is not intended as a species-level index and caution should be exercised when considering species-level results beyond general trends.”* (lines 289-295 in the untracked pdf version of the paper)

We trust that this additional text in the main paper results addresses the reviewer's feedback that we should (1) be more transparent about small absolute differences between groups caused by the variance within groups; (2) interrogate sources of this variance; and (3) caution against reading too much into the details of these species-level results.

I also thank the authors for providing the github repository. However, this does not include the code used to generate the plots or analyses in the text - it seems only to be the the code for generating the 1 or 8km rasters, extracting covariate datasets and processing country-level analyses. I can't see any of the statistical analyses or plotting code. It is also seriously lacking in comments.

RESPONSE: Thank you for this feedback. The *GitHub* repository only includes the code to produce the land cover map used in this study. The code to produce the BII maps is provided in *Figshare*, where we have now also uploaded the code to produce the plots where relevant (some of the simpler plots were produced in excel). This is now made clear in our code availability statement (lines 936-940 in the pdf), and we have also submitted source data for all figures. We have worked to improve the clarity of comments in all our code and in ReadMe documents.

Referee #1 (Remarks on code availability):

The code as currently provided probably enable a very persistent analyst to reproduce the core workflow for generating maps. It does not seem to allow the reproduction of the figures and statistics described in the manuscript. It is not well commented.

RESPONSE: Thank you for taking the time to look through our code. Please see our above response.

Referee #2 (Remarks to the Author):

Clements et al. are appreciated for revising their manuscript (Sub-Saharan Africa's biodiversity intactness assessed with place-based knowledge). I am glad and thank the authors for addressing all my concerns and suggestions, particularly the addition of caveats about the place-based approach they used in the data collection, rearrangement of the results (as I suggested), and revision of the discussions accordingly. While I strongly think the manuscript has improved, some parts are somewhat too long to follow in the method section. I have highlighted that and few other things in the comments below. (Please note that my comments were based on the line numbers in the untracked PDF file.).

RESPONSE: Thank you for the positive feedback on our revisions. The reviewers' previous feedback was really helpful and appreciated. With respect to the new suggestions, we have revised the manuscript to shorten the method sections that were flagged as too long, while retaining the key elements requested by the reviewers on the last round. We address each of the reviewer's points of feedback below.

L1: I really like this revised title - it captures the study well.

RESPONSE: Many thanks for this positive feedback!

L210: Should you have "followed by grassland (68%) biomes" here again when the highest and least have been mentioned?

RESPONSE: We would like to mention the two least-intact biomes here, as we did for the most-intact (arid) biomes. We have reworded the sentence to clarify its meaning: "Comparing sub-Saharan Africa's major biomes, BII is highest in the more arid biomes (desert—86% and shrubland—83%), and lowest in the fynbos (a Mediterranean-type ecosystem and biodiversity hotspot—56%) and grassland (68%) biomes" (lines 191-193 in the untracked pdf version of the paper).

L288-321: I do not really get the narratives here and what exactly they are addressing. Or you probably need to revise it for conciseness?

RESPONSE: We have substantially revised the 'Validation and uncertainties' section to be clear that its purpose is to interrogate the validity of our assessment, including a clearer introductory paragraph on the purpose of the validation exercise (lines 259-266) and a concluding paragraph on the findings (lines 307-311 in the pdf). We have also revised the section for conciseness throughout, though the total word count has not decreased given the need to include an additional assessment in the IUCN paragraph in response to Reviewer 1.

L636: No file was labelled "Supplementary Information 2".

RESPONSE: We have revised all reference to this supplement to be "Supplementary Table 2", which is now labelled as such in the Supplementary Information document.

L709-710: No file was labelled "Supplementary Information 2".

RESPONSE: We have revised all reference to this supplement to be “Supplementary Table 2”, which is now labelled as such in the Supplementary Information document.

L625-782: I believe you have added new paragraphs here, possibly because other reviewer(s) called for that, but this is rather too long, particularly for the main manuscript. You might need to break it into sub-sections and maintain brevity for clarity.

RESPONSE: We have restructured this section and reduced it substantially (from ~1900 words to ~1000 words) to improve clarity (lines 597-680 in the pdf), while retaining the key elements requested by the reviewers on the last round.

L814, 861 and other places: “Supplementary Information” not “Supplementary Information 2”. You might check the file name for this.

RESPONSE: In line with Nature’s formatting guidelines, we have now provided all Supplementary Information in one word document, with a table of contents, and revised all references to this supplementary information in the manuscript accordingly (as Supplementary Table 1, Supplementary Table 2, or Supplementary Methods 1).

L866: Supplementary Information

RESPONSE: Please see our above response.

L811-890: Too long to understand. Summarise for brevity and clarity.

RESPONSE: We have revised this section to improve clarity, reducing it by 150 words (lines 709-772 in the pdf)

L977-1031: Thank you for providing these caveats. I believe these will help in building on the place-based approach in this study and trigger new innovations for monitoring biodiversity.

RESPONSE: Thank you for this positive feedback on this new section.

L1036: remember to add this identifier

RESPONSE: We have provided a link to the dataset on *Figshare*, which we will replace with the doi prior to paper publication (line 932 in the pdf).

Referee #2 (Remarks on code availability):

Unfortunately, I did not go through the codes this time. So, I hope other reviewers would do this.

Referee #3 (Remarks to the Author):

I thank the authors for their thorough work in revising the manuscript and generously considering the comments from all three reviewers. I have gone through the responses to my earlier concerns and I was largely pleased with the extra details given to the i) recruitment of experts, and ii) the process of elicitation. Overall I found that my concerns were ameliorated, since it does seem like the authors did a thorough job following best practices in doing good expert elicitation, and they have provided much more detail highlighting that. However, I have lingering concerns about the implications of the elicitation in how the results are presented. Notably, because of the pragmatic decisions that authors needed to take, there are potential biases introduced in the data.

RESPONSE: Thank you for this positive feedback. We are pleased to hear that our revision ameliorated your concerns regarding our expert elicitation approach, and that our revisions have helped clarify that we followed best practices in expert elicitation. We address the lingering concerns point-by-point below.

I appreciate the difficulties in recruiting experts generally, and even more so for such an ambitious undertaking (200 experts across taxa for almost a continent). I applaud the authors in their efforts to include experts that otherwise routinely are left out because they do not fit classic definitions of “expert” based on status and credentials. In general I found the authors’ commitment to expert elicitation processes (from recruitment to execution) reassuring, however I would caution against any notion that adopting best practices mitigates all issues with error in results.

RESPONSE: Thank you for this appreciation of our efforts to include a diverse group of experts. We agree that adopting best practices does not necessarily mitigate all possible errors and have revised the main text of our manuscript to be more transparent and reflective about potential bias and error in the results.

Specifically, we have revised the first paragraph of the ‘Validation and uncertainties’ section of the results as follows: *“A challenge with broad-scale biodiversity assessments is the feasibility of performing independent validations to document the degree of error, particularly in data-poor regions. Notably, errors can arise from expert biases, which we mitigated by adopting evidence-based guidelines to improve elicitation rigour²⁹. However, it was not possible to eliminate potential error arising, for example, from knowledge gaps and potential systematic biases in the expert group (see Caveats section). To assess these potential errors, we critically evaluated our assessment in multiple ways, including the degree of consensus between experts, and corroboration between our results and other assessments of human pressure and threat.”* (lines 259-266 in the untracked pdf version of the paper)

We have also included a new concluding paragraph in this section: *“Taken together, these diverse comparisons corroborate our BII assessment, in contrast to the existing global BII model which lacks such corroboration. The reported uncertainty around our BII scores gives an indication of uncertainty in the underlying expert scores, noting that this does not account for potential systematic biases and other unknown potential sources of error in our assessment.”* (lines 307-311 in the pdf)

In addition, we have added text to the first paragraph of the ‘Caveats’ section (lines 867-888 in the pdf) to note the additional sources of potential bias that the reviewer raises below (please see our

specific responses to those points below) and revised the concluding sentence to now read: ***“Given the uncertainties inherent in expert elicitation, applications of this BII assessment should take note of the confidence intervals around mean estimates which reflect known uncertainty between experts, while keeping in mind potential unknown uncertainties such as possible systematic bias in our expert group that are not reflected in these confidence intervals.”*** (lines 884-888 in the pdf)

I do think given the extra text that the authors contributed significant effort to recruit relevant experts. However, their new text reveals that they could not fully mitigate selection biases in expert inclusion into the process. As other reviewers have also pointed out, there are geographic disparities in expert participation (notably for experts from Africa, most are from South Africa). I wonder if there are more selection issues present, notably in socioeconomic and employment variables. For example, it is widely known that South Africa is the productive (in terms of academic publications) of African countries and potentially also most likely to be similar to experts from the Global North.

RESPONSE: We do not think these aspects introduced significant bias but understand that the reviewer uses this as an example of where potential bias may have influenced our results, and agree this is possible. We have added some text to better acknowledge the risk of expert bias in the main text (lines 259-266 in the pdf), as quoted above, as well as the ‘Caveats’ section: ***“Our data aggregation approach reduces the effect of expert biases, assuming such biases are independent rather than systematic (e.g., no groupthink). However, expertise was overrepresented for certain geographies (e.g., southern Africa), nationalities (e.g., South Africa), taxonomic groups (e.g., large mammals) and professions (e.g., researchers), which may have resulted in unknown systematic biases in aggregated expert scores.”*** (lines 872-877 in the pdf)

Further, some details about the difficulties in recruiting and retention of experts make me think there may be some systematic selection biases in the experts that provided responses. As an example, the authors note that not all 200 experts were able to take part through the conclusion of the exercise. Attrition often follows specific systematic and particular causal reasons, and attrition bias is a pernicious effect that can render the accuracy and validity of measures suspect. Often participants (or in this case, experts) with other commitments, less economic freedom or roles focused on research having to leave early. If, in these cases, the people who left were experts with more on-the-ground experience in wildlife management, might the authors’ quantitative assessments underrepresent wildlife abundances? This is my main point – that commitments to best practices are great, and pragmatic decisions are understandable, but neither eliminates or mitigates error (especially systematic error).

RESPONSE: We fully agree with the reviewer’s main point – the need for pragmatism while following best practices, but recognizing this does not eliminate potential systematic biases. As a small point of clarification: 200 experts did take part through to the conclusion of the exercise, as explained in our previous round of responses: ***“Of 266 experts that agreed to participate in the elicitation, 66 (25%) did not see the process through to completion”*** (see lines 636-637 in the pdf). We agree with the reviewer that certain kinds of people may have been more likely to see the process through to completion, and we reflect on the potential sources of participation bias in the paper’s Caveats section: ***“Limitations to diverse expert participation likely include digital inequities, lack of incentive for non-researchers to participate in research, and language barriers (particularly in Francophone countries).”*** (lines 877-878 in the pdf). We trust that our additional text in the paper results (quoted above) together with this text better acknowledges potential systematic biases in our assessment.

I appreciate the practical limitations of this study (such as reaching saturation of relevant experts). I don't think every study needs to be perfect in order to be published, however where there are practical considerations that mean a study falls short of a design that would ensure the scope and accuracy, then the study needs to reflect on the errors introduced by the practical considerations of the design. Any study with the scope of this one is prone to error propagation and/or systematic error and this needs to be reflected in the results and discussion.

RESPONSE: As noted above, we have added additional text to reflect on potential bias and error that could have arisen from our approach in the results section (lines 259-266 and 308-311 in the pdf), and also addressed the reviewer's specific lingering concerns in the Caveats section (lines 867-922 in the pdf), as quoted in response to the specific points raised above. However, given the correspondence between our results and several other assessments as reflected in the additional validation analyses we included in our previous round of revisions, we do not feel these potential errors are substantial enough to undermine the contribution of our study, especially when considered in relation to other published work (e.g., the existing global BII map with known validity issues¹) which we feel our work substantially improves upon.

1. Martin et al. The biodiversity intactness index may underestimate losses. Nat. Ecol. Evol. 3, 862–863 (2019).

I do also appreciate that the authors have taken the time to further do validation tests (such as compare against IUCN datasets) but I think this may be yet another database that potentially lines up with some of the systematic biases I'm talking about. That is, my understanding is that IUCN data is usually the result of researchers from the Global North and researchers that conform to traditional Global North standards of expertise (such as credentials). So in some ways it's odd to validate a dataset supposedly built on local and place-based knowledge with more traditional datasets. Validation is a very difficult challenge in these settings and I do not envy the authors in wrestling with this. I say all this not to say that the authors are wrong but that I don't think there are easy analytical and quantitative methods to validate the results or mitigate the potential problems of systematic error. Rather the authors need to acknowledge potential sources and consequences of error (and I would prefer this would be in the results and discussion rather than the methods, but I leave that to the authors and editors).

RESPONSE: We fully agree with the reviewer that the IUCN Red List also has a range of potential biases and errors associated with it, as noted in our response to Reviewer 1 in the previous round. We also agree that validation is extremely challenging in these settings, where validation datasets can share information / sources of error, hence our decision to approach validation in multiple ways (including critical review between experts within our elicitation process; comparing our aggregated results with various other human pressure datasets and our species-level results with the Red List; comparing our aggregated results with previous BII assessments, and considering the alignment of our results with other literature in the discussion). We think these diverse forms of critically evaluation of our assessment are very important given the risk of biases and systematic error raised by the reviewer, while also acknowledging that we cannot assume these other (widely used) datasets are without their own biases and errors. As outlined in our responses above, we now have tried to acknowledge potential error and bias much more clearly in the main text (lines 259-266 and 308-311 in the pdf), and Caveats section (lines 867-922 in the pdf), while not undermining the potential contribution we feel our study makes in relation to current datasets in

use. It certainly isn't perfect, but we do feel our assessment is a significant additional source of data and in many ways an improvement on the main datasets currently in use for biodiversity assessments in the relatively data-poor sub-Saharan African region.

Importantly, reflecting on the potentials for systemic error mean that the authors need to do more than note uncertainties reflected in the confidence intervals around mean estimates (lines 998-999, and repeated in the response to my earlier review). Confidence intervals only reflect potential effects of random error and not systematic error from issues like selection effects. Systematic errors have potential directional consequences to estimates (such as the example I give above regarding the attrition bias possibility).

RESPONSE: We fully concur with this point. To better reflect this, we now state clearly at the start of the results and in the relevant figures what uncertainty is represented by the confidence intervals: "All reported uncertainties around BII values are based on 95% confidence intervals around average expert estimates of intactness in the bii4africa dataset" (lines 151-153 in the pdf) – i.e. that they reflect the variation between experts. We also now state in the results (lines 308-311 in the pdf) that "The reported uncertainty around our BII scores gives an indication of uncertainty in the underlying expert scores, noting that this does not account for potential systematic biases and other unknown potential sources of error in our assessment." and in the Caveats section (lines 884-888 in the pdf): "Given the uncertainties inherent in expert elicitation, applications of this BII assessment should take note of the confidence intervals around mean estimates which reflect known uncertainty between experts, while keeping in mind potential unknown uncertainties such as possible systematic bias in our expert group that are not reflected in these confidence intervals."

Beyond these issues of translating the challenges of expert elicitation to the results gathered, I also have some lingering concerns about some of the specific results, especially quantitative results.

First, before my concerns, I'm glad about some of the changes made to the analyses. Regarding my comments on correlations and t-tests – I was happy to see that non-parametric measures generated the same conclusions as previous. My comments on diagnostics had to do with parametric correlative tests (regression is not a causal analysis, but a correlative one).

RESPONSE: Thank you for this positive feedback on our revised statistical analyses.

Second (and somewhat minor), my understanding of the authors response to my comment about the BII results being somewhat tautological is that instead of validating the BII, the authors are showing that their regional BII does better than the global BII. I do think this is a worthwhile result and I apologize for not understanding it on first read. All responses about validation, importantly, are about comparing the authors' map with indicators of human impact and landuse change (or consequences of impacts, such as species threat status), so they all do share information. I think it would be good to make this more explicit (that the focus was to show the lack of relationship with the global BII). It might help prevent misreads from people like me.

RESPONSE: Thank you for this feedback. To make this more explicit we have clarified the wording in the main text by adding the following sentence to the 'Validation and uncertainties' section: "Taken together, these diverse comparisons corroborate our BII assessment, in contrast to the existing global BII model which lacks such corroboration" (lines 307-308 in the pdf).

More importantly, I do not agree with the authors response about dealing with spatial autocorrelation. Autocorrelation is about shared information and how this affects estimates. Whether or not the data comes from reference data or expert input, data from nearer sites are likely to share information that sites farther away. In fact, if expert scores did not reflect this spatial similarity, I would be highly suspect of the accuracy of expert data (at least in principle). In computing confidence intervals, if samples are included that have shared data (sites closer together are more likely to be more similar) then the reported confidence intervals are likely too narrow due to pseudoreplication. That is, the reported values would reflect overconfidence in the scores as the authors are representing shared information as independent information. Again, I think the authors need to assess whether spatial autocorrelation is present and if necessary recalculate their confidence intervals.

RESPONSE: We have now consulted two geospatial data scientists to interrogate whether spatial autocorrelation is likely to be influencing our reported confidence intervals and have concluded that this is not the case. We believe these concerns arise from us not being sufficiently clear regarding what the confidence intervals (and associated samples) we report represent. We have now revised the manuscript to make this clearer, as detailed below.

Spatial autocorrelation is an issue that needs to be considered in measures of uncertainty when data come from a sample of sites, since sites closer together can share information, as the reviewer states. In our case, input data come from a sample of experts, who were asked to provide intactness estimates for contextually generalised landscapes and groups of species. The sample size represented in each confidence interval is the number of experts that provided an estimate (as opposed to the number of sites from which such estimates were obtained). This measure of uncertainty gives us an indication of confidence that the mean estimate across a sample of experts reflects the true mean across all experts (since, as we explain in our previous round of responses, it was impractical to identify and involve all possible experts).

We understand the reviewer's concern that, if two experts were informing their intactness estimates based exclusively on information from sites that were close together, their estimates may be more similar than those of experts informing their estimates from sites further away, meaning there is pseudo replication and an inflated sample size, thus inflating our measure of confidence in the mean estimate. However, to test for such spatial autocorrelation, we would need to attribute expert knowledge and resulting estimates to a specific site, which is neither possible nor appropriate. Experts' estimates were informed by their unique, cumulative life experiences, including observing diverse species in representative landscapes (which typically encompassed many sites observed over many years), as well as engaging with colleagues, reading literature, etc. These experts were specifically asked to use their knowledge to provide contextually generalised estimates of how a type of land use would influence a type of species, informed by the different specific landscapes within which they have worked. Most experts have worked across multiple different sites, and based on our diverse pool of experts, we are confident that there are no major concentrations of experts in specific sites. Even in the unlikely case that two experts were informing their estimates based on experience with the same single species in the same single site at the same point in time, discounting these experts to pseudo-replicated samples would be discounting all other dimensions of their experience likely to influence their estimates (e.g., life experience, professional training, engagement with literature and people, etc). We are therefore confident that expert's multi-faceted knowledge and resulting estimates are not tied to specific

geolocations, nor is there a high concentration of experts in particular single sites. Converting the sample of experts to a sample of geographical sites is therefore not appropriate, meaning spatial autocorrelation is not an issue in the confidence intervals around expert estimates.

Rather, in our BII assessment we make use of the confidence intervals around expert estimates to reflect uncertainty between experts. To do this, in addition to our 'main' BII map based on average expert estimates, we produced a BII map based on the upper confidence limits of expert estimates and a BII map based on the lower confidence limits of expert estimates. When we report uncertainty around a given BII score (e.g., for the region, for a country, or for a land use), we are deriving this uncertainty from the confidence limit BII maps, as opposed to deriving it from a measure of data variance within the main BII map. Because our reported confidence intervals around the BII values are not based on a sample (and rather the full population of pixels in the confidence limit BII maps), spatial autocorrelation is not an issue, unless it is an issue in the underlying input data, which we address in the previous paragraph.

The reviewer's concerns about spatial autocorrelation highlighted however that we need to be much clearer throughout the paper regarding what our reported confidence intervals represent. We have addressed this through additional text at first mention of these confidence intervals in the results (*"All reported uncertainties around BII values are based on 95% confidence intervals around average expert estimates of intactness in the bii4africa dataset"*), as well as in the legend of Figure 2 and Extended Data Table 1, where this uncertainty is depicted. We provide further details on our approach to quantify uncertainty in the methods (lines 786-789 and 804-806 in the pdf): *"To reflect uncertainty, we similarly estimated 'lower limit' and 'upper limit' BII scores for each pixel based on the lower and upper limits of the 95% confidence intervals around average intactness scores I_{ik} . Lower-limit-BII and upper-limit-BII scores were determined for (a) terrestrial vertebrates and plants; (b) vertebrates; and (c) plants. ... We quantified the uncertainty around these BII values by averaging upper-limit BII scores and lower-limit BII scores across all relevant pixels."*